# Open-Set Domain Adaptation Under Background Distribution Shift: Challenges and A Provably Efficient Solution

**Shravan Chaudhari**                                                                    *schaud35@jh.edu*
*Department of Computer Science, Johns Hopkins University, Baltimore, MD, USA.*

**Yoav Wald**                                                                    *yoav.wald@technion.ac.il*
*Faculty of Data and Decision Sciences, Technion, Haifa, Israel.*
*Center for Data Science, New York University, New York, NY, USA.*

**Suchi Saria**                                                                    *ssaria1@jhu.edu*
*Department of Computer Science, Johns Hopkins University, Baltimore, MD, USA.*
*Bayesian Health, New York, NY, USA.*

**Reviewed on OpenReview:** *https://openreview.net/forum?id=uAJDta7VaQ*

## Abstract

As we deploy machine learning systems in the real world, a core challenge is to maintain a model that is performant even as the data shifts. Such shifts can take many forms: new classes may emerge that were absent during training, a problem known as open-set recognition, and the distribution of known categories may change. Guarantees on open-set recognition are mostly derived under the assumption that the distribution of known classes, which we call *the background distribution*, is fixed. In this paper we develop CoLOR, a method that is guaranteed to solve open-set recognition even in the challenging case where the background distribution shifts. We prove that the method works under benign assumptions that the novel class is separable from the non-novel classes, and provide theoretical guarantees that it outperforms a representative baseline in a simplified overparameterized setting. We develop techniques to make CoLOR scalable and robust, and perform comprehensive empirical evaluations on image and text data. The results show that CoLOR significantly outperforms existing open-set recognition methods under background shift. Moreover, we provide new insights into how factors such as the size of the novel class influences performance, an aspect that has not been extensively explored in prior work. Code is available at `https://github.com/Shra1-25/CoLOR`.

## 1 Introduction

Adapting Machine Learning models to shifts in the data distribution is pivotal to ensuring their robustness and safety in real world applications (Quinonero-Candela et al., 2008; Koh et al., 2021) for fields like healthcare (Finlayson et al., 2021), autonomous driving (Filos et al., 2020; Wong et al., 2020) and more broadly in computer vision (Bendale & Boult, 2016). Maintaining robustness to distribution shift in classification problems includes both identifying familiar objects under new conditions, while detecting the long tail of object categories that has not been observed in the past. In this work we study the problem of adaptation when these two types of distribution shift occur concurrently. Specifically, we address the emergence of novel categories (Panareda Busto & Gall, 2017) alongside shifting distributions of known classes, a scenario referred to as Open-Set Domain Adaptation (OSDA).

Let our training data be $S_{\mathcal{S}}$. The core task here is to detect the novel classes in target data $S_{\mathcal{T}}$ collected under some conditions different from those observed at training time (Panareda Busto & Gall, 2017) while maintaining good performance over the existing classes. The emergence of a novel class in itself constitutes a shift between $S_{\mathcal{S}}$ and $S_{\mathcal{T}}$, however, in real world scenarios *it is likely not the only distribution shift that occurs*

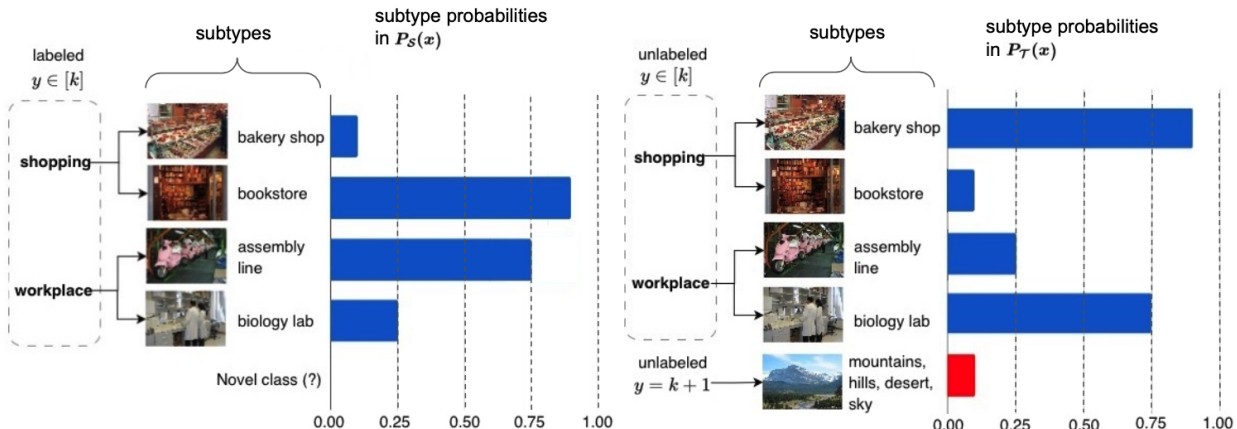

Figure 1: An example of OSDA with background shift. There are two known classes, 'shopping' and 'workplace'. Each class can further have subtypes, but the learner has no access to the labels of these subtypes. **Left:** Source distribution with all samples from $k = 2$ known categories with distribution $P_\mathcal{S}(x)$. **Right** Target distribution $P_\mathcal{T}(x)$ with samples from known categories as well as new classes (mountains, hills, etc.). $P_\mathcal{T}(x)$ consists of varying proportions of known categories and their latent subtypes along with shift due to emerging novel classes. Unlike pure label, covariate or conditional shift, background shift allows any of $P_\mathcal{S}(y)$, $P_\mathcal{S}(x|y)$ or $P_\mathcal{S}(x)$ to change in $P_\mathcal{T}$, as long as the support overlap and separability assumptions are followed.

(Garg et al., 2022; Wald & Saria, 2023). Our study concentrates on scenarios where the novel class exhibits a notable degree of separability from the known classes, and a distribution shift with overlapping supports between source and target distributions exists within the known classes, which we refer to as 'background shift'. Figure 1 depicts a scenario of background shift within indoor scene images taken from the SUN397 dataset (Xiao et al., 2010) while the novel class belongs to an outdoor scene category. Arguably, this is a very common case in domains such as healthcare, e.g. classifying histopathology slides for known and novel tumor types, where the background shift is due to differences in geographic locations (Pocevičiūtė et al., 2025).

Several aspects of our work are understudied in the OSDA literature; *(i)* formally characterizing the shifts of interest; *(ii)* developing methods with formal guarantees (finite sample bounds in our case) that are also effective in practice; *(iii)* examining the effect of the novel class size on performance. Most research primarily address open-set recognition instead of OSDA because they assume there is no distribution shift for known classes (Ge et al., 2017; Neal et al., 2018; Lin & Xu, 2019; Chen et al., 2021; Zeng et al., 2021; Vaze et al., 2022; Esmaeilpour et al., 2022). Other methods tackle OSDA without formally characterizing 'domain shifts' (Liang et al., 2021; Li et al., 2021). They typically validate their methods empirically on benchmarks such as Office-Home (Venkateswara et al., 2017) or VisDA (Peng et al., 2017), where the shift is simulated by changing the domain from real to sketch images or from synthetic to real world images. However, since these methods lack formal guarantees and rely on limited types of simulated shifts, their effectiveness beyond these datasets remain uncertain. Garg et al. (2022) tackle OSDA under shifting class probabilities in $S_\mathcal{T}$ (label shift) and show improved results compared to prior methods. To this end, our theoretical and empirical results emphasize the regimes where our method is expected to outperform alternatives that do not address background shift in OSDA properly. In particular, we show that our approach is more effective when the novel class constitutes a relatively small proportion of the target distribution—an important scenario in practice. Reliably identifying novel classes enables updating models or deferring to human oversight in safety-critical settings such as healthcare and autonomous driving (Amodei et al., 2016; Hendrycks & Gimpel, 2016).

Our contributions to these aspects of OSDA are thus the following:

- **Leverage and improve constrained learning rule for OSDA:** We develop CoLOR (Constrained Learning for Open-set Recognition), a scalable method for OSDA (section 5) that extends Wald & Saria (2023)'s small-model, no-known-class setting via architectural and algorithmic improvements effective for

large models and high-dimensional image and text data. Our theory explains why and when the method performs favorably in the large-scale regime.

- **Formal results on identifiability under background shift:** We characterize sufficient conditions for identifiability of the novel class and analyze sample efficiency of CoLOR on a problem capturing key aspects of OSDA with expressive models (section 4.2). While Garg et al. (2022) solve OSDA under label shift with assumptions weaker than our separability, we prove these fail under the more general background shift (Lemma 1). In a linear-Gaussian framework, we further characterize regimes where CoLOR significantly outperforms a standard domain-discriminator baseline, especially when novel classes are rare.

- **Comprehensive empirical validation across diverse data modalities and datasets:** Extensive experiments across image and text classification (section 6) confirm our theoretical findings. We observe that background shift substantially degrades OSDA baselines, whereas our enhanced adaptation of the constrained learning rule, CoLOR, achieves superior performance on CIFAR100 (Krizhevsky, 2009), Amazon Reviews (Ni et al., 2019) and SUN397 (Xiao et al., 2010). Notably, CoLOR remains robust even when the novel class constitutes a small portion of $S_\mathcal{T}$, where other methods tend to struggle.

**Paper roadmap.** section 3.1 formally defines the problem; section 3.2 establishes impossibility results and the assumptions under which it is solvable; section 4 introduces a 0–1 loss learning rule in a linear Gaussian setting, and section 4.2 extends it to the overparameterized regime relevant to large-scale models, characterizing when it outperforms standard baselines; sections 5 and 6 present the scalable algorithm and supporting experiments, with limitations discussed in section 7.

## 2 Related Work

Our work is closely related to several lines of work involving novelty detection, summarized below.

**Out-of-Distribution (OOD) Detection and Open Set Recognition:** Most works on detecting novel instances locate novelties within $S_\mathcal{T}$ without adapting to it, using $S_\mathcal{S}$ alone to compute an "anomaly score" $s(\mathbf{x})$ that ranks examples by their likelihood of being novel (Ruff et al., 2021; Esmaeilpour et al., 2022). Open Set Recognition (OSR) (Bendale & Boult, 2016; Ge et al., 2017; Liu et al., 2018; Xu et al., 2019; Saito et al., 2018; Vaze et al., 2022) also detects novelties while classifying known classes correctly, but the learner has access to unlabeled $S_\mathcal{T}$, allowing $s(\mathbf{x})$ to adapt to the specific novel class; some OSR methods further refine $s(\mathbf{x})$ via generative models on known instances (Neal et al., 2018; Kong & Ramanan, 2021). However, both OSR and OOD methods degrade under shift between $P_\mathcal{S}$ and $P_{\mathcal{T},[k]}$ (Schölkopf et al., 2001; Cao et al., 2023), motivating shift-adaptive methods.

**Open Set Domain Adaptation:** Works that seek to adapt to $P_{\mathcal{T},[k]}$, while correctly classifying known instances and detecting novel ones, fall into this category. Recent OSDA works primarily focus on narrow domain shifts such as real-to-sketch or synthetic-to-real (Wen & Brbic, 2024; Panareda Busto & Gall, 2017; Choe et al., 2024; Hur et al., 2023; Zhu et al., 2023), limiting generalizability to broader shifts like background shift; Garg et al. (2022) show these methods are not versatile across datasets and propose a method specifically for label shift. Other works leverage multi-modal foundation models like ChatGPT, DALL-E and CLIP for OSDA (Qu et al., 2023; Esmaeilpour et al., 2022), but their pretraining data is not publicly accessible, making it infeasible to curate truly novel classes or shifts and to evaluate fairly. Most OSDA methods lack formal guarantees or a clear characterization of the shifts they address; well-studied Positive and Unlabeled (PU) learning offers a framework to develop more generalizable methods with guarantees (Garg et al., 2022; Wald & Saria, 2023).

**PU-Learning Under Distribution Shift:** It is formally equivalent to OSDA where k=1. In the absence of distribution shift, effective solutions with formal guarantees already exist, primarily based on adjustments to "Domain Discriminator" models (that are trained to distinguish source and target distributions i.e. $P_\mathcal{S}$ and $P_\mathcal{T}$ respectively) (Garg et al., 2021; du Plessis et al., 2014; Blanchard et al., 2010; Elkan & Noto, 2008). Under distribution shift, domain discriminator type methods still have guarantees when given infinite data (Gerych et al., 2022). In practice, however, we will see that they underperform (table 2), even when the separability assumption (see assumption 1) holds. Garg et al. (2022) propose a $k$-way PU learning based method for identifying unseen classes under special case of label shift among $k$ known classes, given certain assumptions. We show that these assumptions are insufficient to handle the more general case of background

shift, where additional conditions are required for reliable recovery. Furthermore, Wald & Saria (2023) propose an algorithm with finite-sample guarantees and demonstrate its performance on small-scale models. In contrast, our formal analysis targets the large-scale, overparameterized learning regime, where their results no longer apply (Belkin et al., 2019). We extend these ideas to the setting of OSDA under background shift.

## 3 Open-Set Domain Adaptation under Background Shift – Problem Definition and Identifiability

In this section, we review the problem setting and develop the theory that motivates our solution. Section 3.1 reviews the problem definition and section 3.2 discusses conditions for identifiability, such that a learner provided with unlimited data can solve the problem. Notably, we show that conditions studied in prior work for the case of label shift, are insufficient for identifiability under background shift.

### 3.1 Problem Definition

For a prediction task with $k$ classes, we are interested in detecting the emergence of a novel class where $y = k + 1$, while classifying a set of known classes $y = 1, \ldots, k$. To treat this formally, we assume a learner is provided with two datasets, the training set $S_{\mathcal{S}} = \{\mathbf{x}_i, y_i\}_{i=1}^{N_{\mathcal{S}}}$ and an unlabelled target dataset $S_{\mathcal{T}} = \{\mathbf{x}_i\}_{i=1}^{N_{\mathcal{T}}}$. The datasets $S_{\mathcal{S}}$ and $S_{\mathcal{T}}$ are sampled i.i.d from joint distribution over $(\mathbf{x} \in \mathcal{X}, y \in \mathcal{Y})$, $P_{\mathcal{S}}$ and $P_{\mathcal{T}}$ respectively, which take on values in $\mathcal{X}, \mathcal{Y} = [k + 1]$ and $P_{\mathcal{S}}(y = k + 1) = 0$. $P_{\mathcal{T}}$ is a mixture distribution:

$$P_{\mathcal{T}}(\mathbf{x}, y) = (1 - \alpha)P_{\mathcal{T}, [k]}(\mathbf{x}, y) + \alpha P_{\mathcal{T}, k+1}(\mathbf{x}). \tag{1}$$

Here we use the notation $\alpha := P_{\mathcal{T}}(y = k+1)$, $P_{\mathcal{T}, k+1}(\mathbf{x}) := P_{\mathcal{T}}(\mathbf{x} \mid y = k+1)$, and $P_{\mathcal{T}, [k]}(\mathbf{x}, y) := P_{\mathcal{T}}(\mathbf{x}, y \mid y \neq k+1)$. Our goal is to learn a model that minimizes the error over all classes, including the novel $k+1$-th class. All errors will be measured w.r.t $P_{\mathcal{T}}$. To do this, we learn a model $h : \mathcal{X} \to \mathbb{R}^{k+1}$ that gives a score for each class. We also assume that a novelty detection score $h_{\text{novel}} : \mathcal{X} \to \mathbb{R}$, derived from the overall model, is given and we will define a binary decision threshold at 0 where $h_{\text{novel}}(\mathbf{x}) > 0$ means $\mathbf{x}$ is classified as a novelty.[1] For example, $h_{\text{novel}}(\mathbf{x}) = h(\mathbf{x})_{k+1} - \max_{\hat{y} \in [k]} h(\mathbf{x})_{\hat{y}}$ or $h(\mathbf{x})_{k+1}$ when $h(\mathbf{x})$ returns a probability distribution.

The success of an OSDA model may be determined by its classification accuracy on data from $P_{\mathcal{T}}$, $\mathcal{R}_{\mathcal{T}}^{l_{01}}(h) = \mathbb{E}_{y, \mathbf{x} \sim P_{\mathcal{T}}}[\arg \max_{\hat{y}} \{h(\mathbf{x})_{\hat{y}}\} \neq y]$, but since we often wish to emphasize novelty detection and not treat the novel class like any other known class, several other notions of success have been defined in the literature. Our theory focuses on the simple case in which $k = 1$, where such considerations are unnecessary, and standard metrics for binary classification can be used. In our experiments, as we mention again in section 6, we also use the OSCR metric of Dhamija et al. (2018). Below we formally define the problem setting and other performance metrics we use for our theory.

**Definition 1** (Open Set Domain Adaptation with Background Shift). *An OSDA problem with hypothesis class $\mathcal{H}$ is a tuple $\langle P_{\mathcal{S}}(\mathbf{x}, y), P_{\mathcal{T}, [k]}(\mathbf{x}, y), P_{\mathcal{T}, k+1}(\mathbf{x}), \alpha, N_{\mathcal{S}}, N_{\mathcal{T}}, N_{\mathcal{T},0}, N_{\mathcal{T},1} \rangle$, where we are given $N_{\mathcal{S}}$ and $N_{\mathcal{T}}$ i.i.d examples from $P_{\mathcal{S}}$ and $P_{\mathcal{T}}$ (defined in eq. (1)) respectively. The problem undergoes **background shift** whenever $P_{\mathcal{S}}(\mathbf{x}, y) \neq P_{\mathcal{T}, [k]}(\mathbf{x}, y)$.*

*We further define $\beta(h; t) = \mathbb{E}_{\mathbf{x} \sim S_{\mathcal{S}}}[\mathbf{1}[h_{\text{novel}}(\mathbf{x}) > t]]$, $\alpha(h) = \mathbb{E}_{\mathbf{x} \sim S_{\mathcal{T}}}[\mathbf{1}[h_{\text{novel}}(\mathbf{x}) > t]]$ as the False Positive Rate (FPR) and recall respectively, of a novelty detector $h_{\text{novel}}$ derived from $h \in \mathcal{H}$ where $t \in \mathbb{R}$ is a scalar decision threshold. For notational convenience, we will define $\beta(h) := \beta(h; 0)$ and likewise for $\alpha(h)$.*

Note that we use background shift as a general term for shifts in the distribution of known classes, i.e. between $P_{\mathcal{S}}$ and $P_{\mathcal{T}, [k]}$, that includes common instances such as label, covariate, and conditional shift, studied in domain adaptation (Lalou et al., 2025). The term 'background shift' is used instead of generic 'shift', to emphasize the open-set aspect of the problem which is our focus, where the emergence of the novel class (via the addition of $P_{\mathcal{T}, k+1}$) together with the background shift constitute the entire shift between $P_{\mathcal{S}}$ and $P_{\mathcal{T}}$.

Without any assumptions on the relationship between $P_{\mathcal{S}}$ and $P_{\mathcal{T}}[k]$ it is impossible to obtain a guarantee of better than chance accuracy Garg et al. (2022); David et al. (2010). This holds even under strong simplifying assumptions, such as the existence of a model that solves the problem without errors: that is, $\mathcal{R}_{\mathcal{T}}^{l_{01}}(h^*) = 0$ for

---

[1]The threshold value is arbitrary and can be changed depending on the specific setting.

some hypothesis $h^*$ [2]. In this paper we will work with the overlap and separability assumptions Combining the conditions for background with the above assumption about the existence of an oracle $h^*$, we obtain the separability assumption with which we will work.

## 3.2 Necessary and Sufficient Assumptions for OSDA under Background Shift

**Assumption 1** (separability). *There exists $h^* \in \mathcal{H}$ such that $\mathcal{R}_{\text{novel}}^{l_{01}}(h^*) = 0$. Furthermore, it holds that* $\text{Supp}(P_{\mathcal{T},[k]}) \subseteq \text{Supp}(P_{\mathcal{S}})$ *i.e. background shift exists between $P_{\mathcal{S}}$ and $P_{\mathcal{T},[k]}$*

According to this assumption, we consider any background shift between $P_{\mathcal{S}}$ and $P_{\mathcal{T}}[k]$ that maintains support overlap such that $\text{Supp}(P_{\mathcal{T},[k]}) \subseteq \text{Supp}(P_{\mathcal{S}})$. Moreover, both parts of the assumption are rather intuitive, and hold at least approximately for many problems we consider in OSDA or novelty detection, such as detection of novel semantic visual concepts. Let us emphasize two aspects of our assumption.

**Separability and background shift characteristics in known classes.** The separability assumption is required only for the novel class $Y = k + 1$ vs. the known ones, we do not explicitly limit the $k$ known classes to be separable amongst themselves. Such an assumption would have placed the shift between $P_{\mathcal{S}}$ and $P_{\mathcal{T},[k]}$ purely in the realm of covariate shift (Shimodaira, 2000). However, background shift with assumption 1 facilitates more general forms of label shift and covariate shift. In our experiments we mostly create distribution shifts such that $P_{\mathcal{T},[k]}(X|Y) \neq P_{\mathcal{S}}(X \mid Y)$ which follow the definition of background shift that are underexplored in existing works.

**Insufficiency of less stringent assumptions.** While generalization bounds for domain adaptation are well known from seminal works such as Ben-David et al. (2010b), they are less common in the Open Set learning literature, hence let us focus on this aspect of our problem, i.e. detecting the novel class. To the best of our knowledge the only characterized sufficient and necessary conditions for (non-separable) OSDA are those in Garg et al. (2022), which *hold only for label shift*, where it is assumed that $P_{\mathcal{S}}(X \mid Y = y) = P_{\mathcal{T}}(X \mid Y = y)$ for all $y \in [k]$. For this special case, they propose two assumptions which are sufficient (when added on top of the label-shift assumption) to guarantee $h^*$ can be learned from observed data. Their first assumption is *(Strong Positivity)*: there exists $X_{sep} \in \mathcal{X}$ such that $P_{\mathcal{T},k+1}(X_{sep}) = 0$ and the matrix $[P_{\mathcal{S}}(\mathbf{x} \mid y)]_{\mathbf{x} \in X_{sep}, y \in [k]}$ is full rank and diagonal. We show that once more general shifts than label shifts are allowed, e.g. background/covariate shift, this condition is no longer sufficient and in fact no algorithm can guarantee better-than-chance detection. The proof for this is in A.4.

**Lemma 1.** *Let $\mathcal{A}$ be an algorithm for Open-Set Domain Adaptation. There are distributions $P_{\mathcal{S}}, P_{\mathcal{T},[k]}, P_{\mathcal{T},k+1}$ such that the problem satisfies strong positivity, and $\exists h^* \in \mathcal{H}$ for which $\mathcal{R}_{\mathcal{T}}^{l_{01}}(h^*) = 0$, while* $\mathbb{E}_{S_{\mathcal{S}},S_{\mathcal{T}}}\left[\mathcal{R}_{\mathcal{T}}^{l_{01}}(\mathcal{A}(S_{\mathcal{S}}, S_{\mathcal{T}}))\right] \geq 0.5.$

The second assumption proposed in Garg et al. (2022) is separability as defined in assumption 1; however, since their assumption is combined with the label shift assumption, the methods they develop are tailored to that scenario and do not apply to our problem. [3]

Assumption 1 guarantees that given an infinitely large sample and an optimization oracle, we can learn to identify the novelties. It is also desirable to have guarantees on the required sample size. In this paper we focus on the finite sample guarantees of detecting the novel class in the overparameterized regime, since results on classification of remaining $k$ non-novel classes can be obtained from prior work on domain adaptation, see for instance Ben-David et al. (2010a) and the survey of Redko et al. (2020). This is to formally study and understand best practices in learning expressive models for open-set problems under background shift.

## 4 Constrained Open-Set Learning Rules and Domain Discriminators

We start by proposing the underlying statistical learning rule which our method build upon and call it simplified CoLOR. We further analyze and compare it with other standard domain discriminator baseline

---

[2] see Prop. 1 in Garg et al. (2022), Prop. 2.1 in Wald & Saria (2023), or discussion in Bekker et al. (2019) for the results of this flavor.

[3] Lemma 1 has a similar flavor to Prop. 3.1 in Wald & Saria (2023), but it is stronger. Our result shows impossibility under an additional assumption of strong positivity. This is significant in the context we consider here, as Garg et al. (2022) give guarantees on OSDA with label shift under this strong positivity assumption, but our lemma shows that this is impossible under the background shifts we consider.

over a synthetic problem setup using linear Gaussians. We focus on the case $k = 1$, i.e., PU-learning (Bekker & Davis, 2020), to study the novelty detection aspect of OSDA. The domain discrimination baseline trains a classifier, via Empirical Risk Minimization (ERM), to distinguish between $P_\mathcal{S}$ and $P_\mathcal{T}$ such that for a 0-1 loss function $l_{01}$, $h_{\mathrm{DD}} = \arg\min_{h \in \mathcal{H}} \frac{1}{N_\mathcal{S} + N_\mathcal{T}} \big( \sum_{\mathbf{x} \in S_\mathcal{S}} l_{01}(h(\mathbf{x}), 0) + \sum_{\mathbf{x} \in S_\mathcal{T}} l_{01}(h(\mathbf{x}), 1) \big)$. Similarly, the learning rule for constrained learning is

$$h_{\mathrm{color}} = \arg\min_{h \in \mathcal{H}} \frac{1}{N_\mathcal{S}} \big( \sum_{\mathbf{x} \in S_\mathcal{S}} l_{01}(h(\mathbf{x}), 0) \big) \text{ s.t. } \frac{1}{N_\mathcal{T}} \sum_{\mathbf{x} \in S_\mathcal{T}} l_{01}(h(\mathbf{x}), 1) \leq 1 - \hat{\alpha} \tag{2}$$

In practice, these rules are often implemented with the logistic loss or a close variation, e.g. (Kiryo et al., 2017).

As noted in section 2, standard PU-learning methods without distribution shift rely on domain discriminators. Constrained learning rules were studied in Blanchard et al. (2010) and later extended under distribution shift by Wald & Saria (2023) using small models. Their analysis, however, is limited to classical learning regimes that assume infinite data, whereas in practice, we prefer high-capacity models and only have limited data. A regime that is beyond the scope of their theory. In this section, we review both approaches and explain why applying these guarantees to such high-capacity models is non-trivial. Our main result in section 4.2 compares constrained learning and domain discrimination in a simplified overparameterized setting that preserves key aspects of open-set training. The model builds on ideas from generalization to minority subgroups (Sagawa et al., 2020; Wald et al., 2022; Puli et al., 2023; Nagarajan et al., 2021). Empirical results in section 6 corroborate conclusions to which the theory points.

## 4.1 Overparameterization and Finite Sample Analysis

In terms of formal guaranties, it is well known that at the limit of infinite data and a family $\mathcal{H}$ that is large enough, the minimizer $h_{\mathrm{DD}}(\mathbf{x})$ will be proportional to the log-odds ratio, $h_{\mathrm{DD}}(\mathbf{x}) = \log(P_\mathcal{T}(\mathbf{x})/P_\mathcal{S}(\mathbf{x}))$. Under assumption 1, this means that the instances of the novel class will be classified as originating from $P_\mathcal{T}$ w.p. 1, while for $\mathbf{x} \sim P_{\mathcal{T},0}$ ($= P_{\mathcal{T},[k]}$ since $k = 1$) it will be bounded away from 1 since $\mathrm{Supp}(P_{\mathcal{T},[k]}) \subseteq \mathrm{Supp}(P_\mathcal{S})$. Hence, under these favorable conditions, $h_{\mathrm{DD}}$ will classify the novel class vs. the single known class with accuracy 1. In practice, we do not perform perfect optimization with infinitely large datasets and highly expressive models. Therefore, it is desirable to study the sample complexity of different learning rules. Without further assumptions, the sample complexity of the domain discrimination approach resembles those in the domain adaptation literature, which depend on the divergence between $P_\mathcal{S}$ and $P_\mathcal{T}$, e.g. (Ben-David et al., 2010a). Wald & Saria (2023) show that instead of following ERM, solving a constrained learning rule, achieves a generalization bound where the error scales with a divergence $d_{\mathcal{H},\beta}(P_\mathcal{S} \| P_\mathcal{T})$ which is usually considerably smaller than common divergences in domain adaptation [4]. Here, $\beta$ is a constant that depends on the Rademacher complexity of $\mathcal{H}$ and the sample size. Thm 4.3 in Wald & Saria (2023) provide more details.

**Why existing theory may not reflect large-scale training.** As reflected by dependence on quantities like the Rademacher complexity of $\mathcal{H}$, the results on constrained learning apply to a regime of lower capacity models that may not be expressive enough to achieve arbitarily low training loss. However, it is rather common to fit much more expressive models, as they often generalize better in practice. Next, we formally study a simple example of overparameterized models, where the number of parameters is larger than the number of training examples. This is a common toy model used to study deep networks that overfit their training data but still generalize well, also known as "benign overfitting" (Belkin et al., 2019).

## 4.2 Why Should Constrained Learning be Effective? Linear-Gaussian Overparameterized OSDA Example with $k = 1$

To capture key aspects of the problem solved in practice, while maintaining a manageable mathematical analysis, we perform a few simplifications: **Data and models.** (i) We study linear models $h(\mathbf{x}) = \mathbf{w}^\top \mathbf{x}$ for $\mathbf{w} \in \mathbb{R}^d$, where $d > N_\mathcal{S} + N_\mathcal{T}$ so the models are expressive and can interpolate the data; (ii) We focus on a problem with two features (along $\boldsymbol{\mu}$ and $\boldsymbol{\eta}$) and Gaussian noise (see definition 2). Here $\boldsymbol{\mu}$ and $\boldsymbol{\eta}$ denote the two informative directions in the problem: the background shift, i.e. the shift in the known class, is in the

---

[4]Intuitively, $\beta$ is a small number, and $d_{\mathcal{H},\beta}(P_\mathcal{S} \| P_\mathcal{T})$ measures how probable a rare event under $P_\mathcal{S}$, i.e. with probability smaller than $\beta$, can become under $P_\mathcal{T}$

direction $\boldsymbol{\mu}$, and its magnitude is proportional to the norm $r_\mu$. While $\boldsymbol{\eta}$ marks the novel class, and higher $r_\eta$ corresponds to a larger signal for learning the novel class. The covariance matrices (e.g., $I_d - r_\mu^{-2}\boldsymbol{\mu}\boldsymbol{\mu}^\top$) remove the corresponding rank-one components so the problem satisfies assumption 1. This construction keeps the model analytically tractable and this type of simplifications are common in works on benign overfitting and robustness to distribution shifts (Nagarajan et al., 2021; Wald et al., 2022; Muthukumar et al., 2021).

**Definition 2.** *Let $d > 0$, $\boldsymbol{\mu}, \boldsymbol{\eta} \in \mathbb{R}^d$, $\sigma = 1/\sqrt{d}$, $\alpha \in (0,1)$, $r_\mu = \|\boldsymbol{\mu}\|$, $r_\eta = \|\boldsymbol{\eta}\|$. A* Linear-Gaussian PU-learning problem *is an OSDA problem (definition 1) with 1 known class where $P_\mathcal{S} = \mathcal{N}(\boldsymbol{\mu}, \sigma^2(I_d - r_\eta^{-2}\boldsymbol{\eta}\boldsymbol{\eta}^\top))$, $P_{\mathcal{T},0} = \mathcal{N}(-\boldsymbol{\mu}, \sigma^2(I_d - r_\eta^{-2}\boldsymbol{\eta}\boldsymbol{\eta}^\top))$ and $P_{\mathcal{T},1} = \mathcal{N}(-\boldsymbol{\eta}, \sigma^2(I_d - r_\mu^{-2}\boldsymbol{\mu}\boldsymbol{\mu}^\top))$.*

**Simplified CoLOR and Domain Discriminator methods.** (iii) The last simplification regards which optimization problem to analyze. In an overparameterized problem, there are many possible minimizers of the empirical risk and the constrained risk in eq. (3). Therefore to analyze the solutions, it is common to consider $h_{\text{DD}}$ and $h_{\text{color}}$ as max-margin classifiers [5] such that:

$$\mathbf{w}_{\text{DD}} = \arg\min_{\mathbf{w}} \|\mathbf{w}\| \qquad\qquad \mathbf{w}_{\text{color}} = \arg\min_{\mathbf{w}} \|\mathbf{w}\|$$

$$\text{s.t. } \mathbf{w}^\top\mathbf{x} \leq -1 \quad \forall\mathbf{x} \in S_\mathcal{S} \qquad \text{s.t. } \mathbf{w}^\top\mathbf{x} \leq 0 \quad \forall\mathbf{x} \in S_\mathcal{S}$$
$$\mathbf{w}^\top\mathbf{x} \geq 1 \quad \forall\mathbf{x} \in S_\mathcal{T} \qquad\qquad \mathbf{w}^\top\mathbf{x} \geq 1 \quad \forall\mathbf{x} \in S_\mathcal{T}$$

That is, while $\mathbf{w}_{\text{DD}}$ is the max-margin classifier on the entire dataset, $\mathbf{w}_{\text{color}}$ maximizes margin only on $S_\mathcal{T}$ while constraining $S_\mathcal{S}$ to lie on the correct side of the decision threshold. Our practical implementation of the learning rule ($h_{\text{color}}$) maximizes the margin on $S_\mathcal{S}$ while constraining $S_\mathcal{T}$ as it leads to a more stable optimization. In the overparameterized regime, the conditions we lay out below ensure that both optimization problems admit feasible solutions with probability 1. Now we present our main result for this section.

**Theorem 1.** *Consider a Linear-Gaussian PU-learning problem with parameters $\boldsymbol{\mu}, \boldsymbol{\eta}, d > 300$, dataset sizes $N_\mathcal{T} > 10$, $N = N_\mathcal{S} + N_\mathcal{T}$, $\alpha \in (0, \frac{N}{1024N_\mathcal{T}})$ and let $\delta \in (0,1)$. For all problems where*

$$\min\{r_\eta, r_\mu\} \geq \frac{16}{\sqrt{N_\mathcal{T}}}, \quad r_\mu \leq \frac{1}{2}\sqrt{N_{\mathcal{T},0}}, \quad \frac{r_\eta}{r_\mu} \leq \frac{4}{N}, \quad c_1\log\left(\frac{c_2}{\delta}\right) \leq \min\left(\sqrt{N}, \sqrt{\frac{d}{N}}\right),$$

*it holds with probability at least $1 - \delta$ that $\text{AU} - \text{ROC}(\mathbf{w}_{DD}) < 0.5$ and $\text{AU} - \text{ROC}(\mathbf{w}_{color}) > 0.9$.*

Detailed derivation of the theorem 1 claims are provided in appendix A.2. The result shows that a constrained learning rule can be arbitrarily more accurate than a domain discriminator.[6] This advantage holds even for overparameterized models that generalize well to domain discrimination. The parameter ranges in theorem 1 indicate when the constrained approach outperforms the baseline, specifically, when the novel-class fraction $\alpha$ is small and the shift magnitude $r_\mu$ is large enough but not too large to violate support overlap between $P_\mathcal{S}$ and $P_{\mathcal{T},0}$. The conditions impose lower bounds on the norms of the source and novel class samples to avoid cases with excessive noisy sampling. Furthermore, the upper bounds on these norms makes it harder to distinguish between known and novel instances. Intuitively, higher-norm features induce stronger separation margins, making them easier for a classifier to detect. Hence, at lower norms, $\mathbf{w}_{\text{color}}$ outperforms the baseline $\mathbf{w}_{\text{DD}}$. We also have a condition on $d$ to ensure overparameterized regime given a lower bound on $N$.

Next, we introduce CoLOR, a practical implementation of the rule in eq. (2), for OSDA under background shift. Experiments in section 6 confirm superior performance over baselines in regimes consistent with theory.

## 5 CoLOR: A solution to OSDA under Background Shift

The simplified constrained learning rule proposed in the previous section cannot be directly applied in practice to large-scale models or real world datasets that deviate from the ideal linear-Gaussian setup we considered. Hence, we first formulate the learning rule for (CoLOR) for OSDA under background shift for $k \geq 1$ with novel classes of proportion, $\alpha$, w.r.t. target dataset and a tolerance $\beta > 0$ for false detection of $S_\mathcal{S}$ as novelties:

$$h_{\text{color}} = \arg\min_{h \in \mathcal{H}}\left(\sum_{i \in S_\mathcal{S}} l_{ce}(h_c(\mathbf{x}_i), y_i) + \hat{\beta}_{\text{emp}}(h_{\text{novel}})\right) \text{ s.t. } \hat{\alpha}_{\text{emp}}(h_{\text{novel}}) < \hat{\alpha}. \tag{3}$$

---

[5]This is based on results that show the implicit bias of gradient descent and the logistic loss to the max-margin classifier on separable data (Soudry et al., 2018)

[6]We prove that the AU-ROC of CoLOR exceeds 0.9, but the proof can be extended to yield arbitrarily high AU-ROC.

where $h = [h_c, h_{\text{novel}}]$, $h_c$ are classification heads for known classes, $h_{\text{novel}}$ is the novelty detection head, $l_{ce}$ is the supervised cross entropy loss, $\hat{\alpha}_{\text{emp}}(h), \hat{\beta}_{\text{emp}}(h)$ are empirical estimates of $\alpha(h), \beta(h)$ over $S_{\mathcal{T}}$ and $S_{\mathcal{S}}$ respectively. $\hat{\alpha}$ is some constant corresponding to the target recall. It is a hyperparameter that needs tuning.

## 5.1 Efficient Architecture for Estimating the Novel Class Ratio ($\alpha$)

To solve the eq. (3), we follow Wald & Saria (2023); Chamon et al. (2022); Cotter et al. (2019) and solve a Lagrangian optimization problem obtained by switching the role of maximization and constraints in eq. (3) and taking its dual, see eq. (4) for the full objective. However, a naïve implementation of this objective has a significant drawback in terms of computational complexity due to hyperparameter optimization. Indeed, prior work on the rate-constrained learning problems we seek to solve (where rate in our problem corresponds to the size of the novel class) is limited to either very small models and datasets (Wald & Saria, 2023), or to certain applications such as fairness (Zafar et al., 2019), where the desired rate is known a-priori.

The computationally challenging part is that our Lagrangian problem needs to be solved for $L$ candidate values of $\hat{\alpha}_{\text{emp}} \in \boldsymbol{\alpha}$, corresponding to the potential estimates of novel class proportion, where $\boldsymbol{\alpha} \in [0,1]^L$. We perform grid search over $\boldsymbol{\alpha}$ and choose the model that obtains largest empirical recall estimate $(\hat{\alpha}_{\text{emp}}(h))$ w.r.t. the task of distinguishing between $P_{\mathcal{S}}$ and $P_{\mathcal{T}}$, while still satisfying the constraint on empirical FPR i.e. false positive rate estimate $(\hat{\beta})$ for the same task as specified in the learning objective. For empirical purposes, we calculate the approximate bound for $\beta$ (constraint on $\hat{\beta}_{\text{emp}}(h)$) using the Rademacher complexity in theorem 1 of Wald & Saria (2023). We find that setting $\beta = 0.01$ is well within the theoretically calculated bounds and works well in practice for all the experiments. Plots in figure 4 provide further insights about the impact of varying $\beta$ on OSDA performance of CoLOR.

To this end we train an architecture $h(\mathbf{x}) = w \circ \phi(\mathbf{x})$ where $\phi : \mathcal{X} \to \mathbb{R}^d$ is a shared representation for classification heads, $w^c : \mathbb{R}^d \to \mathbb{R}^k$, as well as several novelty heads, $w^\alpha = \{w_i^\alpha : \mathbb{R}^d \to \mathbb{R}\}_{i=1}^L$, and each $w_i^\alpha$ corresponds to a candidate value $\hat{\alpha}$ in the search grid $\boldsymbol{\alpha}$. Hence the solution amortizes training time for all candidate values by solving the primal dual optimization problem once. which leads to better performance with large data & models in practice as we see in section 6.

## 5.2 A simple Extension of Constrained Learning for OSDA

To account for the known classes $Y = 1, \ldots, k$ and $L$ novelty heads, we construct $w = [w^c, w^\alpha]$, such that $w : \mathbb{R}^d \to \mathbb{R}^{L+k}$. $w^\alpha$ is responsible for detecting novelties with various proportions while $w^c$ determines the known class of the input $\mathbf{x}$. See algorithm 1 for further details. Note that training multiple novelty heads $([w_1^\alpha, w_2^\alpha, ..., w_L^\alpha])$ simultaneously is more computationally efficient than training a separate model for each candidate $\hat{\alpha}$ iteratively. We hypothesize that having a shared representation for both tasks enables the model to learn a simpler representation that fits both tasks, yielding favorable generalization bounds as suggested by theory on multitask learning (Baxter, 2000; Maurer et al., 2016).

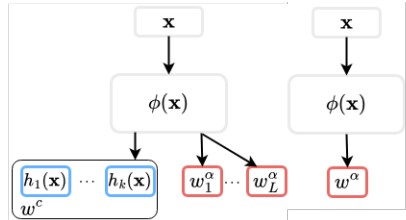

Figure 2: (left) CoLOR architecture for OSDA, heads $w_i^a$ for multiple recall values and classification heads $w^c$, vs. (right) a network optimizing for novelty detection with single recall value as in Wald & Saria (2023).

Let us gather the components for our overall method, CoLOR, that we evaluate in the next section. Figure 2 illustrates the architecture of CoLOR. It is trained by combining the constrained learning rule with $\ell_{\log}$, the binary cross-entropy loss, aiming to detect novel examples in $S_{\mathcal{T}}$,[7] $\lambda_{\hat{\alpha}}$ is a Lagrange multiplier corresponding to the head $h_{\hat{\alpha}} = \phi \circ w^\alpha(\mathbf{x})$, $\ell_\sigma(\cdot)$ is a sigmoid function which serves as an approximation to the indicator function, and the supervised cross entropy loss is $l_{ce}$ for $h_c(\mathbf{x}_i) = w^c \circ \phi(x)$.

$$\mathcal{L}(h) = N_{\mathcal{S}}^{-1} \sum_{i \in S_{\mathcal{S}}} \left( l_{ce}(h_c(\mathbf{x}_i), y_i) + \sum_{\hat{\alpha} \in \boldsymbol{\alpha}} l_{\log}(h_{\hat{\alpha}}(\mathbf{x}_i), 0) \right) + N_{\mathcal{T}}^{-1} \sum_{\substack{i \in S_{\mathcal{T}}, \\ \hat{\alpha} \in \boldsymbol{\alpha}}} \lambda_{\hat{\alpha}} \cdot \left( l_\sigma(h_{\hat{\alpha}}(\mathbf{x}_i)) - \hat{\alpha} \right) \quad (4)$$

---

[7]label 0 in the loss corresponds here to a sample belonging to $S_{\mathcal{S}}$.

In our experiments, the representation $\phi(\mathbf{x})$ is either trained from scratch or built on a pretrained backbone with added fully connected layers. For small datasets (e.g., CIFAR100), we train $\phi(x)$ as ResNet18 (He et al., 2016) from scratch. For larger ones (e.g., SUN397), we use pretrained $\phi(x)$ such as ResNet50 (ImageNet1K_V1) (Russakovsky et al., 2015) or a CLIP-pretrained ViT-L/14 encoder (Radford et al., 2021). For text datasets (e.g., Amazon Reviews), we use RoBERTa (Liu et al., 2019) for embeddings. See appendix A.7 for details.

---

**Algorithm 1** CoLOR: Constrained Learning for Open-set Recognition

---

**Require:** Labelled $S_{\mathcal{S}}$ and unlabelled $S_{\mathcal{T}}$ datasets, hypothesis class $\mathcal{H}$, target FPR $\beta > 0$ and $L$ potential novel class sizes $\boldsymbol{\alpha} \in [0,1]^L$.

1: Split $S_{\mathcal{S}}, S_{\mathcal{T}}$ into train $T_{\mathcal{S}}, T_{\mathcal{T}}$ and validation sets $V_{\mathcal{S}}, V_{\mathcal{T}}$ respectively.
2: Train either the entire model ($\phi$, $w^c$ & $w^\alpha$) or use pretrained $\phi$ and just train the last fully connected layer of $\phi$ along with $w^c$ & $w^\alpha_{\hat{\alpha}}$ to minimize the eq. 4.
3: Let $\hat{\beta}_{\text{emp}}(w^\alpha_{\hat{\alpha}}) = \frac{1}{|V_{\mathcal{S}}|} \sum_{\mathbf{x} \in V_{\mathcal{S}}} w^\alpha_{\hat{\alpha}} \circ \phi(\mathbf{x})$, and $\hat{\alpha}(w^\alpha_{\hat{\alpha}}) = \frac{1}{|V_{\mathcal{T}}|} \sum_{\mathbf{x} \in V_{\mathcal{T}}} w^\alpha_{\hat{\alpha}} \circ \phi(\mathbf{x}) \; \forall \alpha \in \boldsymbol{\alpha}$
4: **return** $\left[ w^c, \operatorname{argmax}_{w^\alpha_{\hat{\alpha}} : \alpha \in \boldsymbol{\alpha}, \hat{\beta}_{\text{emp}}(w^\alpha_{\hat{\alpha}}) < \beta} \hat{\alpha}(w^\alpha_{\hat{\alpha}}) \right]$

---

## 6  Experiments

We are now in place to empirically evaluate CoLOR against variety of baselines across background shifts and novel classes that we create in real data. Here, the main questions we wish to answer are:

1. Does background shift within non-novel instances affect OSDA performance?

2. Can shared representations mitigate the impact of background shift by enhancing robustness in classifying samples from $S_{\mathcal{T},[k]}$ thereby improving overall OSDA performance?

3. What is the effect of novel class ratio $\alpha$ particularly w.r.t. detecting novel classes?

### 6.1  Experimental setting

The experiments to examine these questions are devised over image and text datasets as follows. We randomly draw a class from the set of classes $\mathcal{Y}$ and assign that as the $k+1$-th novel class. Denoting the instances of known classes by $S_k = \{\mathbf{x} : y \in [k]\} \, \forall (\mathbf{x}, y) \in S$ and novel ones as $S_{k+1} = \{\mathbf{x} : y = k+1\} \, \forall (\mathbf{x}, y) \in S$, we create a background shift by further splitting $S_k$ into $S_{\mathcal{S}}$ and $S_{\mathcal{T},[k]}$. We use semantic attributes that are annotated in the metadata of each dataset to create this shift between $S_{\mathcal{S}}$ and $S_{\mathcal{T},[k]}$. The attributes used for each dataset are specified in section 6.2 and elaborated further in section A.7.1. At training time the learner is provided with a labelled dataset $S_{\mathcal{S}}$ and the unlabelled dataset $S_{\mathcal{T}} = S_{\mathcal{T},[k]} \cup S_{k+1}$. The mixture proportion $\alpha = |S_{k+1}|/(|S_{k+1}| + |S_{\mathcal{T},[k]}|)$ is set by adjusting the sizes of the selected novel classes in $S_{\mathcal{T},[k]}$.

### 6.2  Datasets

Most of the large scale image classification models are trained on ImageNet dataset. Hence we use a similar large scale dataset, **SUN397** (Xiao et al., 2010) having completely different categories than ImageNet. We exploit three-level hierarchy structure to create distribution shifts by varying proportions of latent subtypes ($y$) of known class labels $\mathcal{Y}$. This causes a background shift where $\text{Supp}(P_{\mathcal{T},[k]}) \subseteq \text{Supp}(P_{\mathcal{S}})$. Indoor scenes (e.g., shopping/dining places, workplaces) serve as known (in-distribution) classes, while outdoor scenes are randomly selected as novel classes. We also include a **Amazon Product Reviews** text dataset (Ni et al., 2019) to demonstrate the versatility of the method across diverse modalities. Classes are different product categories (prime pantry, musical instruments, etc.), and induce background/covariate shift in known classes based on positive vs. negative sentiment in the review. The novel class is an unknown product category. Finally, we include small scale **CIFAR100** (Krizhevsky, 2009) dataset to evaluate the adaptive methods (like DD, nnPU, uPU, BODA, OSDA) particularly in scenarios when their feature extractors are trained from scratch. Similar to the SUN397 dataset, we leverage the inherent hierarchies of CIFAR100 classes to create a natural background shift by varying latent subtype proportions and include novel classes in the target dataset. Table 1 provides a summary of characteristics and further details on each dataset are in appendix A.7.1.

**Evaluation metrics:** We primarily use Area Under ROC Curve (AUROC) and Area Under Precision-Recall Curve (AUPRC) to evaluate the novel category detection performance, while we use Open-Set Classification

Table 1: Overview of experiment settings. DS = distribution shift, prop. = proportions

| Experiment setup | SUN397 | CIFAR100 | Amazon Reviews |
|---|---|---|---|
| DS factor | varying subtypes prop. | varying subtypes prop. | sentiment |
| no. of novel classes | 12 | 5 | 6 |
| Novel class ratio ($\alpha$) | $0.07 \pm 0.03$ | $0.16 \pm 0.09$ | $0.07 \pm 0.02$ |

Rate (OSCR) to summarize overall OSDA performance for all the methods [8]. OSCR (Dhamija et al., 2018) measures the trade-off between correct classification rate of the known classes and false positive rate of the novel samples.

### 6.3 Baseline methods

We include adaptive methods from novelty detection that access both labelled source and unlabelled target data. These baselines include domain discriminator DD, (Elkan & Noto, 2008; du Plessis et al., 2014), uPU (du Plessis et al., 2014) and nnPU (Kiryo et al., 2017). These methods are modified for OSDA through joint learning approach enabled by a simple architectural modification like figure 2 and aggregating the loss components from both the tasks. We also include another popular OSDA baselines BODA (Saito et al., 2018) and PULSE (Garg et al., 2022) that are agnostic to input data modality. Another adaptive baseline is Li et al. (2023), however, this is specifically designed for vision data. Hence, we report its performance on SUN397 dataset in table 3 along with other vision-specific baselines.

We include an entropy-based method, ARPL (Chen et al., 2021) with Maximum Logit Score as proposed in (Vaze et al., 2022). Such methods are non-adaptive as they do not access target data, yet are a popular choice for novelty detection.[9] We also tested simple and popular baselines such as MSP Hendrycks & Gimpel (2016) and results can be found in appendix tables 17, 18. We further include SHOT (Liang et al., 2021), CAC (Miller et al., 2021) and a zero-shot OOD detection method (ZOC) proposed in Esmaeilpour et al. (2022). Further details about training and hyperparameters are in Appendix A.7.2.

### 6.4 Results

Table 2 summarizes the OSDA performance across all the datasets using all the metrics discussed previously. We observe that CoLOR generally outperforms all the baselines with a significant margin. The OSCR performance of CoLOR is comparable to BODA, ARPL and PULSE for CIFAR100 and Amazon Reviews dataset however, they notably under perform in novelty detection (AUROC and AUPRC) for the same datasets. Additionally, we observe that these methods do not perform as well on the larger dataset of SUN397. We find that the nnPU objective defaults to uPU when using pretrained feature extractors (SUN397 and Amazon Reviews), as the empirical risk remains non-negative for both. This causes similar results of uPU and nnPU on SUN397 and CIFAR100 datasets. In the image datasets, we find that adaptive methods significantly outperform non-adaptive methods overall. We further strengthen our results by extending the study to other richer feature representations like pretrained CLIP ViT-L/14, particularly for SUN397 dataset. This is shown in 3a across different novel classes (X-axis) of SUN397 dataset. CoLOR still outperforms the baselines using either of the backbone architectures. We can see that CLIP ViT-L/14 have overall better performance than ResNet50 (pretrained on ImageNet) in SUN397 dataset due to richer and more robust features. This mitigates the impact of distribution shift, which is more pronounced in SUN397 compared to other datasets. Further results are in appendix tables 8, 10, 12, 14.

### 6.5 Discussion

Based on our observations from the experiments, we address the three critical questions below:
**Background shift within non-novel instances causes a significant drop in the OSDA performance of all the methods that involve fine-tuning using the source data.**
From tables 3, 10 and 11 we observe that all the methods that involve training a closed-set classifier suffer from background shift. Hence, background shift not only harms closed-set performance but also that of

---

[8]Section A.5.1 in appendix provides detailed argument for the preference of AUPRC over AUROC mainly when minority class proportions are very low.

[9]We are using ARPL on SUN397. The paper that proposed ARPL also introduced an improvement with Confused Sampling (ARPL+CS) that uses GANs, but that proved challenging to apply in the SUN397 dataset

[10]The summary statistics are derived by averaging (or aggregating wins) across all novel class identities and randomly generated data splits (between $P_{\mathcal{S}}$ & $P_{\mathcal{T}}$) along with the corresponding standard deviations. Refer A.7.2

Table 2: Performance comparison of adaptive methods for OSDA under background shift demonstrating versatility across data modalities. Detailed results in appendix Tables 8, 12, and 14.[10]

| Metric | Methods | SUN397 ($\alpha = 0.07 \pm 0.03$) | | CIFAR100 ($\alpha = 0.07 \pm 0.02$) | | Amazon Reviews ($\alpha = 0.16 \pm 0.09$) | |
| | | ResNet50 pretrained on ImageNet1K | | ResNet18 (randomly initialized) | | pretrained RoBERTa | |
| | | Summary | Wins | Summary | Wins | Summary | Wins |
|---|---|---|---|---|---|---|---|
| AUROC | DD | $0.91 \pm 0.05$ | 1/15 | $0.70 \pm 0.11$ | 3/25 | $0.72 \pm 0.09$ | 1/30 |
| | uPU | $0.76 \pm 0.14$ | 0/15 | $0.67 \pm 0.10$ | 1/25 | $0.76 \pm 0.08$ | 4/30 |
| | nnPU | $0.76 \pm 0.14$ | 0/15 | $0.67 \pm 0.12$ | 1/25 | $0.76 \pm 0.08$ | 4/30* |
| | BODA | $0.86 \pm 0.06$ | 0/15 | $0.57 \pm 0.06$ | 0/25 | $0.66 \pm 0.10$ | 2/30 |
| | ARPL | $0.71 \pm 0.08$ | 0/15 | $0.76 \pm 0.05$ | 8/25 | $0.70 \pm 0.05$ | 1/30 |
| | CAC | $0.80 \pm 0.05$ | 0/15 | $0.69 \pm 0.04$ | 2/25 | $0.70 \pm 0.07$ | 5/30 |
| | PULSE | $0.73 \pm 0.07$ | 0/15 | $0.72 \pm 0.04$ | 0/25 | $0.63 \pm 0.08$ | 1/30 |
| | CoLOR | $\mathbf{0.98 \pm 0.02}$ | $\mathbf{14/15}$ | $\mathbf{0.77 \pm 0.09}$ | $\mathbf{10/25}$ | $\mathbf{0.79 \pm 0.09}$ | $\mathbf{16/30}$ |
| AUPRC | DD | $0.54 \pm 0.22$ | 1/15 | $0.24 \pm 0.13$ | 2/25 | $0.43 \pm 0.18$ | 0/30 |
| | uPU | $0.21 \pm 0.22$ | 0/15 | $0.18 \pm 0.12$ | 1/25 | $0.51 \pm 0.17$ | 7/30 |
| | nnPU | $0.21 \pm 0.22$ | 0/15 | $0.20 \pm 0.16$ | 2/25 | $0.51 \pm 0.17$ | 7/30* |
| | BODA | $0.45 \pm 0.11$ | 0/15 | $0.09 \pm 0.03$ | 0/25 | $0.28 \pm 0.21$ | 1/30 |
| | ARPL | $0.12 \pm 0.07$ | 0/15 | $0.18 \pm 0.07$ | 4/25 | $0.27 \pm 0.13$ | 0/30 |
| | CAC | $0.18 \pm 0.07$ | 0/15 | $0.15 \pm 0.05$ | 1/25 | $0.3 \pm 0.16$ | 1/30 |
| | PULSE | $0.15 \pm 0.05$ | 0/15 | $0.17 \pm 0.05$ | 0/25 | $0.21 \pm 0.13$ | 0/30 |
| | CoLOR | $\mathbf{0.91 \pm 0.09}$ | $\mathbf{14/15}$ | $\mathbf{0.33 \pm 0.14}$ | $\mathbf{15/25}$ | $\mathbf{0.54 \pm 0.18}$ | $\mathbf{21/30}$ |
| OSCR | DD | $0.68 \pm 0.05$ | 1/15 | $0.51 \pm 0.10$ | 1/25 | $0.50 \pm 0.06$ | 0/30 |
| | uPU | $0.40 \pm 0.11$ | 0/15 | $0.49 \pm 0.09$ | 0/25 | $0.54 \pm 0.05$ | 5/30 |
| | nnPU | $0.40 \pm 0.11$ | 0/15 | $0.48 \pm 0.10$ | 1/25 | $0.54 \pm 0.05$ | 5/30* |
| | BODA | $0.55 \pm 0.10$ | 0/15 | $0.60 \pm 0.04$ | 3/25 | $\mathbf{0.56 \pm 0.06}$ | 7/30 |
| | ARPL | $0.60 \pm 0.06$ | 0/15 | $0.61 \pm 0.04$ | $\mathbf{7/25}$ | $0.56 \pm 0.05$ | 4/30 |
| | CAC | $0.68 \pm 0.05$ | 0/15 | $0.55 \pm 0.04$ | 0/25 | $0.49 \pm 0.07$ | 0/30 |
| | PULSE | $0.65 \pm 0.06$ | 0/15 | $\mathbf{0.62 \pm 0.03}$ | 6/25 | $0.53 \pm 0.07$ | 4/30 |
| | CoLOR | $\mathbf{0.81 \pm 0.04}$ | $\mathbf{14/15}$ | $0.59 \pm 0.08$ | $\mathbf{7/25}$ | $\mathbf{0.56 \pm 0.06}$ | $\mathbf{10/30}$ |

*When nnPU behaves exactly like uPU, both are awarded the win.*

Table 3: OSDA performance with and without background shift (DS) using ResNet50 backbone for SUN397. These results include additional vision-specific baselines like SHOT, ZOC, ANNA

| Method | AUROC | | AUPRC | | OSCR | |
| | w/ BS | w/o BS | w/ BS | w/o BS | w/ BS | w/o BS |
|---|---|---|---|---|---|---|
| DD | $0.91 \pm 0.05$ | $\mathbf{1.00 \pm 0.00}$ | $0.54 \pm 0.22$ | $\mathbf{1.00 \pm 0.00}$ | $0.68 \pm 0.05$ | $\mathbf{0.99 \pm 0.00}$ |
| uPU | $0.76 \pm 0.14$ | $\mathbf{1.00 \pm 0.00}$ | $0.21 \pm 0.22$ | $\mathbf{1.00 \pm 0.00}$ | $0.40 \pm 0.11$ | $\mathbf{0.99 \pm 0.00}$ |
| BODA | $0.86 \pm 0.06$ | $0.88 \pm 0.04$ | $0.45 \pm 0.11$ | $0.24 \pm 0.15$ | $0.55 \pm 0.10$ | $0.93 \pm 0.02$ |
| ARPL | $0.71 \pm 0.08$ | $0.84 \pm 0.04$ | $0.12 \pm 0.07$ | $0.20 \pm 0.07$ | $0.60 \pm 0.06$ | $0.80 \pm 0.04$ |
| CAC | $0.80 \pm 0.05$ | $0.88 \pm 0.03$ | $0.18 \pm 0.07$ | $0.29 \pm 0.07$ | $0.68 \pm 0.05$ | $0.84 \pm 0.03$ |
| PULSE | $0.73 \pm 0.07$ | $0.82 \pm 0.05$ | $0.15 \pm 0.05$ | $0.20 \pm 0.05$ | $0.65 \pm 0.06$ | $0.78 \pm 0.05$ |
| CoLOR | $\mathbf{0.98 \pm 0.02}$ | $\mathbf{1.00 \pm 0.00}$ | $\mathbf{0.91 \pm 0.09}$ | $0.99 \pm 0.03$ | $\mathbf{0.81 \pm 0.04}$ | $0.93 \pm 0.01$ |
| SHOT | $0.71 \pm 0.16$ | $0.84 \pm 0.04$ | $0.19 \pm 0.13$ | $0.20 \pm 0.07$ | $0.22 \pm 0.07$ | $0.80 \pm 0.04$ |
| ZOC* | $0.82 \pm 0.07$ | $0.82 \pm 0.08$ | $0.23 \pm 0.06$ | $0.22 \pm 0.07$ | $0.48 \pm 0.12$ | $0.49 \pm 0.05$ |
| ANNA | $0.93 \pm 0.05$ | $0.95 \pm 0.07$ | $0.73 \pm 0.16$ | $0.90 \pm 0.08$ | $0.60 \pm 0.07$ | $0.82 \pm 0.07$ |

*ZOC uses CLIP ViT-B/32 image encoder with BERT text encoder.*

Table 4: Top-1 accuracy over $S_{\mathcal{T}}$ for closed-set classification on SUN397 data.

| Method | ResNet50 | ViT-L/14 from CLIP |
|---|---|---|
| Source-only | $0.72 \pm 0.03$ | $0.97 \pm 0.01$ |
| DD | $0.75 \pm 0.04$ | $0.97 \pm 0.01$ |
| BODA | $0.70 \pm 0.07$ | $0.75 \pm 0.13$ |
| CAC | $\mathbf{0.85 \pm 0.02}$ | $\mathbf{0.98 \pm 0.01}$ |
| PULSE | $0.82 \pm 0.03$ | $0.97 \pm 0.01$ |
| CoLOR | $0.83 \pm 0.03$ | $\mathbf{0.98 \pm 0.01}$ |

open-set recognition. If we had a better closed set classifier, in the sense that it was more robust to the background shift, then we probably would've ended up with better OSDA performance too (see appendix A.6). CoLOR seems to primarily help when standard techniques like long training, label smoothing, and other techniques used in Vaze et al. (2022) are insufficient to train a model that's robust to shift. ZOC performs zero-shot open-set classification using pretrained CLIP and text decoder models without accessing source and

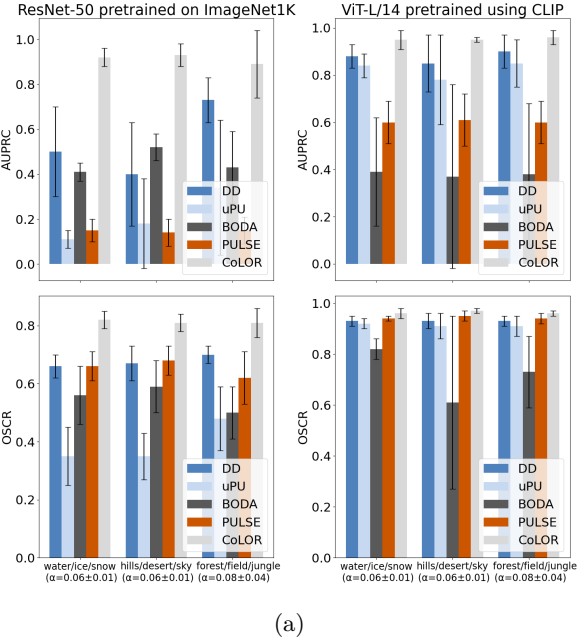
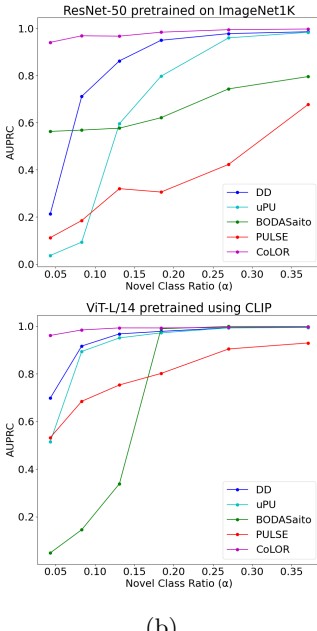

(a)                                                                    (b)

Figure 3: (a) OSDA performance of top performing adaptive methods on SUN397 dataset with background shift using pretrained ResNet50 & CLIP ViT-L/14 backbone architectures. (b) Impact of novel class ratio ($\alpha$) on adaptive methods on SUN397 dataset under background shift.

target data. Hence, it is not impacted by background shift as expected. We have provided further results on CIFAR100 dataset in 12 & 13 although CIFAR100 is a much simpler dataset and hence we see near perfect scores by ZOC due to the use of large CLIP ViT-B/32 model pretrained on huge image caption datasets. We see in tables 10 & 11 that non-adaptive methods do not perform well when scaled to larger SUN397 dataset. **Shared representations obtained by joint learning of closed set classification and novel category detection can help mitigate the impact of background shift within non-novel instances.**

Table 4 compares the Top-1 accuracy for known classes under background shift between $S_{\mathcal{S}}$ and $S_{\mathcal{T}}$. The Source-only method is trained on labelled $S_{\mathcal{S}}$ for classifying known classes in $S_{\mathcal{T}}$, while DD combines closed-set classification with domain discrimination (distinguishing samples from $S_{\mathcal{S}}$ and $S_{\mathcal{T}}$). We can see that DD has better Top-1 accuracy of known classes than the Source-only method on $S_{\mathcal{T}}$ notably for ResNet50 model. CoLOR further improves the performance by employing constrained learning instead of domain discrimination in the joint learning objective to detect novel classes. Performance differences remain incremental for CLIP ViT-L/14 because its pretrained representations are already rich and robust.

**As the novel class ratio $\alpha$ decreases, the performance of existing methods to detect unknown/novel classes significantly decreases.**

Figure 3b illustrates the effect of novel class ratio on the novel class detection performance of existing OSDA methods and CoLOR. A low $\alpha$ significantly deteriorates the open-set classification performance, especially for methods not designed to handle such cases. Existing benchmarks focus on large novel class sizes. We believe that experimenting with smaller sizes would be a valuable step toward creating more realistic benchmarks.

## 7   Future Work

Future research could focus on refining the search criteria for the optimal model head by selecting an appropriate $\beta$ threshold. Additionally, a robust theoretical underpinning is needed to explain why shared representations improve the classification of known classes under distribution shifts. Integrating test-time or model-free domain adaptation methods, such as those by Wang et al. (2020), Zhang et al. (2023), or Saito et al. (2018), could further enhance OSDA performance. It would be valuable to compare our approach with large foundation models. Although curating such a benchmark would require access to publicly available pretraining datasets like LAION, which would require significant effort and present unforeseen challenges.

## Acknowledgments

We acknowledge the guidance of Dr. Rama Chellappa in this project. Furthermore, we acknowledge the support from the DARPA TIAMAT program under Grant No. HR00112490422. The views and conclusions contained in this document are those of the authors and should not be interpreted as representing the official policies, either expressed or implied, of the U.S. Government.

This work was also supported by the Gordon and Betty Moore Foundation through the grant titled "Safety Monitoring of Deployed Clinical AI via the Framework for Real-time Auditing of Individual Predictions (FRAP)" (GBMF ID #12128).

We additionally acknowledge support from the NSF award titled "FW-HTF: Human-Machine Teaming for Medical Decision Making" (FAIN 1840088), which concluded in September 2024.

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

# A Appendix

## A.1 Notations and Keywords

Table 5: Notations for CoLOR learning rules and theoretical analysis.

| Notation | Meaning |
|---|---|
| $k, [k]$ | Number of known (source) classes, known class labels $[1, ...k]$ |
| $k+1$ | Novel / unknown class label |
| $x \in \mathcal{X}$ | Input instance |
| $y \in \mathcal{Y}$ | Class label |
| $S_{\mathcal{S}}$ | Labeled source dataset $\{(x_i, y_i)\}_{i=1}^{N_S}$ |
| $S_{\mathcal{T}}$ | Unlabeled target dataset $\{x_i\}_{i=1}^{N_T}$ |
| $N_S$ | Number of source samples |
| $N_T$ | Number of target samples |
| $N_{T,0}, N_{T,1}$ | Target known vs. novel samples (binary case) |
| $\mathcal{X}, \mathcal{Y}$ | Input space and label space, $\mathcal{Y} = [k+1]$ |
| $P_{\mathcal{S}} = P_{\mathcal{S}}(x,y)$ | Source joint distribution over $(x,y)$ |
| $P_{\mathcal{T}} = P_{\mathcal{T}}(x,y)$ | Target joint distribution over $(x,y)$ |
| $P_{T,[k]}(x,y)$ | Target distribution restricted to known classes |
| $P_{T,k+1}(x)$ | Target distribution of the novel class |
| $\alpha := P_T(y = k+1)$ | Novel class proportion in target |
| $\mathcal{H}$ | Hypothesis class of open-set clasification models |
| $h : \mathcal{X} \to \mathbb{R}^{k+1}$ | Model scoring all $k+1$ classes |
| $h_{\text{novel}}$ | Novelty detection score derived from $h$ |
| $h_c$ | Classification head for known classes |
| $h_{DD}$ | ERM classifier distinguishing $P_S$ vs. $P_T$ |
| $h_{CoLOR}$ | Constrained learning solution (eq. (3)) |
| $w_{DD}$ | Max-margin domain discriminator solution (appendix A.2) |
| $w_{CoLOR}$ | Max-margin constrained separator (appendix A.2) |
| $\ell_{01}$ | Binary 0-1 loss |
| $\ell_{\log}$ | Logistic / binary cross-entropy loss |
| $\ell_{\text{ce}}$ | Supervised cross-entropy loss |
| $t \in \mathbb{R}$ | Decision threshold for novelty |
| $\beta(h) = \beta(h;t) = \mathbb{E}_{\mathbf{x} \sim S_{\mathcal{S}}} [\mathbf{1}[h_{\text{novel}}(\mathbf{x};t) > t]]$ | False Positive Rate (FPR) on source |
| $\alpha(h) = \alpha(h;t) = \mathbb{E}_{\mathbf{x} \sim S_{\mathcal{T}}} [\mathbf{1}[h_{\text{novel}}(\mathbf{x};t) > t]]$ | Recall of novelty detection on target |
| $\hat{\alpha}$ | Target recall hyperparameter |
| $\hat{\beta}$ | Empirical FPR estimate |
| $\hat{\alpha}_{emp}(h)$ | Empirical estimate of target recall $\alpha(h)$ over $S_{\mathcal{T}}$ |
| $\hat{\beta}_{emp}(h)$ | Empirical estimate of source FPR $\beta(h)$ over $S_{\mathcal{S}}$ |
| $\lambda_{\hat{\alpha}}$ | Lagrange multiplier for constraint |
| $\boldsymbol{\alpha} \in [0,1]^L$ | Grid of candidate novel proportions |
| $L$ | Number of novelty heads in CoLOR |
| $\phi(x)$ | Backbone embedding model shared by $h_c$ and $h_{novel}$ heads |
| $w^c$ | Linear head for known-class classification |
| $w_i^{\alpha}$ | Novelty head for candidate $\hat{\alpha}_i$ |
| $d$ | Feature dimension |
| $\boldsymbol{\mu} = \mu$ | Background-shift direction |
| $\boldsymbol{\eta} = \eta$ | Novel-class direction |
| $r_{\mu} = \|\mu\|$ | Magnitude of background shift |
| $r_{\eta} = \|\eta\|$ | Novel-class signal strength |
| $\sigma = 1/\sqrt{d}$ | Noise scale |
| $\mathcal{N}(\cdot, \cdot)$ | Gaussian distribution |
| $\delta \in (0,1)$ | Failure probability in theorem 1 |

Table 6: Keywords for OSDA problem setup, distributions, and evaluation metrics.

| Keyword | Meaning |
| --- | --- |
| Supp($\cdot$) | Support of a distribution |
| Background shift | Shift where $\mathrm{Supp}(P_{T,[k]}) \subseteq \mathrm{Supp}(P_S)$ |
| Assumption 1 | Novel class separable from known classes |
| Linear-Gaussian theory (definition 2) | Simplified overparameterized PU model |
| *AUROC* | Area Under ROC Curve |
| *AUPRC* | Area Under Precision-Recall Curve |
| *OSCR* | Open-Set Classification Rate |
| Theorem 1 | Conditions in linear gaussian problem setup when $AUROC(w_{DD}) < 0.5$ but $AUROC(w_{CoLOR}) > 0.9$ |

## A.2 Theoretical analysis

Restating the statistical learning rule.

$$\mathbf{w}_{\mathrm{DD}} = \arg\min_{\mathbf{w}} \|\mathbf{w}\| \qquad\qquad \mathbf{w}_{\mathrm{color}} = \arg\min_{\mathbf{w}} \|\mathbf{w}\|$$

$$\text{s.t. } \mathbf{w}^\top \mathbf{x} \leq -1 \quad \forall \mathbf{x} \in S_{\mathcal{S}} \qquad \text{s.t. } \mathbf{w}^\top \mathbf{x} \leq 0 \quad \forall \mathbf{x} \in S_{\mathcal{S}}$$

$$\mathbf{w}^\top \mathbf{x} \geq 1 \quad \forall \mathbf{x} \in S_{\mathcal{T}} \qquad\qquad \mathbf{w}^\top \mathbf{x} \geq 1 \quad \forall \mathbf{x} \in S_{\mathcal{T}}$$

*Proof sketch.* The proof consists of four parts. First, we show that $\mathrm{AU-ROC}(\mathbf{w}) = Q(\frac{\mathbf{w}^\top(\boldsymbol{\mu}-\boldsymbol{\eta})}{\sigma\sqrt{\|\mathbf{w}\|^2 - (\mathbf{w}^\top(\boldsymbol{\mu}+\boldsymbol{\eta}))^2}})$, where $Q(\cdot)$ is the inverse Gaussian tail function. This means that whenever $\mathbf{w}_{\mathrm{DD}}^\top(\boldsymbol{\mu}-\boldsymbol{\eta}) \leq 0$, the domain discriminator algorithm will have $\mathrm{AU-ROC}(\mathbf{w}_{\mathrm{DD}}) \leq 0.5$. To derive ranges of problem parameters where this happens, we analyze the convex dual of the maximum-margin problem with an added constraint that $\mathbf{w}_{\mathrm{DD}}^\top(\boldsymbol{\mu}-\boldsymbol{\eta}) > 0$ and obtain a lower bound $\underline{\gamma}$ on the value of the primal problem (i.e. a lower bound on the norm of a "good" solution). Then we guess a solution of the form $\mathbf{w}' = \boldsymbol{\mu} + \sum_{i \in U} \xi_i$, which does not depend on $\boldsymbol{\eta}$ at all and has AU-ROC smaller than 0.5. Intuitively, $U$ is constructed by including the novel class examples and use the noise vectors of these examples to fit them and satisfy the constraints of the problem. Then, by showing that $\|\mathbf{w}'\| < \underline{\gamma}$ we prove that $\mathrm{AU-ROC}(\mathbf{w}_{\mathrm{DD}}) \leq 0.5$. This is summarized in proposition 1. Next in proposition 2, we follow a similar procedure with $\mathbf{w}_{\mathrm{color}}$ of analyzing a problem with an added constraint $\mathbf{w}^\top(\boldsymbol{\mu}-\boldsymbol{\eta}) < \tau$ for some $\tau > 0$. Again using weak duality and guessing a solution, we prove that $\mathbf{w}_{\mathrm{color}}$ will satisfy the constraint, this time showing that it obtains a high AU-ROC. Finally, by finding an intersection of parameters ranges where both bounds on the AU-ROC hold, we prove the main claim in theorem 2 (restatement of theorem 1). $\qquad\square$

**Lemma 2.** *For a Gaussian two-feature PU problem, let $\hat{\boldsymbol{\eta}} = \boldsymbol{\eta}/\|\boldsymbol{\eta}\|$ and $\hat{\boldsymbol{\mu}}$ accordingly. We have that the AU-ROC of a model $\mathbf{w} \in \mathbb{R}^d$ is $Q(\frac{\langle \mathbf{w}, \boldsymbol{\eta}-\boldsymbol{\mu}\rangle}{\sigma\sqrt{(2\|\mathbf{w}\|^2 - \langle\mathbf{w},\hat{\boldsymbol{\eta}}\rangle^2 - \langle\mathbf{w},\hat{\boldsymbol{\mu}}\rangle^2)}})$.*

*Proof.* Let $\mathbf{x}_{\mathrm{novel}} \sim P_{\mathcal{T},1}, \mathbf{x}_{\mathcal{T},0} \sim P_{\mathcal{T},0}$, we have

$$\mathrm{AU-ROC}(\mathbf{w}) = P(\langle\mathbf{w}, \mathbf{x}_{\mathrm{novel}}\rangle < \langle\mathbf{w}, \mathbf{x}_{\mathcal{T},0}\rangle)$$
$$= P(\langle\mathbf{w}, -\boldsymbol{\eta} + \xi_{\mathrm{novel}}\rangle < \langle\mathbf{w}, -\boldsymbol{\mu} + \xi_{\mathcal{T},0}\rangle)$$
$$= P(\langle\mathbf{w}, -\boldsymbol{\eta} + \boldsymbol{\mu}\rangle < \langle\mathbf{w}, -\xi_{\mathrm{novel}} + \xi_{\mathcal{T},0}\rangle),$$

where $\xi_{\mathrm{novel}} \sim \mathcal{N}(0, \sigma^2(I_d - \hat{\boldsymbol{\mu}}\hat{\boldsymbol{\mu}}^\top))$ and $\xi_{\mathcal{T},0} \sim \mathcal{N}(0, \sigma^2(I_d - \hat{\boldsymbol{\eta}}\hat{\boldsymbol{\eta}}^\top))$. Then we have that $\langle\mathbf{w}, \xi_{\mathrm{novel}} - \xi_{\mathcal{T},0}\rangle \sim \mathcal{N}\left(0, \sigma^2\left(2\|\mathbf{w}\|^2 - \langle\mathbf{w},\hat{\boldsymbol{\eta}}\rangle^2 - \langle\mathbf{w},\hat{\boldsymbol{\mu}}\rangle^2\right)\right)$. Dividing both sides of the argument above by a factor $\sigma\sqrt{2\|w\|^2 - \langle\mathbf{w},\hat{\boldsymbol{\eta}}\rangle^2 - \langle\mathbf{w},\hat{\boldsymbol{\mu}}\rangle^2}$ we get that for a variable $\xi \sim \mathcal{N}(0,1)$

$$\mathrm{AU-ROC}(\mathbf{w}) = P\left(\frac{\langle\mathbf{w}, \boldsymbol{\mu} - \boldsymbol{\eta}\rangle}{\sigma\sqrt{2\|\mathbf{w}\|^2 - \langle\mathbf{w},\hat{\boldsymbol{\eta}}\rangle^2 - \langle w,\hat{\boldsymbol{\mu}}\rangle^2}} < \xi\right)$$

$$= Q\left(\frac{\langle \mathbf{w}, \boldsymbol{\mu} - \boldsymbol{\eta}\rangle}{\sigma\sqrt{2\|\mathbf{w}\|^2 - \langle \mathbf{w}, \hat{\boldsymbol{\eta}}\rangle^2 - \langle \mathbf{w}, \hat{\boldsymbol{\mu}}\rangle^2}}\right)$$

Note that when $\langle \mathbf{w}, \boldsymbol{\mu} - \boldsymbol{\eta}\rangle > 0$ then the AU-ROC is lower than 0.5, and when $\langle \mathbf{w}, \boldsymbol{\mu} - \boldsymbol{\eta}\rangle < 0$ then $Q\left(\frac{\langle \mathbf{w}, \boldsymbol{\mu} - \boldsymbol{\eta}\rangle}{\sigma\sqrt{2\|\mathbf{w}\|^2 - \langle \mathbf{w}, \hat{\boldsymbol{\eta}}\rangle^2 - \langle \mathbf{w}, \hat{\boldsymbol{\mu}}\rangle^2}}\right) \geq Q\left(\frac{\langle \mathbf{w}, \boldsymbol{\mu} - \boldsymbol{\eta}\rangle}{\sqrt{2}\sigma\|\mathbf{w}\|}\right)$. Hence the RHS is a lower bound on the AU-ROC.

$\square$

**Proposition 1.** *For $\sigma = 1/\sqrt{d}$, let $\delta \in (0, 1)$ be a failure probability, and $C_d, c_d$ constants, and consider the set of problems where*

$$C_d \log(\frac{c_d}{\delta}) \leq \frac{\sqrt{d}}{4\sqrt{N_{\mathcal{S}} + N_{\mathcal{T}}}},$$

$$r_\mu \geq \frac{1}{2\sqrt{N_{\mathcal{S}} + N_{\mathcal{T}}}},$$

$$r_\eta \leq \frac{r_\mu}{4 + 8r_\mu\sqrt{N_{\mathcal{T},1}}} - \frac{1}{\sqrt{N_{\mathcal{S}} + N_{\mathcal{T}}}}$$

*There exists $C_d, c_d$ such that for any $\delta$, if we fix $\boldsymbol{\mu}, \boldsymbol{\eta}, r_\mu, r_\eta, N_{\mathcal{S}}, N_{\mathcal{T},0}, N_{\mathcal{T},1}$ that are in the above set, and drawing training data $(S_{\mathcal{S}}, S_{\mathcal{T}})$ as described in problem definition. Let $\mathbf{w}_{DD}$ as defined in appendix A.2, then with probability at least $1 - \delta$ we have that $AU - ROC(\mathbf{w}_{DD}) \leq 0.5$.*

*Proof.* **Lower bound on norm of good solutions under DD:**
Lets take a Lagrangian

$$\mathcal{L}(\mathbf{w}, \lambda, \nu) = \|\mathbf{w}\|^2 + \lambda_{\mathcal{S}}^\top(\mathbf{1}_{\mathcal{S}} - \boldsymbol{X}_{\mathcal{S}}\mathbf{w}) + \lambda_{\mathcal{T}}^\top(\mathbf{1}_{\mathcal{T}} + \boldsymbol{X}_{\mathcal{T}}\mathbf{w}) + \nu \cdot (\mathbf{w}^\top(\boldsymbol{\mu} - \boldsymbol{\eta}))$$

and zero the gradient

$$\nabla_{\mathbf{w}}\mathcal{L}(\mathbf{w}, \lambda, \nu) = 2\mathbf{w} - \boldsymbol{X}_{\mathcal{S}}^\top\lambda_{\mathcal{S}} + \boldsymbol{X}_{\mathcal{T}}^\top\lambda_{\mathcal{T}} + \nu \cdot (\boldsymbol{\mu} - \boldsymbol{\eta}) = 0$$

$$\Leftrightarrow \mathbf{w} = \frac{1}{2}\left(\nu \cdot (\boldsymbol{\eta} - \boldsymbol{\mu}) + \boldsymbol{X}_{\mathcal{S}}^\top\lambda_{\mathcal{S}} - \boldsymbol{X}_{\mathcal{T}}^\top\lambda_{\mathcal{T}}\right)$$

Plug back in to Lagrangian:

$$\mathcal{L}(\lambda, \nu) = \lambda_{\mathcal{S}}^\top\mathbf{1}_{\mathcal{S}} + \lambda_{\mathcal{T}}^\top\mathbf{1}_{\mathcal{T}} - \frac{1}{4}\|\nu \cdot (\boldsymbol{\eta} - \boldsymbol{\mu}) + \boldsymbol{Z}^\top\lambda\|^2$$

Here we used the notation $\boldsymbol{Z}$ to denote a matrix whose $i$-th row equals $\mathbf{x}_i y_i$, the set of indices of rows where $\mathbf{x}_i \in S_{\mathcal{S}}$ as $\mathcal{I}_{\mathcal{S}}$, and similarly for $\mathcal{I}_{\mathcal{T},0}, \mathcal{I}_{\mathcal{T},1}$.

Let $\alpha > 0$, to be determined later, $\nu = \frac{(|U|-k)2\alpha}{|U|}$, and define the set $U = \mathcal{I}_{\mathcal{T},1} \cup \mathcal{I}_{\text{Rand}}$ where $\mathcal{I}_{\text{Rand}}$ has $\frac{|U|}{2\alpha}\nu$ indices randomly drawn from $\mathcal{I}_{\mathcal{S}}$ and half from $\mathcal{I}_{\mathcal{T},0}$. Hence, $|U| = N$ Set $\lambda = \frac{\alpha}{|U|}\mathbf{1}_U$ and plug back into the Lagrangian. We have that

$$\boldsymbol{Z}^\top\lambda = \nu\boldsymbol{\mu} + \frac{\alpha}{|U|}\left[k\boldsymbol{\eta} + \sum_{i \in U}\xi_i y_i\right].$$

Hence denoting $\bar{\xi}_U = \frac{1}{|U|}\sum_{i \in U}\xi_i y_i$ as the average added noise vector (multiplied by the label) over set $U$, we have

$$\mathcal{L}(\alpha, \nu) = \alpha - \frac{1}{4}\|\boldsymbol{\eta}(\nu + k\frac{\alpha}{|U|}) + \alpha\bar{\xi}_U\|^2$$

Plugging in the value we already set for $\nu$,

$$\mathcal{L}(\alpha) = \alpha - \frac{1}{4}\|\boldsymbol{\eta}(\frac{\alpha(2|U| - k)}{|U|}) + \alpha\bar{\xi}_U\|^2$$

Taking the maximum over $\alpha$, and denoting the optimal solution to the primal optimization problem by $\mathbf{w}^*$, we get with probability at least $1 - 2\exp(-ct^2)$

$$\|\mathbf{w}^*\|^2 \geq \mathcal{L}(\alpha^*) = 4\|\boldsymbol{\eta}(\frac{2|U| - k}{|U|}) + \bar{\xi}_U\|^{-2}$$

$$= 4\left(r_\eta^2(2 - k/|U|)^2 + \|\bar{\xi}_U\|^2 + 2(2 - k/|U|)\boldsymbol{\eta}^\top\bar{\xi}_U\right)^{-1}$$

$$\geq 4\left(r_\eta^2(2 - k/|U|)^2 + |U|^{-2}\sigma^2(d + 2t\sqrt{d}) + 2(2 - k/|U|)r_\eta\sigma|U|^{-1}t\right)^{-1}$$

**Upper bound on norm of bad solutions under DD:**

Now we will guess a solution $\mathbf{w}_{\text{DD}}$, show it satisfies the constraints of the problem and, under some conditions, achieves lower norm than the bound we obtained in the last section.

**Lemma 3.** *Let $t > 0$ then with probability at least $1 - 2\exp\left[-ct^2k^{-2}\right] - 2\exp\left[-ct^2/2\right] - 3\exp\left[-ct^2/4\right] - 2\exp\left[-ct^2\right]$, the model*

$$\mathbf{w}_{DD} = 2\left(r_\mu^2 - \sigma r_\mu t\right)^{-1} \cdot \boldsymbol{\mu} - 4\sigma^{-2}kd^{-1} \cdot \bar{\xi}_{\mathcal{T},1},$$

*satisfies $y_i\mathbf{w}_{DD}^\top\mathbf{x}_i \geq 1$ for all $i \in U$ and $\mathbf{w}_{DD}^\top(\boldsymbol{\mu} - \boldsymbol{\eta}) \geq 0$.*

*Proof.* For $i \notin \mathcal{I}_{\mathcal{T},1}$,

$$y_i \cdot \mathbf{w}_{\text{DD}}^\top\mathbf{x}_i = \left[2\left(r_\mu^2 - \sigma r_\mu t\right)^{-1} \cdot \boldsymbol{\mu} - 4\sigma^{-2}kd^{-1}\bar{\xi}_{\mathcal{T},1}\right]^\top[\boldsymbol{\mu} + y_i\xi_i]$$

$$= 2(r_\mu^2 - \sigma r_\mu t)^{-1}(r_\mu^2 + y_i\boldsymbol{\mu}^\top\xi_i) - 4\sigma^{-2}kd^{-1}y_i\xi_i^\top\bar{\xi}_{\mathcal{T},1}$$

From eq. (31) we have that $|y_i\xi_i^\top\bar{\xi}_{\mathcal{T},1}| < \sigma^2 t\sqrt{\frac{d}{k}}$ w.p. at least $1 - 2\exp\left[-ct^2\right]$ for $t \leq d\sqrt{|U|}\sigma^2$. Hence, we can say w.p. at least $1 - 2\exp\left[-ct^2\right]$,

$$y_i \cdot \mathbf{w}_{\text{DD}}^\top\mathbf{x}_i \geq 2(r_\mu^2 - \sigma r_\mu t)^{-1}(r_\mu^2 + y_i\boldsymbol{\mu}^\top\xi_i) - 4\sigma^{-2}kd^{-1} \cdot \sigma^2 t\sqrt{\frac{d}{k}}$$

$$= 2(r_\mu^2 - \sigma r_\mu t)^{-1}(r_\mu^2 + y_i\boldsymbol{\mu}^\top\xi_i) - 4t\sqrt{\frac{k}{d}}$$

$$\geq 2(r_\mu^2 - \sigma r_\mu t)^{-1}(r_\mu^2 + y_i\boldsymbol{\mu}^\top\xi_i) - 1 \qquad \text{for } k \leq d/(16t^2) \text{ and } t > 1$$

We also have that $|y_i\boldsymbol{\mu}^\top\xi_i| < t\sigma r_\mu$ w.p. at least $1 - 2\exp(-ct^2/2)$. Putting these together

$$y_i \cdot \mathbf{w}_{\text{DD}}^\top\mathbf{x}_i \geq 2(r_\mu^2 - \sigma r_\mu t)^{-1}(r_\mu^2 - \sigma r_\mu t) - 1 = 1.$$

Now for $i \in \mathcal{I}_{\mathcal{T},1}$, we repeat these steps, but this time use eq. (32) to bound $|\xi_i^\top\bar{\xi}_{\mathcal{T},1}| \geq \frac{\sigma^2 d}{2k}$ with probability at least $1 - 3\exp(-ct^2)$ for $t \leq \frac{\sqrt{d}}{2(2 + \sqrt{|U| - 1})}$ and $|\boldsymbol{\eta}^\top\bar{\xi}_{\mathcal{T},1}| < t\sigma r_\eta k^{-1}$ w.p. at least $1 - 2\exp\left(-ct^2/2\right)$. We get again that

$$y_i \cdot \mathbf{w}_{\text{DD}}^\top\mathbf{x}_i = 2(r_\mu^2 - \sigma r_\mu t)^{-1}(-y_i\boldsymbol{\mu}^\top\boldsymbol{\eta} + y_i \cdot \boldsymbol{\mu}^\top\xi_i) - 4\sigma^{-2}kd^{-1}(-y_i\boldsymbol{\eta}^\top\bar{\xi}_{\mathcal{T},1} + y_i \cdot \xi_i^\top\bar{\xi}_{\mathcal{T},1})$$

$$= 4\sigma^{-2}kd^{-1}(-\boldsymbol{\eta}^\top\bar{\xi}_{\mathcal{T},1} + \xi_i^\top\bar{\xi}_{\mathcal{T},1})$$

$$\geq 4\sigma^{-2}kd^{-1}(-\frac{t\sigma r_\eta}{k} + \frac{\sigma^2 d}{2k})$$

For $r_\eta \leq \frac{\sigma d}{4t}$

$$y_i \cdot \mathbf{w}_{\mathrm{DD}}^\top \mathbf{x}_i \geq 4\sigma^{-2}kd^{-1}(\frac{\sigma^2 d}{4k}) = 1$$

Finally, let us calculate $\mathbf{w}_{\mathrm{DD}}^\top(\boldsymbol{\mu} - \boldsymbol{\eta})$,

$$\begin{aligned}
\mathbf{w}_{\mathrm{DD}}^\top(\boldsymbol{\mu} - \boldsymbol{\eta}) &= 2(r_\mu^2 - \sigma r_\mu t)^{-1}r_\mu^2 + 4\sigma^{-2}kd^{-1}\boldsymbol{\eta}^\top \bar{\xi}_{\mathcal{T},1} \\
&\geq 2(1 - \sigma r_\mu^{-1}t)^{-1} + 4\sigma^{-2}kd^{-1}\sigma r_\eta t k^{-1} \\
&= 2(1 - \sigma r_\mu^{-1}t)^{-1} + 4\sigma^{-1}d^{-1}r_\eta t \\
&\geq 0 \qquad \text{for } r_\mu \geq \sigma t \text{ or more tighter condition is } r_\eta \geq \frac{\sigma d}{2t}(\sigma r_\mu^{-1}t - 1)^{-1}.
\end{aligned}$$

The first inequality holds w.p. at least $1 - 2\exp\{-ct^2\}$. Hence, for $\frac{\sigma d}{2t}(\sigma r_\mu^{-1}t - 1)^{-1} \leq r_\eta \leq \frac{\sigma d}{4t}$, we have w.p. at least $1 - 2\exp\{-ct^2\}$ that

$$\mathbf{w}_{\mathrm{DD}}^\top(\boldsymbol{\mu} - \boldsymbol{\eta}) > 0$$

Overall, taking the union bounds so that these inequalities hold over the entire dataset, we get that the solution $\mathbf{w}_{\mathrm{DD}}$ satisfies all the constraints with probability $1 - 3|U|\exp(-ct^2/4) - 2\exp(-ct^2/2) - 2\exp(-ct^2)$. $\qquad\square$

Requirements for parameters:

- $t \leq d\sqrt{|U|}\sigma^2$

- $r_\mu \geq \sigma t$

- $t \leq \frac{\sqrt{d}}{2(2 + \sqrt{|U|-1})}$

- $\frac{\sigma d}{2t}(\sigma r_\mu^{-1}t - 1)^{-1} \leq r_\eta \leq \frac{\sigma d}{4t}$ (Lower bound is negative, so can be ignored)

**Finding ranges where AU-ROC is smaller than 0.5:**
Compare $\|\mathbf{w}_{\mathrm{DD}}\|$ and lower bounds on $\|\mathbf{w}^*\|$ obtained in earlier parts and find regions where we must have $\|\mathbf{w}_{\mathrm{DD}}\| < \|\mathbf{w}^*\|$

From Section A.2

We have that

$$\|\mathbf{w}_{\mathrm{DD}}\| = \left\| \frac{2}{r_\mu^2 - \sigma r_\mu t} \cdot \boldsymbol{\mu} - \frac{4k}{\sigma^2 d}\bar{\xi}_{\mathcal{T},1} \right\|.$$

Hence from triangle inequality,

$$\|\mathbf{w}_{\mathrm{DD}}\| \leq \frac{2}{r_\mu^2 - \sigma r_\mu t_1} \cdot \|\boldsymbol{\mu}\| + \frac{4k}{\sigma^2 d} \cdot \|\bar{\xi}_{\mathcal{T},1}\|.$$

We know that $\|\boldsymbol{\mu}\| = r_\mu$ and $\bar{\xi}_{\mathcal{T},1} \sim \mathcal{N}(0, \frac{\sigma^2}{k}I_d)$. Hence, using the concentration bound in 29, we have with probability at least $1 - \exp\left(-\frac{t^2}{4}\right)$ that

$$\|\bar{\xi}_{\mathcal{T},1}\| \leq \sigma\sqrt{\frac{d + 2t\sqrt{d}}{k}}$$

Hence with probability at least $1 - \exp\left(-\frac{t^2}{4}\right)$,

$$\|\mathbf{w}_{\text{DD}}\| \leq \frac{2}{r_\mu - \sigma t} + \frac{4k}{\sigma d}\sqrt{\frac{d + 2t\sqrt{d}}{k}}$$

$$\leq \frac{2}{r_\mu - \sigma t} + 8\sigma^{-1}\sqrt{\frac{k}{d}} \qquad \text{for } t \leq \frac{3}{2}\sqrt{d}$$

We also have from section A.2,

$$\|\mathbf{w}^*\| \geq 2\|\boldsymbol{\eta}(\frac{2|U| - k}{|U|}) + \bar{\xi}_U\|^{-1}$$

$$\|\mathbf{w}^*\| \geq 2(\|\boldsymbol{\eta}\|(\frac{2|U| - k}{|U|}) + \|\bar{\xi}_U\|)^{-1}$$

Similarly, using concentration bound in 29 for $\|\bar{\xi}_U\|$, we get with probability at least $1 - \exp\left(-\frac{t^2}{4}\right)$,

$$\|\mathbf{w}^*\| \geq \frac{2}{r_\eta(2 - k/|U|) + \sigma\sqrt{\frac{d + 2t\sqrt{d}}{|U|}}}$$

For $t \leq \frac{3}{2}\sqrt{d} \implies \sqrt{d + 2t\sqrt{d}} \leq 2\sqrt{d}$, hence

$$\|\mathbf{w}^*\| \geq \frac{2}{r_\eta(2 - k/|U|) + 2\sigma\sqrt{\frac{d}{|U|}}}$$

We know that $k \geq 0$, Hence, the following inequality will guarantee 6 with probability at least $1 - \exp\left(-\frac{t^2}{4}\right)$,

$$\|\mathbf{w}^*\| \geq \frac{2}{2r_\eta + 2\sigma\sqrt{\frac{d}{|U|}}} \tag{5}$$

By union bounds w.p. at least $1 - 2\exp\left(-\frac{t^2}{4}\right)$, we want conditions for

$$\frac{2}{2r_\eta + 2\sigma\sqrt{\frac{d}{|U|}}} \geq \frac{2}{r_\mu - \sigma t} + 8\sigma^{-1}\sqrt{\frac{k}{d}}$$

$$\frac{1}{2r_\eta + 2\sigma\sqrt{\frac{d}{|U|}}} \geq \frac{1}{r_\mu - \sigma t} + \frac{4\sqrt{k}}{\sigma\sqrt{d}}$$

$$2r_\eta + 2\sigma\sqrt{\frac{d}{|U|}} \leq \left(\frac{1}{r_\mu - \sigma t} + \frac{4\sqrt{k}}{\sigma\sqrt{d}}\right)^{-1} \tag{6}$$

For $r_\mu \geq 2\sigma t$, $\left(\frac{2}{r_\mu} + \frac{4\sqrt{k}}{\sigma\sqrt{d}}\right)^{-1} \leq \left(\frac{1}{r_\mu - \sigma t} + \frac{2\sqrt{k}}{\sigma\sqrt{d}}\right)^{-1}$. Hence, eq. 6 is satisfied if the following is guaranteed,

$$2r_\eta + 2\sigma\sqrt{\frac{d}{|U|}} \leq \left(\frac{2}{r_\mu} + \frac{4\sqrt{k}}{\sigma\sqrt{d}}\right)^{-1}$$

$$r_\eta \leq \frac{\sigma\sqrt{d}}{4}\left(\frac{r_\mu}{\sigma\sqrt{d} + 2r_\mu\sqrt{k}} - \frac{4}{\sqrt{|U|}}\right)$$

Or more tighter condition that will guarantee eq. 6 is the following using $r_\mu \geq 2\sigma t$

$$r_\eta \leq \frac{\sigma\sqrt{d}}{4}\left(\frac{2t}{\sqrt{d}+4t\sqrt{k}} - \frac{4}{\sqrt{|U|}}\right)$$

This means that the above parameter requirements are sufficient to guarantee that a domain discriminator is a suboptimal solution to the novelty detection problem under background shift with probability at least $1 - 2\exp\left(-\frac{t^2}{4}\right)$.

Parameter requirements to guarantee eq. 6 w.p. at least $1 - 2\exp\left(-\frac{t^2}{4}\right)$:

- $t \leq d\sqrt{|U|}\sigma^2$

- $t \leq \frac{3}{2}\sqrt{d}$          (redundant see next condition below.)

- $t \leq \frac{\sqrt{d}}{2(2+\sqrt{|U|-1})}$

- $r_\mu \geq 2\sigma t$

- $\frac{\sigma d}{2t}(\sigma r_\mu^{-1}t - 1)^{-1} \leq r_\eta \leq \frac{\sigma d}{4t}$          (Negative lower bound, so can be ignoredy)

- $r_\eta \leq \frac{\sigma\sqrt{d}}{4}\left(\frac{r_\mu}{\sigma\sqrt{d}+2r_\mu\sqrt{k}} - \frac{4}{\sqrt{|U|}}\right)$

$\square$

**Proposition 2.** *For $\sigma = 1/\sqrt{d}$, there exist some constants $C_d$ and $c_d$, and for any failure probability $0 \leq \delta \leq 1$, if*

$$C_d \log(\frac{c_d}{\delta}) \leq \frac{\sqrt{d}}{4\sqrt{N_\mathcal{S} + N_\mathcal{T}}},$$

$$\frac{1}{2\sqrt{N_\mathcal{T}}} \leq r_\mu \leq \min\left(2\sqrt{N_\mathcal{T}}(1 - 5\tau), 16\sqrt{N_{\mathcal{T},1}}\tau\right),$$

$$r_\eta \geq \frac{1}{2\sqrt{N_{\mathcal{T},1}}},$$

$$\frac{3}{r_\eta} + \frac{3 - 5\tau}{r_\mu} \leq \frac{3}{8}\sqrt{N_{\mathcal{T},1}} - 2$$

*then with probability at least $1-\delta$ over the drawing of $(S_\mathcal{S}, S_\mathcal{T})$ as described in problem, $AU-ROC(\hat{\mathbf{w}}_{color}) \geq 0.9$ as defined in first part of appendix A.2 is at least $Q\left(\frac{-\tau\sqrt{d}}{\sqrt{2}\|\hat{\mathbf{w}}_{color}\|}\right) \geq Q\left(\frac{-\tau\sqrt{d}}{\sqrt{2}(\frac{3}{r_\eta} + \frac{3-5\tau}{r_\mu}+2)}\right) \geq Q\left(\frac{-\tau\sqrt{d}}{\sqrt{2}(1-\tau)}\right).$*

*Proof.* **Lower bound on norm of bad solutions under CoLOR:**
This time the Lagrangian is

$$\mathcal{L}(\mathbf{w}, \lambda, \nu) = \|\mathbf{w}\|^2 - \lambda_\mathcal{S}^\top(\mathbf{X}_\mathcal{S}\mathbf{w}) + \lambda_\mathcal{T}^\top(\mathbf{1}_\mathcal{T} + \mathbf{X}_\mathcal{T}\mathbf{w}) - \nu \cdot (\mathbf{w}^\top(\boldsymbol{\mu} - \boldsymbol{\eta}) + \tau)$$

and zero the gradient

$$\nabla_\mathbf{w}\mathcal{L}(\mathbf{w}, \lambda, \nu) = 2\mathbf{w} - \mathbf{X}_\mathcal{S}^\top\lambda_\mathcal{S} + \mathbf{X}_\mathcal{T}^\top\lambda_\mathcal{T} - \nu \cdot (\boldsymbol{\mu} - \boldsymbol{\eta}) = 0$$

$$\Leftrightarrow \mathbf{w} = \frac{1}{2}\left(\nu \cdot (\boldsymbol{\mu} - \boldsymbol{\eta}) + \mathbf{X}_\mathcal{S}^\top\lambda_\mathcal{S} - \mathbf{X}_\mathcal{T}^\top\lambda_\mathcal{T}\right)$$

Plug back in to Lagrangian:

$$\mathcal{L}(\lambda, \nu) = -\nu \cdot \tau + \lambda_\mathcal{T}^\top\mathbf{1}_\mathcal{T} - \frac{1}{4}\|\nu \cdot (\boldsymbol{\mu} - \boldsymbol{\eta}) + \mathbf{Z}^\top\lambda\|^2$$

Consider setting $\lambda = \frac{\nu}{|U|}\mathbf{1}_{S_{\mathcal{T}}}$, where $U$ has randomly drawn $|U|$ ($|U| = N_{\mathcal{T}}$) examples from $S_{\mathcal{T}}$. Also set $\nu = \frac{(1-\tau)}{\|\bar{\xi}_{S_{\mathcal{T}}}\|^2}$. We have

$$
\begin{aligned}
\mathbf{Z}^\top \lambda &= -\nu \cdot \boldsymbol{\mu} + \nu \bar{\xi}_{S_{\mathcal{T},0}} + \nu \cdot \boldsymbol{\eta} + \nu \bar{\xi}_{S_{\mathcal{T},1}} \\
&= \nu(\boldsymbol{\eta} - \boldsymbol{\mu}) + \nu \bar{\xi}_{S_{\mathcal{T}}}
\end{aligned}
$$

and then the lagrangian

$$
\begin{aligned}
\mathcal{L}(\lambda, \nu) &= \nu \cdot (1 - \tau) - \frac{1}{4}\|\nu \bar{\xi}_{S_{\mathcal{T}}}\|^2 \\
&= \nu \cdot (1 - \tau) - \frac{1}{4}\nu^2 \left(\|\bar{\xi}_{S_{\mathcal{T}}}\|^2\right) \| \\
\mathcal{L}(\lambda) &= \frac{3}{4}\frac{(1-\tau)^2}{\|\bar{\xi}_{S_{\mathcal{T}}}\|^2}
\end{aligned}
$$

$$
\|\hat{\mathbf{w}}_{\text{color}}\|^2 \geq \mathcal{L}(\lambda^*) = \frac{3}{4}\frac{(1-\tau)^2}{\|\bar{\xi}_{S_{\mathcal{T}}}\|^2}
$$

**Getting feasible solution for our optimization problem:** We will guess a solution of the form:

$$
\mathbf{w}_{\text{color}} = \alpha \boldsymbol{\eta} + (\frac{\alpha r_\eta^2}{r_\mu^2} - \frac{5\tau}{r_\mu^2})\boldsymbol{\mu} - \beta \bar{\xi}_{S_{\mathcal{T},0}}
$$

$$
\mathbf{w}_{\text{color}} = 2\frac{\boldsymbol{\eta}}{r_\eta^2} + (\frac{2}{r_\mu^2} - \frac{5\tau}{r_\mu^2})\boldsymbol{\mu} - \frac{1}{\sigma^2} \cdot \frac{|U| - k}{d}\bar{\xi}_{S_{\mathcal{T},0}}
$$

$$
\mathbf{w}_{\text{color}} = 2\frac{\boldsymbol{\eta}}{r_\eta^2} + (\frac{2}{r_\mu^2} - \frac{5\tau}{r_\mu^2})\boldsymbol{\mu} - (|U| - k)\bar{\xi}_{S_{\mathcal{T},0}}
$$

To have $\mathbf{w}_{\text{color}}^\top \mathbf{x}_i y_i > 1$ for $i \in \mathcal{I}_{\mathcal{T},1}$, we need

$$
\mathbf{w}_{\text{color}}^\top(\boldsymbol{\eta} - \xi_i) = \alpha r_\eta^2 + \beta \xi_i^\top \bar{\xi}_{S_{\mathcal{T},0}} > 1
$$

We have

$$
\begin{aligned}
\mathbf{w}_{\text{color}}^\top(-y_i\boldsymbol{\eta} + y_i\xi_i) &= \mathbf{w}_{\text{color}}^\top(\boldsymbol{\eta} - \xi_i) \\
&= \alpha r_\eta^2 - \alpha \boldsymbol{\eta}^\top \xi_i + \beta \xi_i^\top \bar{\xi}_{\mathcal{T},0}
\end{aligned}
$$

We know that $|\boldsymbol{\eta}^\top \xi_i| < \sigma t r_\eta$ w.p. at least $1 - 2\exp\left(-ct^2/2\right)$ and $|\xi_i \bar{\xi}_{\mathcal{T},0}| \leq \sigma^2 t\sqrt{\frac{d}{|U|-k}}$ w.p. at least $1 - 2\exp(-ct^2)$ for $t \leq d\sqrt{|U|}\sigma^2$. Hence, w.p. at least $1 - 2\exp\left(-ct^2/2\right) - 2\exp(-ct^2)$, we can say that,

$$
\mathbf{w}_{\text{color}}^\top(\boldsymbol{\eta} - \xi_i) \geq \alpha r_\eta^2 - \alpha \sigma t r_\eta - \beta \sigma^2 t\sqrt{\frac{d}{|U| - k}}
$$

Required condition:

$$
\alpha r_\eta^2 - \alpha \sigma t r_\eta - \beta \sigma^2 t\sqrt{\frac{d}{|U| - k}} \geq 1
$$

Possibly set $\alpha = 3r_\eta^{-2}$ and $\beta = \sigma^{-2}\frac{\sqrt{|U|-k}}{d}$.

$$
3 - 3\sigma t r_\eta^{-1} - \frac{t}{\sqrt{d}} \geq 1
$$

$$\frac{t}{\sqrt{d}} \le 2 - 3\sigma t r_\eta^{-1}$$

For $2\sigma t \le r_\eta$,

$$\frac{t}{\sqrt{d}} \le \frac{1}{2}$$

Parameter requirements:

- $\sigma t \le \frac{r_\eta}{2}$
- $t \le \frac{\sqrt{d}}{2}$

To also have $\mathbf{w}_{\text{color}}^\top \mathbf{x}_i y_i > 1$ for $i \in \mathcal{I}_{\mathcal{T},0}$ we need

$$\mathbf{w}_{\text{color}}^\top (\boldsymbol{\mu} + \xi_i) = (\frac{\alpha r_\eta^2}{r_\mu^2} - \frac{5\tau}{r_\mu^2}) r_\mu^2 + \beta \xi_i^\top \bar{\xi}_{S_{\mathcal{T},0}} > 1$$

We have

$$\mathbf{w}_{\text{color}}^\top (-y_i \boldsymbol{\mu} + y_i \xi_i) = \mathbf{w}_{\text{color}}^\top (\boldsymbol{\mu} - \xi_i)$$
$$= \left( \frac{\alpha r_\eta^2}{r_\mu^2} - \frac{5\tau}{r_\mu^2} \right) (\boldsymbol{\mu}^\top \boldsymbol{\mu} - \boldsymbol{\mu}^\top \xi_i) - \beta (\boldsymbol{\mu}^\top \bar{\xi}_{\mathcal{T},0} - \xi_i^\top \bar{\xi}_{\mathcal{T},0})$$

We know that $|\boldsymbol{\mu}^\top \xi_i| \le \sigma t r_\mu$ w.p. at least $1 - 2\exp\left(-ct^2/2\right)$ and $|\xi_i^\top \bar{\xi}_{\mathcal{T},0}| \ge \frac{\sigma^2 d}{2(|U|-k)}$ w.p. at least $1 - 2\exp(-ct^2)$ for $t \le \sigma^2$ and $t \le \frac{\sqrt{d}}{2(2+\sqrt{|U|-1})}$. We also know that $|\boldsymbol{\mu}^\top \bar{\xi}_{\mathcal{T},0}| \le \frac{\sigma t r_\mu}{|U|-k}$ w.p. at least $1 - 2\exp(-ct^2/2)$ Hence, w.p. at least $1 - 4\exp\left(-ct^2/2\right) - 2\exp(-ct^2)$, we can say that,

$$\mathbf{w}_{\text{color}}^\top (-y_i \boldsymbol{\mu} + y_i \xi_i) \ge \left( \frac{\alpha r_\eta^2}{r_\mu^2} - \frac{5\tau}{r_\mu^2} \right) (r_\mu^2 - \sigma t r_\mu) + \beta \left( -\frac{\sigma t r_\mu}{|U| - k} + \frac{\sigma^2 d}{2(|U| - k)} \right)$$
$$\ge \frac{r_\mu^2}{2} \left( \frac{\alpha r_\eta^2}{r_\mu^2} - \frac{5\tau}{r_\mu^2} \right) + \beta \left( \frac{\sigma^2 d - r_\mu^2}{2(|U| - k)} \right) \qquad \text{for } r_\mu \ge 2\sigma t$$

Required condition:

$$\frac{1}{2} \left( \alpha r_\eta^2 - 5\tau \right) + \beta \left( \frac{\sigma^2 d - 2\sigma t r_\mu}{2(|U| - k)} \right) \ge 1$$

Setting $\alpha = 3 r_\eta^{-2}$ we get,

$$\frac{1}{2} \left( 3 - 5\tau \right) + \beta \left( \frac{\sigma^2 d - 2\sigma t r_\mu}{2(|U| - k)} \right) \ge 1$$
$$\beta \left( \frac{\sigma^2 d - 2\sigma t r_\mu}{2(|U| - k)} \right) \ge \frac{5\tau - 1}{2}$$
$$\frac{\beta \sigma t r_\mu}{|U| - k} \le \frac{\beta \sigma^2 d}{2(|U| - k)} + \frac{1 - 5\tau}{2}$$

Setting $\beta = \sigma^{-2} \frac{\sqrt{|U| - k}}{d}$ we get,

$$\frac{r_\mu t}{\sigma d} \le \frac{1}{2} + \frac{1 - 5\tau}{2} \sqrt{|U| - k}$$

Parameter requirements:

- $2\sigma t \le r_\mu$

- $r_\mu t \le \frac{\sigma d}{2}(1 + (1 - 5\tau)\sqrt{|U| - k})$

Finally, we also require $\mathbf{w}_{\text{color}}^\top \mathbf{x}_i y_i > 0$ for $i \in \mathcal{I}_{\mathcal{S},0}$, we need

We have

$$\mathbf{w}_{\text{color}}^\top (y_i \boldsymbol{\mu} + y_i \xi_i) = \mathbf{w}_{\text{color}}^\top (\boldsymbol{\mu} + \xi_i)$$
$$= \left( \frac{\alpha r_\eta^2}{r_\mu^2} - \frac{5\tau}{r_\mu^2} \right)(\boldsymbol{\mu}^\top \boldsymbol{\mu} + \boldsymbol{\mu}^\top \xi_i) - \beta(\boldsymbol{\mu}^\top \bar{\xi}_{\mathcal{T},0} + \xi_i^\top \bar{\xi}_{\mathcal{T},0})$$

We know that $|\boldsymbol{\mu}^\top \xi_i| \le \sigma t r_\mu$ w.p. at least $1 - 2\exp\left(-ct^2/2\right)$ and $|\xi_i^\top \bar{\xi}_{\mathcal{T},0}| \le \sigma^2 t \sqrt{\frac{d}{|U|-k}}$ w.p. at least $1 - 2\exp(-ct^2)$ for $t \le \sigma^2$ and $t \le \frac{\sqrt{d}}{2(2+\sqrt{|U|-1})}$. We also know that $|\boldsymbol{\mu}^\top \bar{\xi}_{\mathcal{T},0}| \le \frac{\sigma t r_\mu}{|U|-k}$ w.p. at least $1 - 2\exp(-ct^2/2)$ Hence, w.p. at least $1 - 4\exp\left(-ct^2/2\right) - 2\exp(-ct^2)$, we can say that,

$$\mathbf{w}_{\text{color}}^\top (-y_i \boldsymbol{\mu} + y_i \xi_i) \ge \left( \frac{\alpha r_\eta^2}{r_\mu^2} - \frac{5\tau}{r_\mu^2} \right)(r_\mu^2 - \sigma t r_\mu) - \beta \left( \frac{\sigma t r_\mu}{|U| - k} + \sigma^2 t \sqrt{\frac{d}{|U| - k}} \right)$$
$$\ge \frac{r_\mu^2}{2} \left( \frac{\alpha r_\eta^2}{r_\mu^2} - \frac{5\tau}{r_\mu^2} \right) - \beta \left( \frac{\sigma t r_\mu}{|U| - k} + \sigma^2 t \sqrt{\frac{d}{|U| - k}} \right) \qquad \text{for } r_\mu \ge 2\sigma t$$

Required condition:

$$\frac{1}{2} \left( \alpha r_\eta^2 - 5\tau \right) - \beta \left( \frac{\sigma t r_\mu}{|U| - k} + \sigma^2 t \sqrt{\frac{d}{|U| - k}} \right) \ge 0$$

Using $\alpha$ and $\beta$,

$$\frac{3 - 5\tau}{2} - \left( \frac{t r_\mu}{\sigma d \sqrt{|U| - k}} + \frac{t}{\sqrt{d}} \right) \ge 0$$

The minimum value of the LHS above would be $\frac{3 - 5\tau}{2} - \left( \frac{t r_\mu}{\sigma d \sqrt{|U|-k}} + \frac{1}{2} \right)$ We want this term to be greater than 0. Hence,

$$\frac{1 - 5\tau}{2} - \frac{t r_\mu}{\sigma d \sqrt{|U| - k}} \ge 0$$
$$\frac{t r_\mu}{\sigma d} \le \frac{1 - 5\tau}{2}\sqrt{|U| - k}$$

Substituting $\alpha = 3r_\eta^{-2}$ and $\beta = \sigma^{-2}\frac{\sqrt{|U|-k}}{d}$, we get,

$$\frac{t r_\mu}{\sigma d} \le \frac{1 - 5\tau}{2}\sqrt{|U| - k}$$

Parameter requirements:

- $t \le \frac{\sqrt{d}}{2}$

- $r_\mu t \le \frac{1}{2}\sigma d(1 - 5\tau)\sqrt{|U| - k}$

We want $\mathbf{w}_{\text{color}}^\top(\boldsymbol{\mu} - \boldsymbol{\eta}) \leq -\tau$ i.e. $\mathbf{w}_{\text{color}}^\top(\boldsymbol{\eta} - \boldsymbol{\mu}) \geq \tau$

We have

$$\mathbf{w}_{\text{color}}^\top(\boldsymbol{\eta} - \boldsymbol{\mu}) = \alpha\boldsymbol{\eta}^\top\boldsymbol{\eta} - \left(\frac{\alpha r_\eta^2}{r_\mu^2} - \frac{5\tau}{r_\mu^2}\right)\boldsymbol{\mu}^\top\boldsymbol{\mu} + \beta \cdot \boldsymbol{\mu}^\top\bar{\xi}_{\mathcal{T},0}$$

We know that $|\boldsymbol{\mu}^\top\bar{\xi}_{\mathcal{T},0}| \leq \frac{\sigma tr_\mu}{|U|-k}$ w.p. at least $1 - 2\exp(-ct^2/2)$ Hence, w.p. at least $1 - 2\exp\left(-ct^2/2\right)$, we can say that,

$$\mathbf{w}_{\text{color}}^\top(\boldsymbol{\eta} - \boldsymbol{\mu}) \geq \alpha r_\eta^2 - \left(\frac{\alpha r_\eta^2}{r_\mu^2} - \frac{5\tau}{r_\mu^2}\right)r_\mu^2 - \beta\frac{\sigma tr_\mu}{|U|-k}$$

Required condition:

$$5\tau - \beta\frac{\sigma tr_\mu}{|U|-k} \geq \tau$$

$$\beta\frac{\sigma tr_\mu}{|U|-k} \leq 4\tau$$

$$\frac{tr_\mu}{\sigma d\sqrt{|U|-k}} \leq 4\tau$$

Parameter requirements:

- $\frac{tr_\mu}{\sigma d} \leq 4\tau\sqrt{|U|-k}$

Bound on the norm of $\mathbf{w}_{\text{color}}$:

$$\mathbf{w}_{\text{color}} = 3\frac{\boldsymbol{\eta}}{r_\eta^2} + \left(\frac{3}{r_\mu^2} - \frac{5\tau}{r_\mu^2}\right)\boldsymbol{\mu} - \sigma^{-2}\frac{\sqrt{|U|-k}}{d}\bar{\xi}_{S_\mathcal{T},0}$$

$$\|\mathbf{w}_{\text{color}}\| \leq \frac{3}{r_\eta} + \left(\frac{3}{r_\mu} - \frac{5\tau}{r_\mu}\right) + \sigma^{-1}\frac{\sqrt{|U|-k}}{d}\sqrt{\frac{d + 2t\sqrt{d}}{|U|-k}}$$

$$\|\mathbf{w}_{\text{color}}\| \leq \frac{3}{r_\eta} + \frac{3}{r_\mu} - \frac{5\tau}{r_\mu} + 2$$

$$\|\mathbf{w}_{\text{color}}\| \leq \frac{3}{r_\eta} + \frac{3 - 5\tau}{r_\mu} + 2$$

$$\text{for } t \leq \frac{3}{2}\sqrt{d}$$

We also want $\|\mathbf{w}_{\text{color}}\| \leq \|\hat{\mathbf{w}}_{\text{color}}\|$. For this, it is sufficient to ensure the following w.p. at least $1 - 2\exp(-ct^2)$:

$$\frac{3}{r_\eta} + \frac{3 - 5\tau}{r_\mu} + 2 \leq \frac{3}{4}\frac{1 - \tau}{\sqrt{(|U|)^{-1}\sigma^2(d + 2t\sqrt{d})}}$$

$$\leq \frac{3}{8}\frac{1 - \tau}{\sqrt{(|U|)^{-1}\sigma^2 d}} \quad \text{for } t \leq \frac{3}{2}\sqrt{d}$$

$$\leq \frac{3}{8}(1 - \tau)\sqrt{|U|} \quad \text{for } \sigma = \frac{1}{\sqrt{d}}$$

$$\frac{3}{r_\eta} + \frac{3 - 5\tau}{r_\mu} \leq \frac{3}{8}(1 - \tau)\sqrt{|U|} - 2$$

which satisfies the constraint in proposition 2.

Parameter requirements if $\alpha = 2r_\eta^{-2}$ and $\beta = \sigma^{-2}\frac{|U|-k}{d}$:

- $t \leq \frac{\sqrt{d}}{2(2+\sqrt{|U|-1})}$

- $t \leq \frac{\sqrt{d}}{2}$

- $2\sigma t \leq r_\eta$

- $2\sigma t \leq r_\mu$

- $r_\mu t \leq \min\left(\frac{1}{2}\sigma d(1-5\tau)\sqrt{|U|-k}, 4\sigma d\tau\sqrt{|U|-k}\right)$

- $\frac{3}{r_\eta} + \frac{3-5\tau}{r_\mu} \leq \frac{3}{8}(1-\tau)\sqrt{|U|} - 2$

Set $\sigma = \frac{1}{\sqrt{d}}$ and $t = \frac{\sqrt{d}}{4\sqrt{|U|}}$

Hence, $\frac{1}{2\sqrt{|U|}} \leq r_\eta$

$\frac{1}{2\sqrt{|U|}} \leq r_\mu$

$r_\mu \leq \min\{2\sqrt{|U|}(1-5\tau), 8\sqrt{|U|}\tau\}$

Now we want,

$$Q\left(\frac{-\tau\sqrt{d}}{\sqrt{2}(1-\tau)}\right) \geq 0.9$$

This means we want,

$$\frac{-\tau\sqrt{d}}{\sqrt{2}(1-\tau)} \leq -1.283$$

$$\frac{\tau\sqrt{d}}{\sqrt{2}(1-\tau)} \geq 1.283$$

$$\tau\left(\sqrt{\frac{d}{2}} + 1.283\right) \geq 1.283$$

$$\tau \geq \frac{1.283}{\sqrt{\frac{d}{2}} + 1.283}$$

From parameter requirements, we can also say,

$$\frac{1.283}{\sqrt{\frac{d}{2}} + 1.283} \leq \tau \leq 0.2 \tag{7}$$

For $d > 300$, we can easily say pick $\tau = 0.1$. This gives us $r_\mu t \leq \min\left(\frac{1}{4}\sigma d\sqrt{|U|-k}, 0.4\sigma d\sqrt{|U|-k}\right) = \frac{1}{4}\sigma d\sqrt{|U|-k}$ This gives us final parameter ranges as follows,

- $t \leq \frac{\sqrt{d}}{2(2+\sqrt{|U|-1})}$

- $t \leq \frac{\sqrt{d}}{2}$

- $2\sigma t \leq r_\eta$

- $2\sigma t \leq r_\mu$

- $r_\mu t \leq \frac{1}{4}\sigma d\sqrt{|U|-k}$

- $\frac{3}{r_\eta} + \frac{3-5\tau}{r_\mu} \leq \frac{3}{8}(1-\tau)\sqrt{|U|} - 2$

$\square$

**Theorem 2.** *(Restating theorem 1)*
*Consider a Linear-Gaussian PU-learning problem with parameters $\boldsymbol{\mu}, \boldsymbol{\eta}, d > 300$, dataset sizes $N_{\mathcal{T}} > 10$, $N = N_{\mathcal{S}} + N_{\mathcal{T}}$, $\alpha \in (0, \frac{N}{1024 N_{\mathcal{T}}})$ and let $\delta \in (0, 1)$. For all problems where*

$$\min\{r_\eta, r_\mu\} \geq \frac{16}{\sqrt{N_{\mathcal{T}}}}, \tag{8}$$

$$r_\mu \leq \frac{1}{2}\sqrt{N_{\mathcal{T},0}}, \tag{9}$$

$$\frac{r_\eta}{r_\mu} \leq \frac{4}{N}, \tag{10}$$

$$c_1 \log\left(\frac{c_2}{\delta}\right) \leq \min\left(\sqrt{N}, \sqrt{\frac{d}{N}}\right) \tag{11}$$

*it holds with probability at least $1 - \delta$ that $\mathrm{AU-ROC}(\mathbf{w}_{DD}) < 0.5$ and $\mathrm{AU-ROC}(\mathbf{w}_{color}) > 0.9$.*

*Proof.* Let us denote $p = \frac{N}{N_{\mathcal{T}}}$. Then based on the stated assumptions on $N_{\mathcal{S}}, N_{\mathcal{T}}, N_{\mathcal{T},0}, N_{\mathcal{T},1}$ we can say,

$$
\begin{aligned}
\sqrt{N \cdot N_{\mathcal{T},1}} &= \sqrt{pN_{\mathcal{T}} \cdot N_{\mathcal{T},1}} \\
&\geq \sqrt{pN_{\mathcal{T},0} \cdot N_{\mathcal{T},1}} \\
&= (\sqrt{p} - 1)\sqrt{N_{\mathcal{T},0} \cdot N_{\mathcal{T},1}} + \sqrt{N_{\mathcal{T},0} \cdot N_{\mathcal{T},1}} \\
&\geq 1 + \sqrt{N_{\mathcal{T},0} \cdot N_{\mathcal{T},1}}
\end{aligned}
\tag{12}
$$

Using eq. (9) we can write, $1 + \sqrt{N_{\mathcal{T},0} \cdot N_{\mathcal{T},1}} \geq 1 + 2r_\mu\sqrt{N_{\mathcal{T},1}}$

From eq. (8) we know that $r_\mu\sqrt{N} \geq 16$. Hence,

$$
\begin{aligned}
r_\mu\sqrt{N} &\geq 8 + \frac{1}{2}r_\mu\sqrt{N} \\
&\geq 8 + \frac{1}{2}r_\mu\sqrt{\frac{p}{\alpha} \cdot N_{\mathcal{T},1}} \\
&\geq 8 + 16r_\mu\sqrt{N_{\mathcal{T},1}} = 8(1 + 2r_\mu\sqrt{N_{\mathcal{T},1}})
\end{aligned}
\tag{13}
$$

Here, the first transition is from eq. (8) while second and third from the range of $\alpha = \frac{pN_{\mathcal{T},1}}{N} \leq \frac{p}{1024}$.

From the proposition 1 and proposition 2, we have the following ranges of parameter ranges for $t = c_1 \log(\frac{c_2}{\delta})$ where the $\mathrm{AUROC}(\mathbf{w}_{DD}) < 0.5$ and $\mathrm{AUROC}(\mathbf{w}_{color}) > 0.9$ with probability at least $1 - \delta$:

$$\sqrt{N} \geq c_1 \log(\frac{c_2}{\delta}) \tag{14}$$

$$\sqrt{d} \geq c_1 \sqrt{N} \log(\frac{c_2}{\delta}) \tag{15}$$

$$\frac{1}{2\sqrt{N_{\mathcal{T}}}} \leq \min(r_\eta, r_\mu) \quad \text{which is guaranteed by eq. (8)} \tag{16}$$

$$r_\mu \leq \sqrt{N \cdot N_{\mathcal{T},0}} \quad \text{which is guaranteed by eq. (9)} \tag{17}$$

$$\frac{r_\eta}{r_\mu} \leq \frac{1}{4}\left(\frac{1}{1 + 2r_\mu\sqrt{N_{\mathcal{T},1}}} - \frac{4}{r_\mu\sqrt{N}}\right) \tag{18}$$

$$\frac{3}{r_\eta} + \frac{3 - 5\tau}{r_\mu} \leq \frac{27}{80}\sqrt{N} - 2 \tag{19}$$

Then putting this together, for parameters that satisfy the eq. (11) we have,

$$\frac{r_\eta}{r_\mu} \leq \frac{4}{N} = 4 \cdot \sqrt{\frac{\alpha}{N \cdot N_{\mathcal{T},1}p}}$$

$$\leq \frac{1}{8}(N \cdot N_{\mathcal{T},1})^{-1/2} \quad \text{as } \alpha \leq \frac{p}{1024}$$

$$\leq \frac{1}{8}(1 + 2r_\mu\sqrt{N_{\mathcal{T},1}})^{-1} \quad \text{from eq. (12)}$$

$$\leq \frac{1}{4}\left(\frac{1}{1 + 2r_\mu\sqrt{N_{\mathcal{T},1}}} - \frac{4}{r_\mu\sqrt{N}}\right) \quad \text{from eq. (13)}$$

which satisfies the constraint in proposition 1.

$$\frac{1}{2\sqrt{N_{\mathcal{T}}}} \leq r_\mu < \min\left(\frac{1}{2}\sqrt{d}(1-5\tau)\sqrt{N_{\mathcal{T},0}}, 4\sqrt{d}\tau\sqrt{N_{\mathcal{T},0}}\right),$$

$$< \frac{1}{4}\sqrt{dN_{\mathcal{T},0}} \quad \text{using } \tau = 0.1$$

$$\frac{3}{r_\eta} + \frac{3-5\tau}{r_\mu} \leq \frac{3}{8}(1-\tau)\sqrt{N_{\mathcal{T}}} - 2$$

$$\frac{3}{r_\eta} + \frac{2.5}{r_\mu} \leq \frac{27}{80}\sqrt{N_{\mathcal{T}}} - 2 \quad \text{using } \tau = 0.1$$

$$3 + \frac{2.5r_\eta}{r_\mu} \leq \frac{27}{80}r_\eta(\sqrt{N_{\mathcal{T}}} - 2)$$

$$\frac{r_\eta}{r_\mu} \leq \frac{10}{25}\left(\frac{27}{80}r_\eta(\sqrt{N_{\mathcal{T}}} - 2) - 3\right)$$

We know from eq. (8) that $r_\eta(\sqrt{N_{\mathcal{T}}} - 2) \geq 16 \implies \frac{27}{5} \leq \frac{27}{80}r_\eta(\sqrt{N_{\mathcal{T}}} - 2)$.

$$\frac{r_\eta}{r_\mu} \leq \frac{24}{25}$$

$$= \frac{10}{25}\left(\frac{27}{5} - 3\right)$$

$$\leq \frac{10}{25}\left(\frac{27}{80}r_\eta(\sqrt{N_{\mathcal{T}}} - 2) - 3\right)$$

This means $\frac{r_\eta}{r_\mu} \leq \min\left(\frac{24}{25}, \frac{4}{N_{\mathcal{T}}}\right)$ which satisfies eq. (10). $\qquad \square$

**Theorem 3.** *(Restating theorem 1)*
*Consider a Linear-Gaussian PU-learning problem with parameters $\boldsymbol{\mu}, \boldsymbol{\eta}, d > 300$, dataset sizes $N_{\mathcal{T}} > 10$, $N = N_{\mathcal{S}} + N_{\mathcal{T}}$, $\alpha \in (0, \frac{N}{1024N_{\mathcal{T}}})$ and let $\delta \in (0,1)$. For all problems where*

$$\frac{16}{\sqrt{N_{\mathcal{T}}}} \leq r_\eta \leq \frac{3}{8}\left(\sqrt{N_{\mathcal{T},1}}\right)^{-1}, \tag{20}$$

$$\frac{1}{\sqrt{N_{\mathcal{T},1}}} \leq r_\mu \leq \frac{1}{2}\sqrt{N_{\mathcal{T},0}}, \tag{21}$$

$$c_1\log\left(\frac{c_2}{\delta}\right) \leq \min\left(\sqrt{N}, \sqrt{\frac{d}{N}}\right) \tag{22}$$

*it holds with probability at least $1 - \delta$ that $\mathrm{AU{-}ROC}(\mathbf{w}_{\mathrm{DD}}) < 0.5$ and $\mathrm{AU{-}ROC}(\mathbf{w}_{\mathrm{color}}) > 0.9$.*

**Theorem 4.** *(Restating theorem 1)*
*Consider a Linear-Gaussian PU-learning problem with parameters $\boldsymbol{\mu}, \boldsymbol{\eta}, d > 300$, dataset sizes $N_{\mathcal{T}} > 10$, $N = N_{\mathcal{S}} + N_{\mathcal{T}}$, $\alpha \in (0, \frac{N}{1024 N_{\mathcal{T}}})$ and let $\delta \in (0, 1)$. For all problems where*

$$\frac{16}{\sqrt{N_{\mathcal{T}}}} \leq \min\left(r_\eta, r_\mu\right), \tag{23}$$

$$r_\mu \leq \tfrac{1}{2} \frac{\sqrt{N_{\mathcal{T},0}}}{N_{\mathcal{T},1}}, \tag{24}$$

$$\frac{r_\eta}{r_\mu} \leq \sqrt{\frac{N_{\mathcal{T},1}}{N_{\mathcal{T},0}}} \tag{25}$$

$$c_1 \log\left(\tfrac{c_2}{\delta}\right) \leq \min\left(\sqrt{N}, \sqrt{\frac{d}{N}}\right) \tag{26}$$

*it holds with probability at least $1 - \delta$ that $\mathrm{AU-ROC}(\mathbf{w}_{\mathrm{DD}}) < 0.5$ and $\mathrm{AU-ROC}(\mathbf{w}_{\mathrm{color}}) > 0.9$.*

### A.3 Concentration statements

If $\xi_i \sim \mathcal{N}(0, \sigma^2 \mathbf{I}_d)$ for all $i \in [N]$, then

- Bound on norm of vector (Ledoux & Talagrand, 2013, Eq. 3.5)

$$P(\|\xi_i\| \leq t) > 1 - 4 \exp\left\{-\frac{t^2}{8 d \sigma^2}\right\}. \tag{27}$$

- For $U \subseteq [N]$ consider $\{\xi_j\}_{j \in U}$ and $\xi_i, i \notin U$,

$$P\left(\left|\langle \xi_i, \frac{1}{|U|} \sum_{j \in U} \xi_j \rangle\right| < |U|^{-1/2} t\right) > 1 - 2 \exp\left[-c \min\{\frac{t^2}{\sigma^4 d}, \frac{t}{4\sigma^2}\}\right] \tag{28}$$

  This is a derivation from Bernstein's inequality, see e.g. Lemma 3 in Puli et al. (2023).

- For $i \in U$ we have,

$$P\left(\|\xi_i\|^2 > \sigma^2(d + 2t\sqrt{d})\right) \leq \exp\left(-\frac{t^2}{4}\right) \tag{29}$$

- If $Z$ is a chi-square variable with d-degrees of freedom, then it holds for any $v > 0$

$$P(Z \leq d - 2\sqrt{dv}) \leq \exp\left(-v\right)$$

  It follows that

$$P(Z \geq d - 2\sqrt{dv}) \geq 1 - \exp\left(-v\right)$$

  Hence, the lower bound on $\|\xi_i\|^2 \geq \sigma^2(d - 2\sqrt{dv})$ w.p. at least $1 - \exp(-v)$.
  Upon setting $t = \sqrt{v}$, we get

$$P\left(\|\xi_i\|^2 \geq \sigma^2(d - 2t\sqrt{d})\right) \geq 1 - \exp\left(-ct^2\right) \tag{30}$$

- Let $u \in [d]$, then

$$\|\xi_{i,u} \cdot \sum_{j \in U \setminus i} \xi_{j,u}\|_{\psi_1} \leq \|\xi_{i,u}\|_{\psi_2} \|\sum_{j \in U \setminus i} \xi_{j,u}\|_{\psi_2} \leq \sqrt{|U|}\sigma^2.$$

  Recall Bernstein (Vershynin, 2018) says that for independent sub-exponential variables $X_1, \dots, X_N$, for every $t > 0$,

$$P\left(\left|\sum_{i=1}^{N} X_i\right| \geq t\right) \leq 2 \exp\left[-c \min\left\{\frac{t^2}{\sum_{i=1}^{N} \|X_i\|_{\psi_1}^2}, \frac{t}{\max_i \|X_i\|_{\psi_1}}\right\}\right]$$

Applying this inequality to our case (where $N = d$ and the $X_i$s are the products of $\xi$ variables on the LHS above),

$$P(|\sum_{j \in U \setminus i} \xi_i^\top \xi_j| \geq t) \leq 2 \exp\left[-c \min\left\{\frac{t^2}{d|U|\sigma^4}, \frac{t}{\sqrt{|U|}\sigma^2}\right\}\right].$$

Now **assume that** $t < d\sqrt{|U|}\sigma^2$, then

$$P(|\sum_{j \in U \setminus i} \xi_i^\top \xi_j| \geq t) \leq 2 \exp\left[-c \min\left\{\frac{t^2}{d|U|\sigma^4}, \frac{t}{\sqrt{|U|}\sigma^2}\right\}\right]$$

$$= 2 \exp\left[-c\frac{t^2}{d|U|\sigma^4}\right].$$

Replacing variables $t$ with $t\sqrt{d|U|\sigma^4}$, and divding both sides of the inequality by $|U|$, we end up with

$$P\left(\left||U|^{-1}\sum_{j \in U \setminus i} \xi_i^\top \xi_j\right| \geq \sigma^2 t\sqrt{d|U|^{-1}}\right) \leq 2 \exp\left[-ct^2\right] \tag{31}$$

- We know that for $\left|\langle\xi_i, \frac{1}{|U|}\sum_{j \in U}\xi_j\rangle\right| = \frac{1}{|U|}\|\xi_i\|^2 + \frac{1}{|U|}\left|\langle\xi_i, \frac{1}{|U|}\sum_{j \in U, i \neq j}\xi_j\rangle\right|$

$$\langle\xi_i^\top \bar{\xi}_U\rangle = |U|^{-1}\|\xi_i\|^2 + \langle\xi_i, |U|^{-1}\sum_{j \in U, j \neq i}\xi_j\rangle$$

$$\left|\langle\xi_i^\top \bar{\xi}_U\rangle\right| \geq \left||U|^{-1}\|\xi_i\|^2\right| - \left|\langle\xi_i, |U|^{-1}\sum_{j \in U, j \neq i}\xi_j\rangle\right|$$

Taking union bound of inequalities 30 and 31, we get

$$P\left(|\langle\xi_i^\top \bar{\xi}_U\rangle| \geq |U|^{-1}\sigma^2(d - 2t\sqrt{d}) - |U|^{-1}\sigma^2 t\sqrt{d(|U|-1)}\right) \geq 1 - 3\exp(-ct^2)$$

$$P\left(|\langle\xi_i^\top \bar{\xi}_U\rangle| \geq |U|^{-1}\sigma^2\left(d - 2t\sqrt{d} - t\sqrt{d(|U|-1)}\right)\right) \geq 1 - 3\exp(-ct^2)$$

If we assume $t \leq \frac{\sqrt{d}}{2(2+\sqrt{|U|-1})}$, we get

$$P\left(|\langle\xi_i^\top \bar{\xi}_U\rangle| \geq \frac{\sigma^2 d}{2|U|}\right) \geq 1 - 3\exp(-ct^2) \tag{32}$$

## A.4 Proof of Lemma 1

Let us recall the strong positivity assumption stated in the main paper, which appears in Garg et al. (2022).

**Assumption 2** (Strong positivity). *There exists $X_{sep} \subseteq \mathcal{X}$ such that $P_{\mathcal{T},1}(X_{sep}) = 0$ and the matrix $[P_{\mathcal{S}}(\mathbf{x} \mid y)]_{\mathbf{x} \in X_{sep} my \in [k]}$ is full rank and diagonal.*

We restate and prove the claim that Open-Set Domain Adaptation is not learnable under this assumption, once the label shift assumption is removed.

**Lemma.** *Let $\mathcal{A}$ be an algorithm for Open-Set Domain Adaptation. There are distributions $P_{\mathcal{S}}, P_{\mathcal{T},[k]}$ and $P_{\mathcal{T},k+1}$ such that the problem satisfies strong positivity, and $\exists h^* \in \mathcal{H}$ for which $R_{\mathcal{T}}^{l_{01}}(h^*) = 0$, while $\mathbb{E}_{S_{\mathcal{S}},S_{\mathcal{T}}}\left[R_{\mathcal{T}}^{l_{01}}(\mathcal{A}(S_{\mathcal{S}}, S_{\mathcal{T}}))\right] \geq 0.5.$*

*Proof.* Define the following distributions over 4 states

$$[P_{\mathcal{S}}(x \mid y)]_{x \in \mathcal{X}, y \in [k]} = \begin{bmatrix} 1 - \varepsilon & 0 & 0 & \varepsilon \\ 0 & 1 - \varepsilon & \varepsilon & 0 \end{bmatrix},$$

$$P_{\mathcal{S}}(Y) = [\frac{1 - 2\varepsilon}{1 - \varepsilon}, \frac{\varepsilon}{1 - \varepsilon}, 0]$$

for some $\varepsilon > 0$, and two other distribution over $\mathcal{X}$, $Q(X) = [0, 0, 1, 0]$ and $D(X) = [0, 0, 0, 1]$. Consider 2 Open-Set Domain Adaptation problems where $k = 2$ and $\alpha = 0.5$:

- One where $P_{\mathcal{T}, [k]} = Q$ and $P_{\mathcal{T}, k+1} = D$, which means that $P_{\mathcal{T}, [k]}(X \mid Y = y) = [0, 0, 1, 0]$ and we set $P_{\mathcal{T}, [k]}(Y) = [\frac{1 - 2\varepsilon}{1 - \varepsilon}, \frac{\varepsilon}{1 - \varepsilon}, 0]$ although we can set it to any arbitrary distribution.

- For the second problem $P_{\mathcal{T}, [k]} = D$ and $P_{\mathcal{T}, k+1} = Q$, which entails similarly to the first case that $P_{\mathcal{T}, [k]}(X \mid Y = y) = [0, 0, 0, 1]$ for $y \in [k]$, while we keep $P_{\mathcal{S}}$ and the rest of the details as they are in the first problem.

It is clear that under the hypothesis class $\mathcal{H}$ of all binary classifiers on $\mathcal{X}$, it holds that $R_{\mathcal{T}}^{l_{01}}(h^*) = 0$. Now we will show that both problems satisfy strong positivity (note that they also satisfy that $\text{Supp}(P_{\mathcal{T}, [k]}) \subseteq \text{Supp}(P_{\mathcal{S}})$), and also $P_{\mathcal{S}}(X, Y)$ and $P_{\mathcal{T}}(X)$ are the same for both problems.

Once this is shown, we can conclude our result, since any observed dataset that is an input to $\mathcal{A}$ is equally likely in both problems. However, any hypothesis $h$ that achieves $R_{\mathcal{T}}^{l_{01}}(h) = \delta$ on the first problem, achieves risk $1 - \delta$ on the other problem since $P_{\mathcal{T}, [k]}$ and $P_{\mathcal{T}, k+1}$ switch roles between the two problems.

To show that the problems satisfy strong positivity, consider $X_{sep}$ as the first and second states. We have that for both problem $P_{\mathcal{T}, k+1}(X_{sep} = 0$ since both $D(X_{sep}) = 0$ and $Q(X_{sep}) = 0$, while

$$[P_{\mathcal{S}}(x \mid y)]_{x \in X_{sep}, y \in [k]} = \begin{bmatrix} 1 - \varepsilon & 0 \\ 0 & 1 - \varepsilon \end{bmatrix},$$

which is a full rank and diagonal matrix. Hence the strong positivity condition is satisfied. We defined the same $P_{\mathcal{S}}(X, Y)$ for both problems, so it is left to show that $P_{\mathcal{T}}(X)$ also equals for them. This is also straightforward as for both problem $P_{\mathcal{T}}(X) = 0.5 \cdot Q + 0.5 \cdot D$, which concludes the proof. □

### A.5 Additional details on experimental setting

### A.5.1 AUROC vs AUPRC scores for novel class detection

It is a common argument in the machine learning literature that AUPRC scores are more suitable for evaluating methodologies in class-imbalanced scenarios. However, this stance is nuanced, as some research, such as McDermott et al. (2024) suggests favoring AUROC over AUPRC in certain imbalanced conditions. McDermott et al. (2024) further notes that AUPRC inherently emphasizes the performance on samples with higher scores. Given our experimental focus on assessing models' ability to assign higher scores for novelty detection, AUPRC emerges as the most relevant metric for our analysis, especially when the proportions of positive class (novel class in our case) is very low.

### A.5.2 Discussion

Building on the results and observations from the previous section, we proceed to further analyze and understand the aspects of OSDA under conditions of background shift. The curve plots in 3b compare the novelty detection performance of the methods shown , it is evident that CoLOR outperforms other methods particularly when the novel class ratio $\alpha$ is less than 0.2. However, as $\alpha$ is increased beyond 0.3, the other baselines rapidly catch up to the AUPRC performance of CoLOR. From Table 4 we observe that using constrained learning to acquire shared representations benefits the classification performance on known classes. The source-only method mentioned in 4 serves as a baseline, trained exclusively on the $S_{\mathcal{S}}$ and evaluated on the $S_{\mathcal{T}}$ without employing any strategies to mitigate shift effects. Furthermore, Table 3 provides insights on the impact of distribution shift on the overall OSDA performance of all the methods using ViT-L/14 visual encoder pretraiend using CLIP.

We specify our empirical test of measuring separability in A.7.1. Based on this test of separability, we observe that novel classes in Amazon Reviews dataset are not perfectly separable and hence violate the assumption of separability to some extent. However, we see in our result in tables 2, 14 and 16 that CoLOR still outperforms other baselines. Hence, we realize that CoLOR is robust to certain violations of the separability assumption.

### A.5.3   Implementation of CoLOR

It is important to note that each model head $h_{\hat{\alpha}}$ independently solve constrained problem in 4 using primal-dual optimization while remaining $k$ heads focus on classifying samples (from $S_{\mathcal{S}}$) from $k$ known categories using ground truths $\mathcal{Y}_{\mathcal{S}}$.

### A.6   Combining CoLOR with Domain Adaptation methods

OSDA performance depends on the robustness of the closed-set classifier. If the classifier were more robust to the specific shift, OSDA performance would improve. Since CoLOR operates on top of the closed-set classifier, any method that improves robustness to a specific shift (e.g., domain adaptation, shift-robust training) could be used together with CoLOR. CoLOR would ensure that the domain shift does not impact the performance of novelty detection while the robust closed-set classifier would aid in improving the performance over known classes. This flexibility enables CoLOR to be used with any architecture and make them robust against distribution shift during test time.

### A.7   Extending constrained learning objective to OSDA

In practice, constraints such as $\beta(h) \leq \beta$ are enforced using a differentiable approximation of a step function, e.g. via a sigmoid $\sum_{\mathbf{x} \in S_{\mathcal{S}}} \sigma(h(\mathbf{x})) \leq \beta$, and objectives are optimized using the logistic loss. We either train an entire model (encoder + classifier) from scratch or just add two fully-connected (FC) layers on top of a pretrained encoder and only train these additional FC layers. The first FC layer provides the shared representation acquired through learning from the related tasks while second FC layer uses this shared representation to classify the known classes and detect novel identities.

### A.7.1   Dataset

One important factor to consider while creating shifts is that $S_{\mathcal{S}}$ and $S_{\mathcal{T},k+1}$ should be distinguishable. As dataset separability is a difficult quantity to measure we train a classifier (oracle) for each dataset to distinguish novel groups from samples belonging to known categories. We then use the learnability of the oracle as a criterion to ensure the separability of novel classes. This means that we calculate the AUROC and AUPRC scores of the oracle for the task of supervised novelty detection. Higher AUROC and AUPRC would correspond to higher separability. We consider an AUROC and AUPRC higher than 0.98 as ideal to ensure separability between novel class identities and known classes. It is difficult to ensure such high separability for all the datasets, particularly for Amazon Reviews dataset which does not perfectly satisfy the separability. Yet we observe that CoLOR is robust to background shift in such settings outperforming all the baselines as observed in Tables 2, 14 and 16.

We conduct 5 repetitions of an experiment for each dataset and for every identity of the novel class. Each repetition uses a unique random seed, representing a distinct background shift setting. These settings are generated by randomly varying the subtype proportions of the known categories within the SUN397 dataset.

**SUN397:** It consists of images of scenes/places from various locations. The dataset is provided with 3 levels of hierarchy where Level-1 is grouped as indoor, natural outdoor and man-made outdoor scenes. We use indoor classes as in-distribution classes while we choose novel classes from natural outdoor scenes. Level-2 hierarchy has shopping, workplace, homes/hotels, etc. under indoor category while outdoor natural contains classes like water/ice/snow, mountains/hills desert/sky, forest, etc. Each of these level-2 categories have subcategories (level-3 classes) that form these level-2 groupings. We randomly select 8 level-3 subtypes (like bakery shop or banquet hall) per level-2 category (shopping/dining places) and vary the subtype proportions to create background shift between source and target. Furthermore, novel classes are randomly selected from the level-2 categories of outdoor natural group.

**CIFAR100:** It consists of 60,000 32x32 colour images in 20 primary classes (superclasses) each of which have 5 subcategories composing a total of 100 classes Krizhevsky et al. (2009). The training set has 50,000 images (i.e. 500 images per subcategory and 2500 images per primary class) while the test set has 10000 images (i.e. 100 images per subcategory and 500 images per primary class). We retain these splits for our

experiments. We use 4 primary categories as aquatic mammals, flowers, fishes and birds. The subcategories of the aquatic mammals are beaver, dolphin, otter, seal are whale while that of flowers are orchid, poppy, rose, sunflower and tulip. Similarly fishes and birds have 5 subcategories each. We vary the marginal distribution of these subcategories to create a subpopulation shift leading to a background shift between source and target data while maintaining no label shift w.r.t. primary categories i.e. aquatic mammals and flowers. The novel category is randomly selected from the remaining unseen categories.

**Amazon Reviews** The dataset is heavily skewed with respect to sentiments and product categories. Hence, we select 6 product categories having similar orders of magnitude of the number of reviews namely 'Digital Music', 'Industrial & Scientific', 'Luxury Beauty', 'Musical Intstruments', 'Prime Pantry' and 'Software'. To prevent further skewness in the dataset due to sentiments, we restrict the sample size per sentiment per category to 500 reviews in the training set and 125 reviews per sentiment per product category. We induce a background shift based on sentiments. Reviews with rating strictly below 3.0 (out of 5.0) are considered negative sentiments and those with rating strictly above 3.0 are considered positive sentiments whereas reviews having a rating of exactly 3.0 are discarded from the dataset. The minimum rating is 1.0 while the maximum is 5.0.

### A.7.2 Hyper-Parameters and Training

We consistently set the FPR threshold $\beta = 0.01$ without optimizing it at all based on validation dataset. For each novel class identity, we repeat the experiments for 5 different randomly generated splits between $P_{\mathcal{S}}$ and $P_{\mathcal{T}}$ adhering to the definition and assumptions of background shift. For CIFAR100, we use ResNet18 backbone followed by a linear layer for classification and train the whole model from scratch for all the methods. For Amazon Reviews dataset, we use pretrained RoBERTa features followed by 2 linear layers for classification. For Amazon Reviews, due to computational limitations we resort to linear probing rather than finetuning the whole model and only finetune the last two linear layers. The primary hyperparameters we tune for stable convergence are learning rate, L2 weight penalty scalar to prevent overfitting, logit multiplier values that act as temperature controllers for softmax/sigmoid scores and gradient clipping to avoid exploding gradients. These hyper parameters are tuned based on a sample training and validation sets but are kept constant throughout the dataset and baseline across different seed values and novelty cases. Furthermore, the methods CoNoC & CoLOR require additional hyperparameters like dual learning rate and lagrange multipliers. Each output node is associated with Lagrange multipliers, which address a distinct primal-dual optimization problem owing to diverse target recall constraints. These multipliers are initialized to 1.0, while the dual learning rate is meticulously calibrated for CIFAR100 & Amazon Reviews datasets individually to ensure stable learning dynamics conducive to minimizing both the objective surrogate loss function for FPR and recall inequality constraints. Note that for ZOC, We used cosine similarity between image embeddings and the known class text embeddings to obtain the closed-set class predictions. Table 7 displays all the hyperparameters used for each of the baselines and datasets.

### A.7.3 Impact of search grid range and density on the performance

For all the methods, we keep the search grid of candidate target recall values consistent through all the experiments i.e. $\boldsymbol{\alpha} = [0.02, 0.05, 0.10, 0.15, 0.20, 0.25, 0.30, 0.35, 0.40, 0.45]$.

**Choice of the search grid range:**

$\boldsymbol{\alpha} = [0.02, 0.45]$ is not claimed to be universal, but chosen to study the most informative regime. We observed that when the size of the novel class (true size, not the one estimated by the model) is below 1%, all baselines failed. Accordingly, we set the lower bound of our search grid on that scale, but we show in experiments that including lower values does not change performance by much. As for the upper bound, we observe that our method is relatively robust when the true novel class size is small (i.e. the model wouldn't choose a larger alpha if it had the possibility). We cap the upper bound of the gris at 0.45 because when the novel class constitutes a large fraction of the target data, novelty detection becomes relatively easier. In such regimes, even simple inspection of random data points or sampling strategies can easily detect novel instances, enabling labeling and the use of alternative solutions. Our primary focus is therefore on the more challenging and practically relevant regime where novel classes are relatively rare (typically between 0.02 and 0.2). We further confirm this through our ablation study in table 22. We note that this lower bound is not fundamental and may depend on factors such as dataset characteristics, feature representations, and model architecture. In practice, we recommend determining an appropriate $\boldsymbol{\alpha}$ range via simulation: known classes can be temporarily treated as unknown, and their detectability can be evaluated under varying assumed

novel class ratios. This procedure provides empirical guidance for both the bottom value and density of the grid range.

**Choice of search grid density:** We added experiments on the SUN397 dataset using a ResNet50 backbone pretrained on ImageNet-1K (Table 19 and Figure 5 in the revised draft). These results show that CoLOR is performant for a fixed $\alpha$ range with only minor performance variations across varying grid densities. This indicates that the method does not require finely tuned grids to achieve strong performance, and that the chosen density is sufficient to cover the relevant operating regime.

Table 7: Hyperparameters: lr = learning rate, dlr = dual learning rate (CoLOR), L2 penalty = L2 weight penalty scaler, lm = logit multiplier, clip = gradient clipping value. The two values separated by "/" in learning rate column of SUN397 dataset correspond to the linear probing of ResNet50 (pretrained on ImageNet) and ViT (pretrained using CLIP) backbones respectively.

| Method | CIFAR100 | | | | | Amazon Reviews | | | | | SUN397 | | | | |
|---|---|---|---|---|---|---|---|---|---|---|---|---|---|---|---|
| | lr | dlr | L2 penalty | lm | clip | lr | dlr | L2 penalty | lm | clip | lr | dlr | L2 penalty | lm | clip |
| DD | $1e-2$ | $-$ | $3e-5$ | 1.2 | 5.0 | $1e-2$ | $-$ | $1e-4$ | 1.0 | 1.0 | $1e-2/1e-1$ | $-$ | $3e-5$ | 1.2 | 5.0 |
| uPU | $1e-3$ | $-$ | $3e-7$ | 1.2 | 5.0 | $1e-3$ | $-$ | $1e-4$ | 1.0 | 1.0 | $1e-3/1e-1$ | $-$ | $3e-7$ | 1.2 | 5.0 |
| nnPU | $1e-3$ | $-$ | $3e-7$ | 1.2 | 5.0 | $1e-3$ | $-$ | $1e-4$ | 1.0 | 1.0 | $1e-3/1e-1$ | $-$ | $3e-7$ | 1.2 | 5.0 |
| BODA | $1e-3$ | $-$ | $3e-3$ | 1.2 | 5.0 | $1e-3$ | $-$ | $1e-4$ | 1.0 | 1.0 | $1e-3$ | $-$ | $3e-3$ | 1.2 | 5.0 |
| ARPL | $1e-2$ | $-$ | $3e-5$ | 1.0 | 100.0 | $-$ | $-$ | $-$ | $-$ | $-$ | $1e-2$ | $-$ | $3e-5$ | 1.0 | 100.0 |
| PULSE | $1e-3$ | $-$ | $3e-5$ | 1.2 | 5.0 | $1e-3$ | $-$ | $1e-4$ | 1.0 | 1.0 | $1e-3$ | $-$ | $3e-5$ | 1.2 | 5.0 |
| CoLOR | $1e-3$ | $2e-2$ | $3e-7$ | 1.2 | 5.0 | $1e-3$ | $6e-2$ | $1e-4$ | 1.0 | 1.0 | $1e-3$ | $2e-2$ | $3e-7$ | 1.2 | 5.0 |

## A.8 Performance comparison based on average relative & absolute AU-ROC and AU-PRC scores

Refer to tables 12, 14, 8, 10, 11, 3, 4 below.

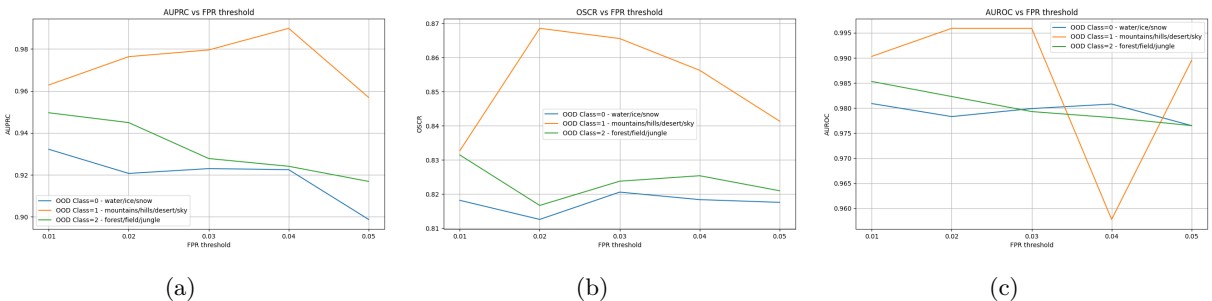

Figure 4: Effects of varying the FPR threshold on CoLOR method performance on SUN397 dataset.

## A.9 Analysis of toy example

We have $N_{\mathcal{S}} = |S_{\mathcal{S}}|$, $N_{\mathcal{T},0} = (1-\alpha)|S_{\mathcal{T}}|$, $N_{\mathcal{T},1} = \alpha|S_{\mathcal{T}}|$ for examples from $P_{\mathcal{S}} = \mathcal{N}(\mu, \sigma(I_d - r_\eta^{-2}\eta\eta^\top))$, $P_{\mathcal{T},0} = \mathcal{N}(-\mu, \sigma(I_d - r_\eta^{-2}\eta\eta^\top)$ and $P_{\mathcal{T},1} = \mathcal{N}(\eta, \sigma(I_d - r_\mu^{-2}\mu\mu^\top))$ respectively. Here, $r_\mu = \|\mu\|$ and $r_\eta$ respectively, we set $\eta^\top\mu = 0$ and let the novelties and non-novel data lie in orthogonal subspaces to each other. This is for simplicity of analysis, and to mimic the case where at optimality we can perfectly detect the novelties.

Consider a linear model $\mathbf{w} \in \mathbb{R}^d$, i.e. its bianry predictions are $\mathbf{1}_{\mathbf{w}^\top\mathbf{x}>0}$ and the scores are $\mathbf{w}^\top\mathbf{x}$. We are interested in the AU-ROC obtained w.r.t novelty detection, i.e. distinguishing $P_{\mathcal{T},0}$ and $P_{\mathcal{T},1}$. This is measured by $Q(\frac{\langle\mathbf{w},2\mu-\eta\rangle}{\|\mathbf{w}\|})$ where $Q$ is the Gaussian tail function.

We will compare two learning rules. Taking $\ell$ as the logistic loss, we are interested in:

$$\min_{w\in\mathbb{R}^d} \sum_{\mathbf{x}_i\in S_{\mathcal{S}}} \ell(w^\top x_i, 0) + \sum_{\mathbf{x}_i\in S_{\mathcal{T}}} \ell(w^\top\tilde{x}_i, 1),$$

and

$$\min_{w\in\mathbb{R}^d} \sum_{\mathbf{x}_i\in S_{\mathcal{T}}} \ell(w^\top\tilde{x}_i, 1) \text{ s.t. } w^\top x_i < 0 \ \forall\mathbf{x}_i \in S_{\mathcal{S}}$$

Table 8: SUN397 dataset with distribution shift due to varying proportions of subtypes of scenes/places. All the methods here use ResNet50 backbone pretrained on ImageNet1K_V1 Russakovsky et al. (2015). AUROC and AUPRC represent the performance for the novel category detection task while OSCR measures overall performance of the methods on both known and unknown classes. $\alpha$ is the mixture proportion column for the respective novel classes.

| Metric | Method | Novel Classes (natural outdoor scenes/places) | | | |
| | | $\alpha = 0.06 \pm 0.01$ | $\alpha = 0.06 \pm 0.01$ | $\alpha = 0.08 \pm 0.04$ | |
| | | [water, ice, snow, etc.] | [mountains, hills, desert, sky, etc.] | [forest, field, jungle, etc.] | Summary |
|---|---|---|---|---|---|
| AUROC | DD | $0.90 \pm 0.06$ | $0.88 \pm 0.05$ | $0.94 \pm 0.02$ | $0.91 \pm 0.05$ |
| | uPU | $0.71 \pm 0.13$ | $0.72 \pm 0.18$ | $0.84 \pm 0.08$ | $0.76 \pm 0.14$ |
| | nnPU | $0.71 \pm 0.13$ | $0.72 \pm 0.18$ | $0.84 \pm 0.08$ | $0.76 \pm 0.14$ |
| | BODA | $0.86 \pm 0.03$ | $0.90 \pm 0.01$ | $0.82 \pm 0.09$ | $0.86 \pm 0.06$ |
| | SHOT | $0.77 \pm 0.08$ | $0.64 \pm 0.21$ | $0.70 \pm 0.18$ | $0.71 \pm 0.16$ |
| | ARPL | $0.73 \pm 0.03$ | $0.72 \pm 0.08$ | $0.68 \pm 0.12$ | $0.71 \pm 0.08$ |
| | ANNA | $0.92 \pm 0.03$ | $0.96 \pm 0.02$ | $0.92 \pm 0.08$ | $0.93 \pm 0.05$ |
| | CAC | $0.79 \pm 0.05$ | $0.82 \pm 0.05$ | $0.78 \pm 0.04$ | $0.80 \pm 0.05$ |
| | PULSE | $0.75 \pm 0.05$ | $0.76 \pm 0.05$ | $0.69 \pm 0.10$ | $0.73 \pm 0.07$ |
| | CoLOR | $\mathbf{0.98 \pm 0.02}$ | $\mathbf{0.98 \pm 0.01}$ | $\mathbf{0.98 \pm 0.02}$ | $\mathbf{0.98 \pm 0.02}$ |
| AUPRC | DD | $0.50 \pm 0.20$ | $0.40 \pm 0.23$ | $0.73 \pm 0.10$ | $0.54 \pm 0.22$ |
| | uPU | $0.11 \pm 0.04$ | $0.18 \pm 0.20$ | $0.34 \pm 0.30$ | $0.21 \pm 0.22$ |
| | nnPU | $0.11 \pm 0.04$ | $0.18 \pm 0.20$ | $0.34 \pm 0.30$ | $0.21 \pm 0.22$ |
| | BODA | $0.41 \pm 0.04$ | $0.52 \pm 0.06$ | $0.43 \pm 0.16$ | $0.45 \pm 0.11$ |
| | SHOT | $0.19 \pm 0.08$ | $0.17 \pm 0.19$ | $0.21 \pm 0.14$ | $0.19 \pm 0.13$ |
| | ARPL | $0.11 \pm 0.02$ | $0.11 \pm 0.04$ | $0.15 \pm 0.11$ | $0.12 \pm 0.07$ |
| | ANNA | $0.71 \pm 0.13$ | $0.77 \pm 0.13$ | $0.72 \pm 0.23$ | $0.73 \pm 0.16$ |
| | CAC | $0.17 \pm 0.05$ | $0.17 \pm 0.06$ | $0.20 \pm 0.10$ | $0.18 \pm 0.07$ |
| | PULSE | $0.15 \pm 0.05$ | $0.14 \pm 0.06$ | $0.15 \pm 0.06$ | $0.15 \pm 0.05$ |
| | CoLOR | $\mathbf{0.92 \pm 0.04}$ | $\mathbf{0.93 \pm 0.05}$ | $\mathbf{0.89 \pm 0.15}$ | $\mathbf{0.91 \pm 0.09}$ |
| OSCR | DD | $0.66 \pm 0.04$ | $0.67 \pm 0.06$ | $0.70 \pm 0.03$ | $0.68 \pm 0.05$ |
| | uPU | $0.35 \pm 0.10$ | $0.35 \pm 0.08$ | $0.48 \pm 0.11$ | $0.40 \pm 0.11$ |
| | nnPU | $0.35 \pm 0.10$ | $0.35 \pm 0.08$ | $0.48 \pm 0.11$ | $0.40 \pm 0.11$ |
| | BODA | $0.56 \pm 0.10$ | $0.59 \pm 0.09$ | $0.50 \pm 0.09$ | $0.55 \pm 0.10$ |
| | SHOT | $0.25 \pm 0.05$ | $0.18 \pm 0.07$ | $0.24 \pm 0.08$ | $0.22 \pm 0.07$ |
| | ARPL | $0.61 \pm 0.01$ | $0.61 \pm 0.06$ | $0.58 \pm 0.08$ | $0.60 \pm 0.06$ |
| | ANNA | $0.58 \pm 0.07$ | $0.62 \pm 0.05$ | $0.59 \pm 0.09$ | $0.60 \pm 0.07$ |
| | CAC | $0.68 \pm 0.05$ | $0.70 \pm 0.07$ | $0.66 \pm 0.03$ | $0.68 \pm 0.05$ |
| | PULSE | $0.66 \pm 0.05$ | $0.68 \pm 0.05$ | $0.62 \pm 0.09$ | $0.65 \pm 0.06$ |
| | CoLOR | $\mathbf{0.82 \pm 0.03}$ | $\mathbf{0.81 \pm 0.03}$ | $\mathbf{0.81 \pm 0.05}$ | $\mathbf{0.81 \pm 0.04}$ |

## A.10 Synthetic Experiment

To study the impact of separability between novel class and known classes, we conduct a synthetic experiment following the theoretical setup defined in definition 2 of the main draft as it allows us to explicitly control separability. In this study, we consider a simple Gaussian mixture setup and vary the angular separation between known and novel class means as well as the absolute distance of the novel class mean $(r_\eta)$ to gradually vary the novel class separability. This enables us to directly examine how CoLOR's performance changes as the separability assumption is increasingly violated. Please refer to fig. 6, table 25 and table 26 for results and further details.

```
\usepackage[accepted]{tmlr}.
```

Table 9: SUN397 dataset without distribution shift due to varying proportions of subtypes of scenes/places. All the methods here use ResNet50 backbone pretrained on ImageNet1K_V1 Russakovsky et al. (2015). AUROC and AUPRC represent the performance for the novel category detection task while OSCR measures overall performance of the methods on both known and unknown classes. $\alpha$ is the mixture proportion column for the respective novel classes.

| Metric | Method | Novel Classes (natural outdoor scenes/places) | | | |
|---|---|---|---|---|---|
| | | $\alpha = 0.06 \pm 0.01$ | $\alpha = 0.06 \pm 0.01$ | $\alpha = 0.08 \pm 0.04$ | |
| | | [water, ice, snow, etc.] | [mountains, hills, desert, sky, etc.] | [forest, field, jungle, etc.] | Summary |
| AUROC | DD | **1.00 ± 0.00** | **1.00 ± 0.00** | **1.00 ± 0.00** | **1.00 ± 0.00** |
| | uPU | 0.99 ± 0.00 | 0.99 ± 0.01 | 1.00 ± 0.00 | 0.99 ± 0.01 |
| | nnPU | 0.99 ± 0.00 | 0.99 ± 0.01 | 1.00 ± 0.00 | 0.99 ± 0.01 |
| | BODA | 0.86 ± 0.06 | 0.91 ± 0.01 | 0.85 ± 0.05 | 0.87 ± 0.05 |
| | SHOT | 0.63 ± 0.08 | 0.58 ± 0.12 | 0.61 ± 0.14 | 0.61 ± 0.11 |
| | ARPL | 0.86 ± 0.03 | 0.81 ± 0.03 | 0.84 ± 0.05 | 0.84 ± 0.04 |
| | ANNA | 0.95 ± 0.05 | 0.98 ± 0.02 | 0.92 ± 0.11 | 0.95 ± 0.07 |
| | CAC | 0.89 ± 0.05 | 0.89 ± 0.01 | 0.87 ± 0.03 | 0.88 ± 0.03 |
| | PULSE | 0.82 ± 0.08 | 0.84 ± 0.02 | 0.79 ± 0.04 | 0.82 ± 0.05 |
| | CoLOR | **1.00 ± 0.00** | **1.00 ± 0.00** | **1.00 ± 0.00** | **1.00 ± 0.00** |
| AUPRC | DD | **1.00 ± 0.00** | **1.00 ± 0.00** | 0.99 ± 0.01 | **1.00 ± 0.00** |
| | uPU | 0.95 ± 0.05 | 0.90 ± 0.13 | 0.98 ± 0.02 | 0.94 ± 0.09 |
| | nnPU | 0.95 ± 0.05 | 0.90 ± 0.13 | 0.98 ± 0.02 | 0.94 ± 0.09 |
| | BODA | 0.40 ± 0.11 | 0.52 ± 0.02 | 0.40 ± 0.06 | 0.44 ± 0.09 |
| | SHOT | 0.09 ± 0.02 | 0.09 ± 0.06 | 0.11 ± 0.06 | 0.10 ± 0.05 |
| | ARPL | 0.20 ± 0.04 | 0.16 ± 0.04 | 0.25 ± 0.09 | 0.20 ± 0.07 |
| | ANNA | 0.90 ± 0.05 | 0.94 ± 0.03 | 0.86 ± 0.13 | 0.90 ± 0.08 |
| | CAC | 0.29 ± 0.08 | 0.27 ± 0.06 | 0.30 ± 0.08 | 0.29 ± 0.07 |
| | PULSE | 0.21 ± 0.07 | 0.21 ± 0.05 | 0.19 ± 0.03 | 0.20 ± 0.05 |
| | CoLOR | 0.96 ± 0.03 | **1.00 ± 0.00** | **1.00 ± 0.00** | 0.99 ± 0.02 |
| OSCR | DD | 0.86 ± 0.00 | 0.86 ± 0.00 | 0.86 ± 0.00 | 0.86 ± 0.00 |
| | uPU | 0.72 ± 0.01 | 0.72 ± 0.01 | 0.72 ± 0.01 | 0.72 ± 0.01 |
| | nnPU | 0.72 ± 0.01 | 0.72 ± 0.01 | 0.72 ± 0.01 | 0.72 ± 0.01 |
| | BODA | 0.71 ± 0.03 | 0.70 ± 0.01 | 0.61 ± 0.05 | 0.67 ± 0.06 |
| | SHOT | 0.19 ± 0.05 | 0.16 ± 0.04 | 0.17 ± 0.08 | 0.18 ± 0.06 |
| | ARPL | 0.82 ± 0.02 | 0.78 ± 0.03 | 0.80 ± 0.05 | 0.80 ± 0.04 |
| | ANNA | 0.82 ± 0.05 | 0.85 ± 0.02 | 0.79 ± 0.10 | 0.82 ± 0.07 |
| | CAC | 0.84 ± 0.05 | 0.85 ± 0.01 | 0.83 ± 0.03 | 0.84 ± 0.03 |
| | PULSE | 0.78 ± 0.08 | 0.81 ± 0.02 | 0.76 ± 0.03 | 0.78 ± 0.05 |
| | CoLOR | **0.92 ± 0.03** | **0.93 ± 0.01** | **0.92 ± 0.02** | **0.92 ± 0.02** |

Table 10: SUN397 dataset with distribution shift due to varying proportions of subtypes of scenes/places. All the principled methods (DD, uPU, nnPU, BODA & CoLOR) here use pretrained CLIP ViT-L/14 backbone from Radford et al. (2021).

| Metric | Method | Novel Classes (natural outdoor scenes/places) | | | |
| --- | --- | --- | --- | --- | --- |
| | | $\alpha = 0.06 \pm 0.01$ | $\alpha = 0.06 \pm 0.01$ | $\alpha = 0.08 \pm 0.04$ | |
| | | [water, ice, snow, etc.] | [mountains, hills, desert, sky, etc.] | [forest, field, jungle, etc.] | Summary |
| AUROC | DD | $0.96 \pm 0.03$ | $0.96 \pm 0.03$ | $0.96 \pm 0.02$ | $0.96 \pm 0.02$ |
| | uPU | $0.95 \pm 0.03$ | $0.94 \pm 0.05$ | $0.95 \pm 0.04$ | $0.95 \pm 0.04$ |
| | nnPU | $0.95 \pm 0.03$ | $0.94 \pm 0.05$ | $0.95 \pm 0.04$ | $0.95 \pm 0.04$ |
| | BODA | $0.91 \pm 0.06$ | $0.82 \pm 0.18$ | $0.79 \pm 0.19$ | $0.84 \pm 0.15$ |
| | ZOC | $0.84 \pm 0.02$ | $0.85 \pm 0.03$ | $0.78 \pm 0.12$ | $0.82 \pm 0.07$ |
| | CAC | $0.98 \pm 0.02$ | $0.99 \pm 0.01$ | $0.98 \pm 0.01$ | $0.98 \pm 0.01$ |
| | PULSE | $0.96 \pm 0.02$ | $0.96 \pm 0.01$ | $0.95 \pm 0.01$ | $0.96 \pm 0.01$ |
| | CoLOR | $\mathbf{0.99 \pm 0.02}$ | $\mathbf{0.99 \pm 0.01}$ | $\mathbf{0.99 \pm 0.01}$ | $\mathbf{0.99 \pm 0.01}$ |
| AUPRC | DD | $0.88 \pm 0.05$ | $0.85 \pm 0.12$ | $0.90 \pm 0.07$ | $0.87 \pm 0.08$ |
| | uPU | $0.84 \pm 0.05$ | $0.78 \pm 0.19$ | $0.85 \pm 0.10$ | $0.82 \pm 0.12$ |
| | nnPU | $0.84 \pm 0.05$ | $0.78 \pm 0.19$ | $0.85 \pm 0.10$ | $0.82 \pm 0.12$ |
| | BODA | $0.39 \pm 0.23$ | $0.37 \pm 0.39$ | $0.38 \pm 0.30$ | $0.38 \pm 0.29$ |
| | ZOC | $0.21 \pm 0.05$ | $0.23 \pm 0.04$ | $0.26 \pm 0.08$ | $0.23 \pm 0.06$ |
| | CAC | $0.77 \pm 0.12$ | $0.82 \pm 0.10$ | $0.78 \pm 0.08$ | $0.79 \pm 0.10$ |
| | PULSE | $0.6 \pm 0.09$ | $0.61 \pm 0.11$ | $0.60 \pm 0.09$ | $0.60 \pm 0.09$ |
| | CoLOR | $\mathbf{0.95 \pm 0.04}$ | $\mathbf{0.95 \pm 0.01}$ | $\mathbf{0.96 \pm 0.03}$ | $\mathbf{0.95 \pm 0.03}$ |
| OSCR | DD | $0.93 \pm 0.02$ | $0.93 \pm 0.03$ | $0.93 \pm 0.02$ | $0.93 \pm 0.02$ |
| | uPU | $0.92 \pm 0.02$ | $0.91 \pm 0.05$ | $0.91 \pm 0.04$ | $0.92 \pm 0.03$ |
| | nnPU | $0.92 \pm 0.02$ | $0.91 \pm 0.05$ | $0.91 \pm 0.04$ | $0.92 \pm 0.03$ |
| | BODA | $0.82 \pm 0.04$ | $0.61 \pm 0.34$ | $0.73 \pm 0.14$ | $0.72 \pm 0.21$ |
| | ZOC | $0.53 \pm 0.03$ | $0.52 \pm 0.05$ | $0.47 \pm 0.08$ | $0.51 \pm 0.06$ |
| | CAC | $0.96 \pm 0.02$ | $0.96 \pm 0.01$ | $0.95 \pm 0.02$ | $0.96 \pm 0.02$ |
| | PULSE | $0.94 \pm 0.01$ | $0.95 \pm 0.02$ | $0.94 \pm 0.02$ | $0.94 \pm 0.02$ |
| | CoLOR | $\mathbf{0.96 \pm 0.02}$ | $\mathbf{0.97 \pm 0.01}$ | $\mathbf{0.96 \pm 0.01}$ | $\mathbf{0.96 \pm 0.01}$ |

Table 11: SUN397 dataset **without** any intended distribution shift. All the principled methods (DD, uPU, nnPU, BODA & CoLOR) here use pretrained CLIP ViT-L/14 backbone from Radford et al. (2021).

| Metric | Method | Novel Classes (natural outdoor scenes/places) | | | |
|---|---|---|---|---|---|
| | | $\alpha = 0.06 \pm 0.00$ | $\alpha = 0.05 \pm 0.01$ | $\alpha = 0.07 \pm 0.03$ | |
| | | [water, ice, snow, etc.] | [mountains, hills, desert, sky, etc.] | [forest, field, jungle, etc.] | Summary |
| AUROC | DD | $\mathbf{1.00 \pm 0.00}$ | $\mathbf{1.00 \pm 0.00}$ | $\mathbf{1.00 \pm 0.00}$ | $\mathbf{1.00 \pm 0.00}$ |
| | uPU | $\mathbf{1.00 \pm 0.00}$ | $\mathbf{1.00 \pm 0.00}$ | $\mathbf{1.00 \pm 0.00}$ | $\mathbf{1.00 \pm 0.00}$ |
| | nnPU | $\mathbf{1.00 \pm 0.00}$ | $\mathbf{1.00 \pm 0.00}$ | $\mathbf{1.00 \pm 0.00}$ | $\mathbf{1.00 \pm 0.00}$ |
| | BODA | $0.87 \pm 0.03$ | $0.88 \pm 0.04$ | $0.90 \pm 0.05$ | $0.88 \pm 0.04$ |
| | ARPL | $0.86 \pm 0.03$ | $0.81 \pm 0.03$ | $0.84 \pm 0.05$ | $0.84 \pm 0.04$ |
| | ZOC | $0.84 \pm 0.02$ | $0.86 \pm 0.02$ | $0.77 \pm 0.12$ | $0.82 \pm 0.08$ |
| | CAC | $0.99 \pm 0.01$ | $0.99 \pm 0.00$ | $0.99 \pm 0.01$ | $0.99 \pm 0.01$ |
| | PULSE | $0.98 \pm 0.01$ | $0.98 \pm 0.01$ | $0.97 \pm 0.02$ | $0.98 \pm 0.01$ |
| | CoLOR | $\mathbf{1.00 \pm 0.00}$ | $\mathbf{1.00 \pm 0.00}$ | $\mathbf{1.00 \pm 0.00}$ | $\mathbf{1.00 \pm 0.00}$ |
| AUPRC | DD | $\mathbf{1.00 \pm 0.00}$ | $\mathbf{1.00 \pm 0.00}$ | $\mathbf{1.00 \pm 0.00}$ | $\mathbf{1.00 \pm 0.00}$ |
| | uPU | $\mathbf{1.00 \pm 0.00}$ | $\mathbf{1.00 \pm 0.00}$ | $\mathbf{1.00 \pm 0.00}$ | $\mathbf{1.00 \pm 0.00}$ |
| | nnPU | $\mathbf{1.00 \pm 0.00}$ | $\mathbf{1.00 \pm 0.00}$ | $\mathbf{1.00 \pm 0.00}$ | $\mathbf{1.00 \pm 0.00}$ |
| | BODA | $0.18 \pm 0.04$ | $0.20 \pm 0.06$ | $0.34 \pm 0.22$ | $0.24 \pm 0.15$ |
| | ARPL | $0.20 \pm 0.04$ | $0.16 \pm 0.04$ | $0.25 \pm 0.09$ | $0.20 \pm 0.07$ |
| | ZOC | $0.19 \pm 0.02$ | $0.21 \pm 0.04$ | $0.25 \pm 0.11$ | $0.22 \pm 0.07$ |
| | CAC | $0.87 \pm 0.10$ | $0.91 \pm 0.03$ | $0.86 \pm 0.08$ | $0.88 \pm 0.08$ |
| | PULSE | $0.76 \pm 0.08$ | $0.76 \pm 0.08$ | $0.74 \pm 0.11$ | $0.75 \pm 0.09$ |
| | CoLOR | $0.99 \pm 0.01$ | $0.99 \pm 0.01$ | $0.99 \pm 0.01$ | $0.99 \pm 0.01$ |
| OSCR | DD | $\mathbf{0.99 \pm 0.00}$ | $\mathbf{0.99 \pm 0.00}$ | $\mathbf{0.99 \pm 0.00}$ | $\mathbf{0.99 \pm 0.00}$ |
| | uPU | $\mathbf{0.99 \pm 0.00}$ | $\mathbf{0.99 \pm 0.00}$ | $\mathbf{0.99 \pm 0.00}$ | $\mathbf{0.99 \pm 0.00}$ |
| | nnPU | $\mathbf{0.99 \pm 0.00}$ | $\mathbf{0.99 \pm 0.00}$ | $\mathbf{0.99 \pm 0.00}$ | $\mathbf{0.99 \pm 0.00}$ |
| | BODA | $0.94 \pm 0.01$ | $0.94 \pm 0.01$ | $0.92 \pm 0.02$ | $0.93 \pm 0.02$ |
| | ARPL | $0.82 \pm 0.02$ | $0.78 \pm 0.03$ | $0.80 \pm 0.05$ | $0.80 \pm 0.04$ |
| | ZOC | $0.50 \pm 0.01$ | $0.51 \pm 0.02$ | $0.45 \pm 0.07$ | $0.49 \pm 0.05$ |
| | CAC | $0.98 \pm 0.01$ | $0.99 \pm 0.00$ | $0.98 \pm 0.01$ | $0.98 \pm 0.01$ |
| | PULSE | $0.98 \pm 0.01$ | $0.98 \pm 0.01$ | $0.97 \pm 0.02$ | $0.97 \pm 0.02$ |
| | CoLOR | $0.98 \pm 0.00$ | $0.98 \pm 0.00$ | $0.98 \pm 0.00$ | $0.98 \pm 0.00$ |

Table 12: CIFAR100 dataset with background shift due to varying proportions of subtypes. (AUROC) and (AUPRC) represent the performance for the task of novel class detection while Open-Set Classification Rate (OSCR) measures overall performance of the methods on both known and unknown classes. $\alpha$ is in the range of 0.05 to 0.10. *Note that all the adaptive methods use ResNet18 and are trained from scratch.*

| Metric | Method | Novel Classes ($\alpha = 0.07 \pm 0.02$) | | | | | |
|--------|--------|------|------|-----------|--------|----------|---------|
| | | Baby | Man | Butterfly | Rocket | Streetcar | Summary |
| AUROC | DD | $0.76 \pm 0.03$ | $0.61 \pm 0.13$ | $0.64 \pm 0.10$ | $0.78 \pm 0.09$ | $0.72 \pm 0.07$ | $0.70 \pm 0.11$ |
| | uPU | $0.71 \pm 0.07$ | $0.62 \pm 0.09$ | $0.56 \pm 0.13$ | $0.74 \pm 0.06$ | $0.73 \pm 0.05$ | $0.67 \pm 0.10$ |
| | nnPU | $0.69 \pm 0.09$ | $0.58 \pm 0.10$ | $0.60 \pm 0.09$ | $0.74 \pm 0.06$ | $0.72 \pm 0.16$ | $0.67 \pm 0.12$ |
| | BODA | $0.57 \pm 0.07$ | $0.54 \pm 0.05$ | $0.59 \pm 0.03$ | $0.54 \pm 0.07$ | $0.61 \pm 0.05$ | $0.57 \pm 0.06$ |
| | ARPL | $0.79 \pm 0.05$ | $0.80 \pm 0.03$ | $0.73 \pm 0.05$ | $0.73 \pm 0.04$ | $0.73 \pm 0.05$ | $0.76 \pm 0.05$ |
| | ZOC | $\mathbf{0.97 \pm 0.01}$ | $\mathbf{0.98 \pm 0.01}$ | $\mathbf{0.82 \pm 0.03}$ | $\mathbf{0.98 \pm 0.01}$ | $\mathbf{1.00 \pm 0.00}$ | $\mathbf{0.95 \pm 0.07}$ |
| | CAC | $0.71 \pm 0.05$ | $0.72 \pm 0.04$ | $0.71 \pm 0.02$ | $0.64 \pm 0.03$ | $0.68 \pm 0.02$ | $0.69 \pm 0.04$ |
| | PULSE | $0.72 \pm 0.04$ | $0.74 \pm 0.03$ | $0.70 \pm 0.02$ | $0.70 \pm 0.02$ | $0.72 \pm 0.06$ | $0.72 \pm 0.04$ |
| | CoLOR | $0.77 \pm 0.06$ | $0.70 \pm 0.12$ | $0.68 \pm 0.04$ | $0.83 \pm 0.04$ | $0.85 \pm 0.03$ | $0.77 \pm 0.09$ |
| AUPRC | DD | $0.25 \pm 0.03$ | $0.15 \pm 0.10$ | $0.15 \pm 0.09$ | $0.37 \pm 0.12$ | $0.27 \pm 0.17$ | $0.24 \pm 0.13$ |
| | uPU | $0.22 \pm 0.18$ | $0.11 \pm 0.03$ | $0.11 \pm 0.05$ | $0.28 \pm 0.14$ | $0.18 \pm 0.10$ | $0.18 \pm 0.12$ |
| | nnPU | $0.21 \pm 0.16$ | $0.09 \pm 0.03$ | $0.11 \pm 0.04$ | $0.32 \pm 0.09$ | $0.27 \pm 0.25$ | $0.20 \pm 0.16$ |
| | BODA | $0.10 \pm 0.05$ | $0.08 \pm 0.02$ | $0.09 \pm 0.02$ | $0.08 \pm 0.02$ | $0.09 \pm 0.03$ | $0.09 \pm 0.03$ |
| | ARPL | $0.22 \pm 0.07$ | $0.23 \pm 0.06$ | $0.16 \pm 0.05$ | $0.15 \pm 0.05$ | $0.15 \pm 0.06$ | $0.18 \pm 0.07$ |
| | ZOC | $\mathbf{0.89 \pm 0.04}$ | $\mathbf{0.76 \pm 0.05}$ | $\mathbf{0.32 \pm 0.07}$ | $\mathbf{0.77 \pm 0.05}$ | $\mathbf{0.96 \pm 0.02}$ | $\mathbf{0.74 \pm 0.23}$ |
| | CAC | $0.18 \pm 0.06$ | $0.16 \pm 0.06$ | $0.17 \pm 0.04$ | $0.11 \pm 0.03$ | $0.14 \pm 0.07$ | $0.15 \pm 0.05$ |
| | PULSE | $0.18 \pm 0.06$ | $0.21 \pm 0.06$ | $0.15 \pm 0.05$ | $0.13 \pm 0.02$ | $0.16 \pm 0.04$ | $0.17 \pm 0.05$ |
| | CoLOR | $0.35 \pm 0.12$ | $0.20 \pm 0.11$ | $0.21 \pm 0.06$ | $0.43 \pm 0.03$ | $0.44 \pm 0.14$ | $0.33 \pm 0.14$ |
| OSCR | DD | $0.55 \pm 0.04$ | $0.44 \pm 0.12$ | $0.44 \pm 0.11$ | $0.57 \pm 0.10$ | $0.52 \pm 0.07$ | $0.51 \pm 0.10$ |
| | uPU | $0.51 \pm 0.08$ | $0.45 \pm 0.08$ | $0.42 \pm 0.12$ | $0.53 \pm 0.05$ | $0.53 \pm 0.05$ | $0.49 \pm 0.09$ |
| | nnPU | $0.50 \pm 0.09$ | $0.41 \pm 0.07$ | $0.44 \pm 0.07$ | $0.54 \pm 0.08$ | $0.52 \pm 0.16$ | $0.48 \pm 0.10$ |
| | BODA | $0.61 \pm 0.04$ | $0.61 \pm 0.04$ | $0.61 \pm 0.02$ | $0.57 \pm 0.05$ | $0.62 \pm 0.04$ | $0.60 \pm 0.04$ |
| | ARPL | $0.63 \pm 0.04$ | $0.64 \pm 0.03$ | $0.60 \pm 0.04$ | $0.60 \pm 0.04$ | $0.59 \pm 0.05$ | $0.61 \pm 0.04$ |
| | ZOC | $\mathbf{0.82 \pm 0.02}$ | $\mathbf{0.83 \pm 0.02}$ | $\mathbf{0.70 \pm 0.02}$ | $\mathbf{0.82 \pm 0.02}$ | $\mathbf{0.82 \pm 0.01}$ | $\mathbf{0.84 \pm 0.01}$ |
| | CAC | $0.56 \pm 0.04$ | $0.57 \pm 0.04$ | $0.56 \pm 0.03$ | $0.51 \pm 0.03$ | $0.54 \pm 0.02$ | $0.55 \pm 0.04$ |
| | PULSE | $0.62 \pm 0.03$ | $0.64 \pm 0.04$ | $0.61 \pm 0.02$ | $0.60 \pm 0.02$ | $0.62 \pm 0.05$ | $0.62 \pm 0.03$ |
| | CoLOR | $0.58 \pm 0.06$ | $0.54 \pm 0.12$ | $0.52 \pm 0.05$ | $0.64 \pm 0.05$ | $0.65 \pm 0.02$ | $0.59 \pm 0.08$ |

Table 13: CIFAR100 dataset without background shift. (AUROC) and (AUPRC) represent the performance for the task of novel class detection while Open-Set Classification Rate (OSCR) measures overall performance of the methods on both known and unknown classes. $\alpha$ is set to 0.05. *Note that all the adaptive methods use ResNet18 and are trained from scratch.*

| Metric | Method | Novel Classes ($\alpha = 0.05 \pm 0.00$) | | | | | |
|---|---|---|---|---|---|---|---|
| | | Baby | Man | Butterfly | Rocket | Streetcar | Summary |
| AUROC | DD | $0.82 \pm 0.04$ | $0.71 \pm 0.09$ | $0.71 \pm 0.11$ | $0.84 \pm 0.09$ | $0.72 \pm 0.15$ | $0.76 \pm 0.11$ |
| | uPU | $0.75 \pm 0.10$ | $0.67 \pm 0.08$ | $0.64 \pm 0.12$ | $0.84 \pm 0.06$ | $0.74 \pm 0.11$ | $0.73 \pm 0.11$ |
| | nnPU | $0.75 \pm 0.05$ | $0.62 \pm 0.05$ | $0.66 \pm 0.07$ | $0.82 \pm 0.04$ | $0.79 \pm 0.08$ | $0.73 \pm 0.10$ |
| | BODA | $0.59 \pm 0.0$ | $0.58 \pm 0.02$ | $0.59 \pm 0.05$ | $0.59 \pm 0.02$ | $0.62 \pm 0.06$ | $0.60 \pm 0.04$ |
| | ARPL | $0.78 \pm 0.03$ | $0.82 \pm 0.03$ | $0.76 \pm 0.03$ | $0.73 \pm 0.04$ | $0.74 \pm 0.05$ | $0.77 \pm 0.05$ |
| | ZOC | $\mathbf{0.97 \pm 0.01}$ | $\mathbf{0.98 \pm 0.01}$ | $\mathbf{0.82 \pm 0.02}$ | $\mathbf{0.98 \pm 0.00}$ | $\mathbf{1.00 \pm 0.00}$ | $\mathbf{0.95 \pm 0.07}$ |
| | CAC | $0.71 \pm 0.01$ | $0.72 \pm 0.02$ | $0.69 \pm 0.03$ | $0.62 \pm 0.03$ | $0.71 \pm 0.01$ | $0.69 \pm 0.04$ |
| | PULSE | $0.73 \pm 0.02$ | $0.77 \pm 0.03$ | $0.72 \pm 0.01$ | $0.70 \pm 0.02$ | $0.72 \pm 0.05$ | $0.73 \pm 0.04$ |
| | CoLOR | $0.82 \pm 0.05$ | $0.76 \pm 0.04$ | $0.76 \pm 0.05$ | $0.82 \pm 0.03$ | $0.82 \pm 0.05$ | $0.80 \pm 0.05$ |
| AUPRC | DD | $0.39 \pm 0.10$ | $0.15 \pm 0.06$ | $0.20 \pm 0.09$ | $0.48 \pm 0.23$ | $0.25 \pm 0.16$ | $0.29 \pm 0.18$ |
| | uPU | $0.20 \pm 0.11$ | $0.13 \pm 0.06$ | $0.11 \pm 0.05$ | $0.45 \pm 0.12$ | $0.21 \pm 0.13$ | $0.22 \pm 0.15$ |
| | nnPU | $0.18 \pm 0.08$ | $0.08 \pm 0.02$ | $0.10 \pm 0.05$ | $0.39 \pm 0.12$ | $0.27 \pm 0.14$ | $0.21 \pm 0.14$ |
| | BODA | $0.07 \pm 0.01$ | $0.06 \pm 0.00$ | $0.06 \pm 0.01$ | $0.07 \pm 0.01$ | $0.07 \pm 0.01$ | $0.07 \pm 0.01$ |
| | ARPL | $0.15 \pm 0.06$ | $0.20 \pm 0.05$ | $0.18 \pm 0.06$ | $0.11 \pm 0.03$ | $0.12 \pm 0.02$ | $0.15 \pm 0.05$ |
| | ZOC | $\mathbf{0.87 \pm 0.03}$ | $\mathbf{0.68 \pm 0.04}$ | $\mathbf{0.25 \pm 0.04}$ | $\mathbf{0.70 \pm 0.05}$ | $\mathbf{0.94 \pm 0.02}$ | $\mathbf{0.69 \pm 0.25}$ |
| | CAC | $0.12 \pm 0.02$ | $0.13 \pm 0.02$ | $0.09 \pm 0.01$ | $0.08 \pm 0.01$ | $0.12 \pm 0.02$ | $0.11 \pm 0.02$ |
| | PULSE | $0.12 \pm 0.01$ | $0.16 \pm 0.03$ | $0.12 \pm 0.02$ | $0.12 \pm 0.03$ | $0.11 \pm 0.02$ | $0.13 \pm 0.03$ |
| | CoLOR | $0.35 \pm 0.12$ | $0.17 \pm 0.05$ | $0.21 \pm 0.07$ | $0.39 \pm 0.05$ | $0.33 \pm 0.11$ | $0.29 \pm 0.12$ |
| OSCR | DD | $0.63 \pm 0.05$ | $0.51 \pm 0.15$ | $0.52 \pm 0.17$ | $0.62 \pm 0.17$ | $0.53 \pm 0.18$ | $0.56 \pm 0.15$ |
| | uPU | $0.54 \pm 0.12$ | $0.49 \pm 0.08$ | $0.45 \pm 0.11$ | $0.63 \pm 0.06$ | $0.53 \pm 0.13$ | $0.53 \pm 0.12$ |
| | nnPU | $0.55 \pm 0.06$ | $0.44 \pm 0.06$ | $0.45 \pm 0.07$ | $0.58 \pm 0.06$ | $0.59 \pm 0.10$ | $0.52 \pm 0.09$ |
| | BODA | $0.62 \pm 0.04$ | $0.63 \pm 0.03$ | $0.61 \pm 0.02$ | $0.56 \pm 0.03$ | $0.64 \pm 0.01$ | $0.61 \pm 0.04$ |
| | ARPL | $0.66 \pm 0.02$ | $0.68 \pm 0.02$ | $0.64 \pm 0.03$ | $0.62 \pm 0.03$ | $0.63 \pm 0.04$ | $0.65 \pm 0.04$ |
| | ZOC | $\mathbf{0.81 \pm 0.01}$ | $\mathbf{0.81 \pm 0.00}$ | $\mathbf{0.70 \pm 0.02}$ | $\mathbf{0.81 \pm 0.00}$ | $\mathbf{0.82 \pm 0.01}$ | $\mathbf{0.79 \pm 0.05}$ |
| | CAC | $0.55 \pm 0.01$ | $0.57 \pm 0.01$ | $0.53 \pm 0.02$ | $0.49 \pm 0.03$ | $0.55 \pm 0.02$ | $0.54 \pm 0.04$ |
| | PULSE | $0.63 \pm 0.02$ | $0.66 \pm 0.02$ | $0.63 \pm 0.02$ | $0.61 \pm 0.02$ | $0.63 \pm 0.04$ | $0.63 \pm 0.03$ |
| | CoLOR | $0.65 \pm 0.02$ | $0.60 \pm 0.03$ | $0.59 \pm 0.05$ | $0.63 \pm 0.04$ | $0.63 \pm 0.06$ | $0.62 \pm 0.04$ |

Table 14: Amazon Reviews dataset with sentiment based distribution shift. AUROC and AUPRC represent the performance for the task of novel class detection while OSCR measures overall performance of the methods on both known and unknown classes. $\alpha$ is the mixture proportion column for the respective novel classes.

| Metric | Method | Novel Classes | | | | | | |
|---|---|---|---|---|---|---|---|---|
| | | $\alpha = 0.32$ | $\alpha = 0.11$ | $\alpha = 0.07$ | $\alpha = 0.07$ | $\alpha = 0.22$ | $\alpha = 0.18$ | |
| | | Musical Instruments | Digital Music | Software | Luxury Beauty | Industrial & Scientific | Prime Pantry | Summary |
| AUROC | DD | $0.75 \pm 0.05$ | $0.81 \pm 0.04$ | $0.80 \pm 0.04$ | $0.73 \pm 0.04$ | $0.58 \pm 0.04$ | $0.63 \pm 0.06$ | $0.72 \pm 0.09$ |
| | uPU | $0.79 \pm 0.04$ | $0.85 \pm 0.06$ | $0.79 \pm 0.03$ | $0.76 \pm 0.03$ | $0.67 \pm 0.04$ | $0.67 \pm 0.09$ | $0.76 \pm 0.08$ |
| | nnPU | $0.79 \pm 0.04$ | $0.85 \pm 0.06$ | $0.79 \pm 0.03$ | $0.76 \pm 0.03$ | $0.67 \pm 0.04$ | $0.67 \pm 0.09$ | $0.76 \pm 0.08$ |
| | BODA | $0.76 \pm 0.03$ | $0.61 \pm 0.13$ | $0.56 \pm 0.06$ | $0.67 \pm 0.07$ | $0.72 \pm 0.04$ | $0.65 \pm 0.09$ | $0.66 \pm 0.10$ |
| | ARPL | $0.66 \pm 0.04$ | $0.75 \pm 0.06$ | $0.70 \pm 0.05$ | $0.66 \pm 0.06$ | $0.72 \pm 0.02$ | $0.74 \pm 0.03$ | $0.70 \pm 0.05$ |
| | CAC | $0.70 \pm 0.03$ | $0.72 \pm 0.05$ | $0.70 \pm 0.03$ | $0.56 \pm 0.08$ | $\mathbf{0.74 \pm 0.05}$ | $0.74 \pm 0.02$ | $0.70 \pm 0.07$ |
| | PULSE | $0.60 \pm 0.03$ | $0.57 \pm 0.07$ | $0.65 \pm 0.06$ | $0.58 \pm 0.03$ | $0.70 \pm 0.02$ | $\mathbf{0.75 \pm 0.02}$ | $0.63 \pm 0.08$ |
| | CoLOR | $\mathbf{0.82 \pm 0.03}$ | $\mathbf{0.87 \pm 0.04}$ | $\mathbf{0.84 \pm 0.06}$ | $\mathbf{0.77 \pm 0.04}$ | $0.68 \pm 0.09$ | $0.72 \pm 0.11$ | $\mathbf{0.79 \pm 0.09}$ |
| AUPRC | DD | $0.68 \pm 0.06$ | $0.62 \pm 0.08$ | $0.32 \pm 0.12$ | $0.27 \pm 0.09$ | $0.33 \pm 0.05$ | $0.38 \pm 0.07$ | $0.43 \pm 0.18$ |
| | uPU | $0.73 \pm 0.04$ | $0.68 \pm 0.10$ | $0.39 \pm 0.07$ | $0.37 \pm 0.10$ | $0.43 \pm 0.07$ | $0.44 \pm 0.14$ | $0.51 \pm 0.17$ |
| | nnPU | $0.73 \pm 0.04$ | $0.68 \pm 0.10$ | $0.39 \pm 0.07$ | $0.37 \pm 0.10$ | $0.43 \pm 0.07$ | $0.44 \pm 0.14$ | $0.51 \pm 0.17$ |
| | BODA | $0.67 \pm 0.07$ | $0.19 \pm 0.11$ | $0.07 \pm 0.01$ | $0.14 \pm 0.04$ | $0.38 \pm 0.06$ | $0.25 \pm 0.07$ | $0.28 \pm 0.21$ |
| | ARPL | $0.43 \pm 0.04$ | $0.27 \pm 0.13$ | $0.14 \pm 0.04$ | $0.12 \pm 0.03$ | $0.36 \pm 0.04$ | $0.33 \pm 0.03$ | $0.27 \pm 0.13$ |
| | CAC | $0.50 \pm 0.04$ | $0.24 \pm 0.06$ | $0.14 \pm 0.02$ | $0.08 \pm 0.01$ | $0.41 \pm 0.07$ | $0.38 \pm 0.03$ | $0.30 \pm 0.16$ |
| | PULSE | $0.38 \pm 0.02$ | $0.14 \pm 0.04$ | $0.10 \pm 0.02$ | $0.09 \pm 0.01$ | $0.33 \pm 0.03$ | $0.36 \pm 0.03$ | $0.21 \pm 0.13$ |
| | CoLOR | $\mathbf{0.76 \pm 0.04}$ | $\mathbf{0.69 \pm 0.07}$ | $\mathbf{0.49 \pm 0.12}$ | $\mathbf{0.35 \pm 0.14}$ | $\mathbf{0.44 \pm 0.12}$ | $\mathbf{0.50 \pm 0.15}$ | $\mathbf{0.54 \pm 0.18}$ |
| OSCR | DD | $0.54 \pm 0.04$ | $0.54 \pm 0.05$ | $0.53 \pm 0.05$ | $0.47 \pm 0.02$ | $0.43 \pm 0.03$ | $0.46 \pm 0.04$ | $0.50 \pm 0.06$ |
| | uPU | $0.59 \pm 0.03$ | $\mathbf{0.59 \pm 0.06}$ | $0.53 \pm 0.04$ | $0.49 \pm 0.02$ | $0.52 \pm 0.03$ | $0.51 \pm 0.03$ | $0.54 \pm 0.05$ |
| | nnPU | $0.59 \pm 0.03$ | $\mathbf{0.59 \pm 0.06}$ | $0.53 \pm 0.04$ | $0.49 \pm 0.02$ | $0.52 \pm 0.03$ | $0.51 \pm 0.03$ | $0.54 \pm 0.05$ |
| | BODA | $0.50 \pm 0.01$ | $0.53 \pm 0.05$ | $\mathbf{0.56 \pm 0.04}$ | $\mathbf{0.51 \pm 0.03}$ | $\mathbf{0.62 \pm 0.01}$ | $0.62 \pm 0.06$ | $\mathbf{0.56 \pm 0.06}$ |
| | ARPL | $0.55 \pm 0.04$ | $0.53 \pm 0.02$ | $0.54 \pm 0.04$ | $\mathbf{0.51 \pm 0.03}$ | $0.62 \pm 0.04$ | $0.60 \pm 0.02$ | $0.56 \pm 0.05$ |
| | CAC | $0.50 \pm 0.02$ | $0.47 \pm 0.04$ | $0.46 \pm 0.03$ | $0.37 \pm 0.04$ | $0.57 \pm 0.05$ | $0.56 \pm 0.03$ | $0.49 \pm 0.07$ |
| | PULSE | $0.52 \pm 0.03$ | $0.47 \pm 0.04$ | $0.54 \pm 0.04$ | $0.49 \pm 0.03$ | $0.63 \pm 0.03$ | $\mathbf{0.64 \pm 0.06}$ | $0.53 \pm 0.07$ |
| | CoLOR | $\mathbf{0.61 \pm 0.04}$ | $\mathbf{0.59 \pm 0.05}$ | $\mathbf{0.56 \pm 0.07}$ | $0.50 \pm 0.05$ | $0.56 \pm 0.03$ | $0.55 \pm 0.07$ | $\mathbf{0.56 \pm 0.06}$ |

Table 15: AUPRC scores with & w/o joint learning for novelty detection on CIFAR100 dataset with distribution shift.

| Novel Class | $\alpha$ | Absolute AUPRC | | | | | | | |
|---|---|---|---|---|---|---|---|---|---|
| | | DD w/o joint learning | DD | uPU w/o joint learning | uPU | nnPU w/o joint learning | nnPU | CoLOR w/o joint learning | CoLOR |
| Baby | $0.23 \pm 0.05$ | $0.69 \pm 0.10$ | $0.77 \pm 0.04$ | $0.63 \pm 0.10$ | $0.64 \pm 0.10$ | $0.61 \pm 0.08$ | $0.61 \pm 0.11$ | $0.65 \pm 0.13$ | $\mathbf{0.80 \pm 0.08}$ |
| Tulip | $0.29 \pm 0.12$ | $0.41 \pm 0.14$ | $0.39 \pm 0.14$ | $0.27 \pm 0.11$ | $0.25 \pm 0.04$ | $0.26 \pm 0.12$ | $0.29 \pm 0.10$ | $0.50 \pm 0.16$ | $\mathbf{0.53 \pm 0.16}$ |
| Crocodile | $0.23 \pm 0.05$ | $0.58 \pm 0.08$ | $0.64 \pm 0.04$ | $0.47 \pm 0.09$ | $0.44 \pm 0.12$ | $0.47 \pm 0.11$ | $0.42 \pm 0.11$ | $0.61 \pm 0.15$ | $\mathbf{0.68 \pm 0.10}$ |
| Dolphin | $0.29 \pm 0.12$ | $\mathbf{0.65 \pm 0.12}$ | $\mathbf{0.65 \pm 0.12}$ | $0.50 \pm 0.06$ | $0.49 \pm 0.08$ | $0.50 \pm 0.08$ | $0.45 \pm 0.11$ | $\mathbf{0.65 \pm 0.12}$ | $0.64 \pm 0.19$ |
| Man | $0.23 \pm 0.05$ | $0.59 \pm 0.19$ | $0.64 \pm 0.13$ | $0.31 \pm 0.04$ | $0.41 \pm 0.14$ | $0.28 \pm 0.12$ | $0.46 \pm 0.13$ | $0.68 \pm 0.09$ | $\mathbf{0.79 \pm 0.07}$ |

Table 16: AUPRC scores with & w/o joint learning for novelty detection on Amazon Reviews dataset with distribution shift.

| Novel Class | $\alpha$ | Absolute AUPRC | | | | | | | |
|---|---|---|---|---|---|---|---|---|---|
| | | DD w/o joint learning | DD | uPU w/o joint learning | uPU | nnPU w/o joint learning | nnPU | CoLOR w/o joint learning | CoLOR |
| Musical Instruments | 0.16 | $0.37 \pm 0.09$ | $0.40 \pm 0.1$ | $0.34 \pm 0.09$ | $0.45 \pm 0.13$ | $0.34 \pm 0.09$ | $0.45 \pm 0.13$ | $0.48 \pm 0.11$ | $\mathbf{0.55 \pm 0.13}$ |
| Digital Music | 0.05 | $0.28 \pm 0.07$ | $0.29 \pm 0.09$ | $0.23 \pm 0.10$ | $0.24 \pm 0.12$ | $0.23 \pm 0.10$ | $0.24 \pm 0.12$ | $0.33 \pm 0.13$ | $\mathbf{0.35 \pm 0.15}$ |
| Software | 0.06 | $0.36 \pm 0.05$ | $0.37 \pm 0.11$ | $0.31 \pm 0.13$ | $0.36 \pm 0.08$ | $0.31 \pm 0.13$ | $0.36 \pm 0.08$ | $0.48 \pm 0.11$ | $\mathbf{0.49 \pm 0.12}$ |
| Luxury Beauty | 0.07 | $0.27 \pm 0.15$ | $0.25 \pm 0.08$ | $0.28 \pm 0.12$ | $0.35 \pm 0.11$ | $0.28 \pm 0.12$ | $0.35 \pm 0.11$ | $0.35 \pm 0.14$ | $\mathbf{0.36 \pm 0.13}$ |
| Industrial & Scientific | 0.12 | $0.13 \pm 0.01$ | $0.14 \pm 0.03$ | $0.11 \pm 0.02$ | $0.16 \pm 0.05$ | $0.11 \pm 0.02$ | $0.16 \pm 0.05$ | $0.17 \pm 0.02$ | $\mathbf{0.21 \pm 0.08}$ |

Table 17: MSP/Entropy based OOD detection on SUN397 with low $\alpha = 0.05 \pm 0.01$ on three types of novel categories

| | [water, ice, snow, etc.] | | [mountains, hills, desert, sky, etc.] | | [forest, field, jungle, etc.] | |
|---|---|---|---|---|---|---|
| | AUROC | AUPRC | AUROC | AUPRC | AUROC | AUPRC |
| With Shift | | | | | | |
| MSP | $0.57 \pm 0.05$ | $0.05 \pm 0.01$ | $0.58 \pm 0.06$ | $0.05 \pm 0.01$ | $0.56 \pm 0.05$ | $0.07 \pm 0.03$ |
| Entropy | $0.58 \pm 0.07$ | $0.08 \pm 0.04$ | $0.58 \pm 0.08$ | $0.07 \pm 0.03$ | $0.56 \pm 0.07$ | $0.10 \pm 0.06$ |
| Without Shift | | | | | | |
| MSP | $0.59 \pm 0.05$ | $0.04 \pm 0.00$ | $0.59 \pm 0.05$ | $0.05 \pm 0.01$ | $0.58 \pm 0.07$ | $0.06 \pm 0.03$ |
| Entropy | $0.59 \pm 0.07$ | $0.08 \pm 0.02$ | $0.59 \pm 0.08$ | $0.08 \pm 0.04$ | $0.59 \pm 0.10$ | $0.09 \pm 0.03$ |

Table 18: MSP/Entropy based OOD detection on SUN397 with high $\alpha = 0.5 \pm 0.10$ on three types of novel categories

|  | [water, ice, snow, etc.] | | [mountains, hills, desert, sky, etc.] | | [forest, field, jungle, etc.] | |
|---|---|---|---|---|---|---|
|  | AUROC | AUPRC | AUROC | AUPRC | AUROC | AUPRC |
| With Shift | | | | | | |
| MSP | $0.52 \pm 0.06$ | $0.55 \pm 0.05$ | $0.61 \pm 0.06$ | $0.28 \pm 0.04$ | $0.60 \pm 0.08$ | $0.50 \pm 0.09$ |
| Entropy | $0.52 \pm 0.06$ | $0.58 \pm 0.05$ | $0.63 \pm 0.06$ | $0.45 \pm 0.07$ | $0.62 \pm 0.09$ | $0.67 \pm 0.04$ |
| Without Shift | | | | | | |
| MSP | $0.59 \pm 0.06$ | $0.47 \pm 0.04$ | $0.61 \pm 0.08$ | $0.27 \pm 0.04$ | $0.58 \pm 0.08$ | $0.50 \pm 0.05$ |
| Entropy | $0.59 \pm 0.08$ | $0.59 \pm 0.07$ | $0.63 \pm 0.10$ | $0.41 \pm 0.10$ | $0.59 \pm 0.10$ | $0.63 \pm 0.09$ |

Table 19: Effect of background shift on Top-1 accuracies of remaining baselines. We do not expect any performance reduction of ZOC as it is a zero-shot method that does not utilize source data at all, rendering any distribution shift between $P_{\mathcal{S}}$ and $P_{\mathcal{T}}$ irrelevant. Such models are influenced only by target datasets that drift from their pretraining datasets.

| Methods | w/ DS | w/o DS |
|---|---|---|
| BODA (ViT-L/14) | $0.75 \pm 0.13$ | $0.94 \pm 0.00$ |
| ARPL (custom default) | $0.75 \pm 0.03$ | $0.91 \pm 0.01$ |
| ZOC (custom default) | $0.62 \pm 0.05$ | $0.59 \pm 0.00$ |
| ANNA (custom default) | $0.65 \pm 0.04$ | $0.87 \pm 0.01$ |

Table 20: CoLOR performance for different target recall constraints for the SUN397 dataset with novel identity as outdoor scenes from the group [water, ice, snow, etc.] and novel class size $\alpha = 0.06 \pm 0.01$ using ResNet50 model. Note that out of $\|\boldsymbol{\alpha}\|$ novelty detection heads each corresponding to a candidate value $\hat{\alpha} \in \boldsymbol{\alpha}$, we select the head achieving highest recall in the validation set ($\mathrm{argmax}_{w_{\hat{\alpha}}^{\alpha}:\alpha \in \boldsymbol{\alpha}, \hat{\beta}(w_{\hat{\alpha}}^{\alpha}) < \beta} \hat{\alpha}(w_{\hat{\alpha}}^{\alpha})$). Consequently, the reported AUROC, AUPRC and OSCR performance correspond to that selected head. Scores of the selected novelty detection head for each seed are highlighted in bold below whereas the highest scores amongst all the novelty detection heads for each seed have been underlined. These heads having scores underlined were not selected besides having highest scores for a metric, because they simply did not satisfy the selection criteria, i.e. their validation recall values were not the highest for the corresponding seed among all the novelty detection heads.

| Target Recall | Validation Recall across 5 seeds | | | | | AUROC across 5 seeds | | | | | AUPRC across 5 seeds | | | | | OSCR across 5 seeds | | | | |
|---|---|---|---|---|---|---|---|---|---|---|---|---|---|---|---|---|---|---|---|---|
| $\hat{\alpha}$ | 0 | 8 | 1057 | 103 | 573 | 0 | 8 | 1057 | 103 | 573 | 0 | 8 | 1057 | 103 | 573 | 0 | 8 | 1057 | 103 | 573 |
| 0.02 | **0.18** | 0.06 | 0.10 | **0.08** | 0.08 | **0.98** | 0.97 | 0.95 | **0.99** | 0.94 | **0.93** | 0.87 | 0.87 | **0.94** | 0.43 | **0.82** | 0.82 | 0.75 | **0.84** | 0.67 |
| 0.05 | 0.18 | 0.06 | 0.00 | 0.08 | 0.15 | 0.99 | 0.96 | 0.48 | 1.00 | 0.99 | 0.94 | 0.82 | 0.05 | 0.95 | 0.95 | 0.83 | 0.81 | 0.25 | 0.85 | 0.78 |
| 0.10 | 0.17 | 0.02 | **0.14** | 0.06 | 0.09 | 0.98 | 0.82 | **0.95** | 1.00 | 0.93 | 0.93 | 0.13 | **0.85** | 0.96 | 0.38 | 0.81 | 0.66 | **0.76** | 0.85 | 0.65 |
| 0.15 | 0.16 | 0.10 | 0.01 | 0.06 | 0.04 | 0.99 | 0.95 | 0.42 | 0.99 | 0.95 | 0.95 | 0.80 | 0.05 | 0.92 | 0.64 | 0.84 | 0.80 | 0.23 | 0.84 | 0.65 |
| 0.20 | 0.12 | **0.12** | 0.14 | 0.05 | 0.03 | 0.97 | **0.97** | 0.95 | 0.99 | 0.93 | 0.79 | **0.90** | 0.86 | 0.94 | 0.42 | 0.76 | **0.85** | 0.78 | 0.84 | 0.64 |
| 0.25 | 0.07 | 0.00 | 0.07 | 0.03 | **0.18** | 0.99 | 0.68 | 0.47 | 0.98 | **0.99** | 0.83 | 0.08 | 0.05 | 0.80 | **0.97** | 0.77 | 0.52 | 0.27 | 0.81 | **0.82** |
| 0.30 | 0.12 | 0.01 | 0.00 | 0.07 | 0.00 | 0.99 | 0.81 | 0.44 | 1.00 | 0.86 | 0.92 | 0.13 | 0.05 | 0.95 | 0.29 | 0.77 | 0.65 | 0.25 | 0.85 | 0.61 |
| 0.35 | 0.11 | 0.04 | 0.02 | 0.01 | 0.04 | 0.97 | 0.81 | 0.47 | 0.84 | 0.83 | 0.55 | 0.13 | 0.05 | 0.21 | 0.25 | 0.74 | 0.65 | 0.26 | 0.69 | 0.59 |
| 0.40 | 0.06 | 0.00 | 0.09 | 0.01 | 0.00 | 0.94 | 0.84 | 0.43 | 0.87 | 0.78 | 0.41 | 0.16 | 0.05 | 0.25 | 0.18 | 0.73 | 0.69 | 0.23 | 0.71 | 0.57 |
| 0.45 | 0.04 | 0.00 | 0.00 | 0.07 | 0.00 | 0.99 | 0.74 | 0.54 | 0.97 | 0.79 | 0.89 | 0.11 | 0.06 | 0.82 | 0.22 | 0.77 | 0.58 | 0.34 | 0.82 | 0.57 |

Figure 5: Effect of density of search grid $\boldsymbol{\alpha}$ on the performance of CoLOR. We consider the range [0.02, 0.45] to search the candidate $\alpha$ and vary the density (i.e. number of candidate $\alpha$s) in the same interval). We report AUROC, AUPRC, OSCR, and Top-1 Accuracy for different novel class groupings. We use ResNet50 features here that are pretrained on ImageNet1K.

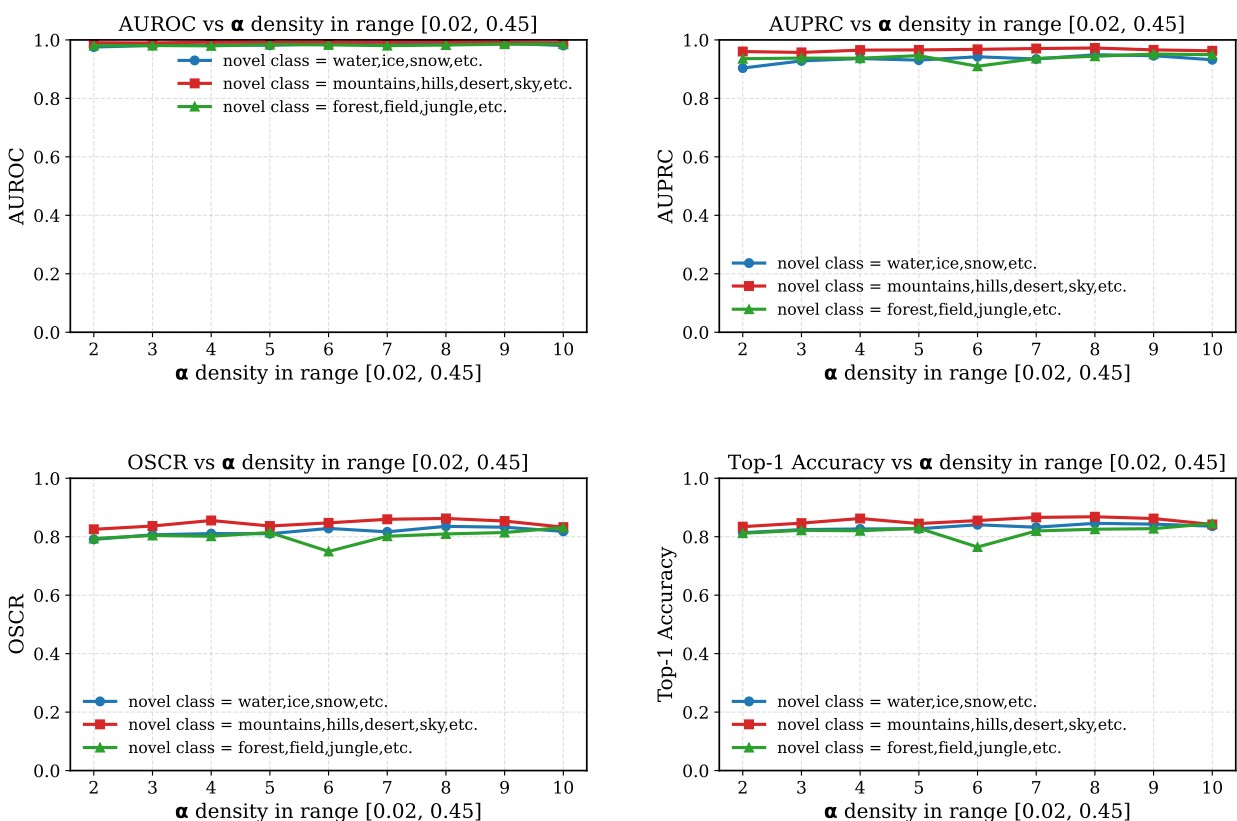

Table 21: Effect of density of search grid $\boldsymbol{\alpha}$ on the performance of CoLOR. We consider the range [0.02, 0.45] to search the candidate $\alpha$ and vary the density (i.e. number of candidate $\alpha$s) in the same interval. We report AUROC, AUPRC, OSCR, and Top-1 Accuracy for different novel class groupings: (Group 0: [water, ice, snow, etc.]; Group 1: [mountains, hills, desert, sky, etc.]; Group 2:[forest, field, jungle, etc.]). We use ResNet50 features here that are pretrained on ImageNet1K.

| $\boldsymbol{\alpha}$ density | Performance Metrics across 3 novel class groups. | | | | | | | | | | | |
| | AUROC | | | AUPRC | | | OSCR | | | Top-1 Acc. | | |
| | 0 | 1 | 2 | 0 | 1 | 2 | 0 | 1 | 2 | 0 | 1 | 2 |
|---|---|---|---|---|---|---|---|---|---|---|---|---|
| 2 | 0.98 | 0.99 | 0.98 | 0.90 | 0.96 | 0.94 | 0.79 | 0.83 | 0.79 | 0.81 | 0.83 | 0.81 |
| 3 | 0.98 | 0.99 | 0.98 | 0.93 | 0.96 | 0.94 | 0.81 | 0.84 | 0.80 | 0.82 | 0.85 | 0.82 |
| 4 | 0.98 | 0.99 | 0.98 | 0.94 | 0.97 | 0.94 | 0.81 | 0.86 | 0.80 | 0.83 | 0.86 | 0.82 |
| 5 | 0.98 | 0.99 | 0.98 | 0.93 | 0.97 | 0.95 | 0.81 | 0.84 | 0.81 | 0.83 | 0.84 | 0.83 |
| 6 | 0.99 | 0.99 | 0.98 | 0.94 | 0.97 | 0.91 | 0.83 | 0.85 | 0.75 | 0.84 | 0.86 | 0.76 |
| 7 | 0.98 | 0.99 | 0.98 | 0.93 | 0.97 | 0.94 | 0.82 | 0.86 | 0.80 | 0.83 | 0.87 | 0.82 |
| 8 | 0.99 | 0.99 | 0.98 | 0.95 | 0.97 | 0.94 | 0.84 | 0.86 | 0.81 | 0.85 | 0.87 | 0.83 |
| 9 | 0.99 | 0.99 | 0.99 | 0.95 | 0.97 | 0.95 | 0.83 | 0.85 | 0.81 | 0.84 | 0.86 | 0.83 |
| 10 | 0.98 | 0.99 | 0.99 | 0.93 | 0.96 | 0.95 | 0.82 | 0.83 | 0.83 | 0.84 | 0.84 | 0.85 |

Table 22: Effect of range of search grid $\alpha$ on the performance of CoLOR. We keep the density (i.e. number of candidate $\alpha$s in the search interval) constant to 10 and vary the range. We report AUROC, AUPRC, OSCR, and Top-1 Accuracy for different novel class groupings: (Group 0: [water, ice, snow, etc.]; Group 1: [mountains, hills, desert, sky, etc.]; Group 2:[forest, field, jungle, etc.]). We use ResNet50 features here that are pretrained on ImageNet1K.

| $\alpha$ range | AUROC 0 | AUROC 1 | AUROC 2 | AUPRC 0 | AUPRC 1 | AUPRC 2 | OSCR 0 | OSCR 1 | OSCR 2 | Top-1 Acc. 0 | Top-1 Acc. 1 | Top-1 Acc. 2 |
|---|---|---|---|---|---|---|---|---|---|---|---|---|
| | | | | | Performance Metrics across 3 novel class groups. | | | | | | | |
| $[0.001, 0.01]$ | 0.98 | 0.99 | 0.98 | 0.90 | 0.96 | 0.94 | 0.79 | 0.83 | 0.79 | 0.81 | 0.83 | 0.81 |
| $[0.01, 0.1]$ | 0.98 | 0.99 | 0.98 | 0.93 | 0.96 | 0.94 | 0.81 | 0.84 | 0.80 | 0.82 | 0.85 | 0.82 |
| $[0.1, 1.0]$ | 0.98 | 0.99 | 0.98 | 0.94 | 0.97 | 0.94 | 0.81 | 0.86 | 0.80 | 0.83 | 0.86 | 0.82 |
| $[0.2, 0.45]$ | 0.97 | 0.98 | 0.99 | 0.59 | 0.87 | 0.95 | 0.72 | 0.77 | 0.83 | 0.75 | 0.79 | 0.85 |
| $[0.3, 0.45]$ | 0.98 | 0.99 | 0.99 | 0.70 | 0.95 | 0.94 | 0.75 | 0.78 | 0.76 | 0.77 | 0.78 | 0.77 |
| $[0.4, 0.45]$ | 0.95 | 0.99 | 0.98 | 0.58 | 0.95 | 0.87 | 0.71 | 0.78 | 0.75 | 0.75 | 0.79 | 0.77 |
| $[0.5, 1.0]$ | 0.96 | 0.99 | 0.98 | 0.69 | 0.96 | 0.90 | 0.74 | 0.77 | 0.76 | 0.77 | 0.78 | 0.77 |
| $[0.6, 1.0]$ | 0.97 | 0.99 | 0.99 | 0.61 | 0.96 | 0.97 | 0.75 | 0.77 | 0.76 | 0.78 | 0.78 | 0.76 |
| $[0.7, 1.0]$ | 0.97 | 0.99 | 0.99 | 0.61 | 0.92 | 0.91 | 0.74 | 0.78 | 0.72 | 0.77 | 0.78 | 0.73 |
| $[0.8, 1.0]$ | 0.97 | 0.99 | 0.99 | 0.63 | 0.92 | 0.96 | 0.74 | 0.78 | 0.73 | 0.76 | 0.78 | 0.73 |
| $[0.9, 1.0]$ | 0.97 | 0.99 | 0.99 | 0.74 | 0.99 | 0.94 | 0.73 | 0.77 | 0.75 | 0.76 | 0.78 | 0.76 |
| $[0.02, 0.45]$ (ours) | 0.98 | 0.0.99 | 0.0.99 | 0.93 | 0.96 | 0.95 | 0.82 | 0.83 | 0.83 | 0.84 | 0.84 | 0.85 |

Table 23: Impact of joint learning on closed-set classification and novelty detection on SUN397 across three novel groups. We compare three model variants: (i) Source-only, using only closed-set classification heads ($w^c$); (ii) ConOp, using only novelty detection heads ($w^\alpha$) learned via constrained optimization; and (iii) the full CoLOR model trained with a multitask objective for both classification and novelty detection. All methods are evaluated on $S_\mathcal{T}$ using ImageNet-pretrained ResNet50 features.

| Metric | Method | Novel Classes (natural outdoor scenes/places) | | | Summary |
|---|---|---|---|---|---|
| | | $\alpha = 0.06 \pm 0.01$ [water, ice, snow, etc.] | $\alpha = 0.06 \pm 0.01$ [mountains, hills, desert, sky, etc.] | $\alpha = 0.08 \pm 0.04$ [forest, field, jungle, etc.] | |
| AUROC | ConOp | $0.96 \pm 0.02$ | $0.97 \pm 0.04$ | $0.97 \pm 0.02$ | $0.96 \pm 0.03$ |
| | CoLOR | $\mathbf{0.98 \pm 0.02}$ | $\mathbf{0.98 \pm 0.01}$ | $\mathbf{0.98 \pm 0.02}$ | $\mathbf{0.98 \pm 0.02}$ |
| AUPRC | ConOp | $0.77 \pm 0.11$ | $0.78 \pm 0.21$ | $0.87 \pm 0.08$ | $0.80 \pm 0.14$ |
| | CoLOR | $\mathbf{0.92 \pm 0.04}$ | $\mathbf{0.93 \pm 0.05}$ | $\mathbf{0.89 \pm 0.15}$ | $\mathbf{0.91 \pm 0.09}$ |
| Top-1 Acc. | Source-only | $0.71 \pm 0.03$ | $0.73 \pm 0.05$ | $0.72 \pm 0.03$ | $0.72 \pm 0.03$ |
| | CoLOR | $\mathbf{0.77 \pm 0.11}$ | $\mathbf{0.78 \pm 0.21}$ | $\mathbf{0.87 \pm 0.08}$ | $\mathbf{0.80 \pm 0.14}$ |
| OSCR | CoLOR | $\mathbf{0.82 \pm 0.03}$ | $\mathbf{0.81 \pm 0.03}$ | $\mathbf{0.81 \pm 0.05}$ | $\mathbf{0.81 \pm 0.04}$ |

Table 24: Impact of joint learning on closed-set classification and novelty detection on SUN397 across three novel groups. We compare three model variants: (i) Source-only, using only closed-set classification heads ($w^c$); (ii) ConOp, using only novelty detection heads ($w^\alpha$) learned via constrained optimization; and (iii) the full CoLOR model trained with a multitask objective for both classification and novelty detection. All the methods are evaluated on $S_\mathcal{T}$ and using pretrained CLIP ViT-L/14 features.

| Metric | Method | Novel Classes (natural outdoor scenes/places) | | | Summary |
|---|---|---|---|---|---|
| | | $\alpha = 0.06 \pm 0.01$ [water, ice, snow, etc.] | $\alpha = 0.06 \pm 0.01$ [mountains, hills, desert, sky, etc.] | $\alpha = 0.08 \pm 0.04$ [forest, field, jungle, etc.] | |
| AUROC | ConOp | $\mathbf{0.99 \pm 0.01}$ | $\mathbf{0.99 \pm 0.00}$ | $\mathbf{0.99 \pm 0.00}$ | $\mathbf{0.99 \pm 0.01}$ |
| | CoLOR | $\mathbf{0.99 \pm 0.02}$ | $\mathbf{0.99 \pm 0.01}$ | $\mathbf{0.99 \pm 0.01}$ | $\mathbf{0.99 \pm 0.01}$ |
| AUPRC | ConOp | $\mathbf{0.97 \pm 0.02}$ | $\mathbf{0.96 \pm 0.03}$ | $\mathbf{0.97 \pm 0.02}$ | $\mathbf{0.97 \pm 0.02}$ |
| | CoLOR | $0.95 \pm 0.04$ | $0.95 \pm 0.01$ | $0.96 \pm 0.03$ | $0.95 \pm 0.03$ |
| Top-1 Acc. | Source-only | $0.97 \pm 0.01$ | $0.97 \pm 0.01$ | $0.96 \pm 0.01$ | $0.97 \pm 0.01$ |
| | CoLOR | $\mathbf{0.98 \pm 0.01}$ | $\mathbf{0.98 \pm 0.00}$ | $\mathbf{0.97 \pm 0.01}$ | $\mathbf{0.98 \pm 0.01}$ |
| OSCR | CoLOR | $\mathbf{0.96 \pm 0.02}$ | $\mathbf{0.97 \pm 0.01}$ | $\mathbf{0.96 \pm 0.01}$ | $\mathbf{0.96 \pm 0.01}$ |

Figure 6: Synthetic experiment setup following the definition 2 to measure novelty detection performance as a function of angular separation between $\mu$ and $\eta$ vectors, $\theta$, (Top) and $r_\eta$ (Bottom). We measure AUROC (Left) and AUPRC (Right). Here, $\alpha = 0.15$, $d = 3000$, $\sigma = 1/d$, $r_\mu = 0.1$. We train on 2000 samples and test on 3000 samples.

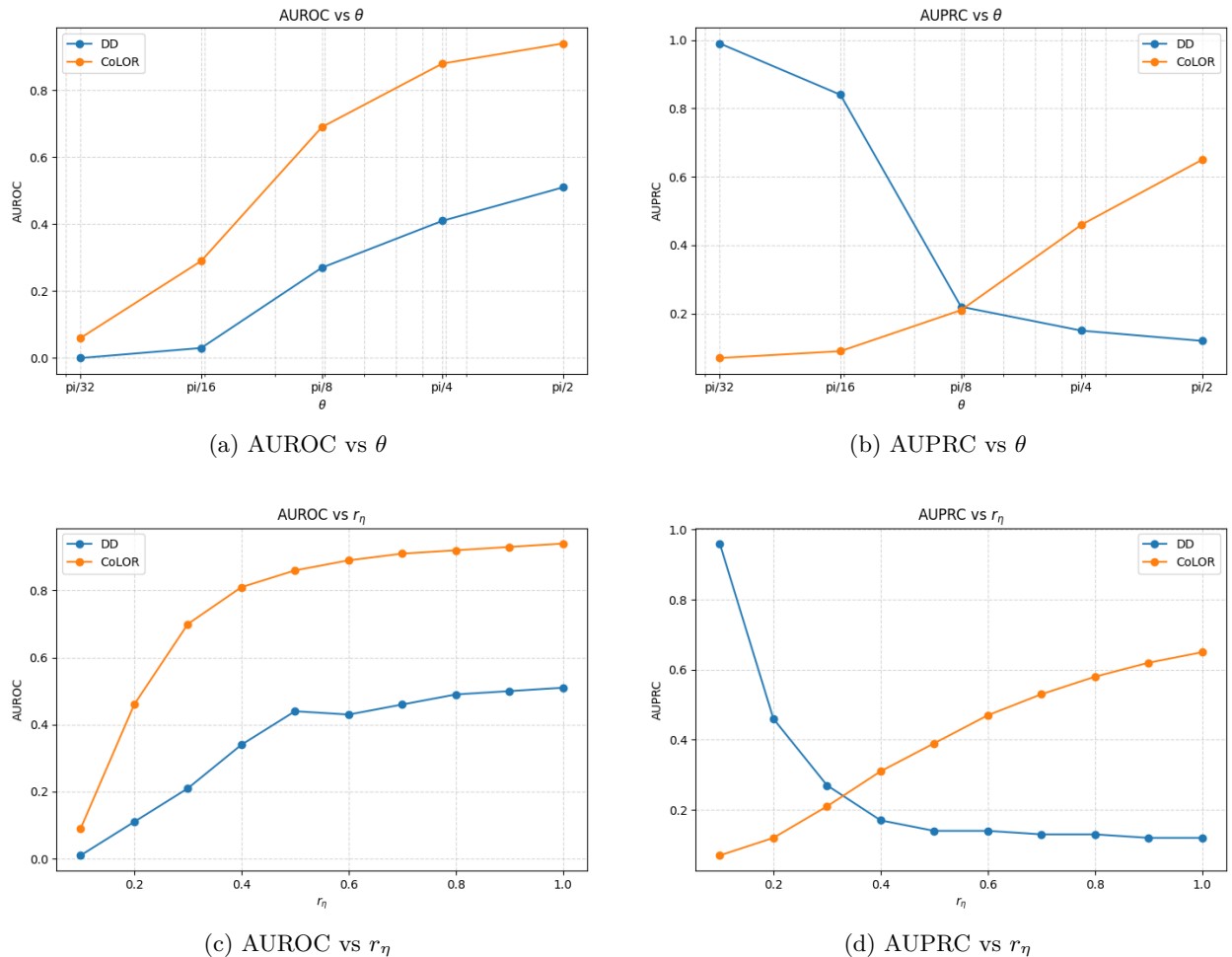

(a) AUROC vs $\theta$

(b) AUPRC vs $\theta$

(c) AUROC vs $r_\eta$

(d) AUPRC vs $r_\eta$

Table 25: Novelty detection performance as a function of angular separation between $\mu$ and $\eta$ vectors, $\theta$, in synthetic experiment setup following the definition 2. Here, $\alpha = 0.15$, $d = 3000$, $\sigma = 1/d$, $r_\mu = 0.1$, $r_\eta = 1.0$. We train on 2000 samples and test on 3000 samples.

| $\theta$ | DD | | CoLOR | |
|---|---|---|---|---|
| | AUROC | AUPRC | AUROC | AUPRC |
| $\pi/2$ | 0.51 | 0.12 | 0.94 | 0.65 |
| $\pi/4$ | 0.41 | 0.15 | 0.88 | 0.46 |
| $\pi/8$ | 0.27 | 0.22 | 0.69 | 0.21 |
| $\pi/16$ | 0.03 | 0.84 | 0.29 | 0.09 |
| $\pi/32$ | 0.00 | 0.99 | 0.06 | 0.07 |

Table 26: Novelty detection performance as a function of $r_\eta$ in synthetic experiment setup following the definition 2. $\theta = \frac{\pi}{2}$ is the angular separation between $\mu$ and $\eta$ vectors. Here, $\alpha = 0.15$, $d = 3000$, $\sigma = 1/d$, $r_\mu = 0.1$. We train on 2000 samples and test on 3000 samples.

| $r_\eta$ | DD | | CoLOR | |
|---|---|---|---|---|
| | AUROC | AUPRC | AUROC | AUPRC |
| 1.0 | 0.51 | 0.12 | 0.94 | 0.65 |
| 0.9 | 0.50 | 0.12 | 0.93 | 0.62 |
| 0.8 | 0.49 | 0.13 | 0.92 | 0.58 |
| 0.7 | 0.46 | 0.13 | 0.91 | 0.53 |
| 0.6 | 0.43 | 0.14 | 0.89 | 0.47 |
| 0.5 | 0.44 | 0.14 | 0.86 | 0.39 |
| 0.4 | 0.34 | 0.17 | 0.81 | 0.31 |
| 0.3 | 0.21 | 0.27 | 0.70 | 0.21 |
| 0.2 | 0.11 | 0.46 | 0.46 | 0.12 |
| 0.1 | 0.01 | 0.96 | 0.09 | 0.07 |

