| | DD | **1.00 ± 0.00** | **1.00 ± 0.00** | **1.00 ± 0.00** | **1.00 ± 0.00** |
| | uPU | **1.00 ± 0.00** | **1.00 ± 0.00** | **1.00 ± 0.00** | **1.00 ± 0.00** |
| AUROC | nnPU | **1.00 ± 0.00** | **1.00 ± 0.00** | **1.00 ± 0.00** | **1.00 ± 0.00** |
| | BODA | 0.87 ± 0.03 | 0.88 ± 0.04 | 0.90 ± 0.05 | 0.88 ± 0.04 |
| | ARPL | 0.86 ± 0.03 | 0.81 ± 0.03 | 0.84 ± 0.05 | 0.84 ± 0.04 |
| | ZOC | 0.84 ± 0.02 | 0.86 ± 0.02 | 0.77 ± 0.12 | 0.82 ± 0.08 |
| | CAC | 0.99 ± 0.01 | 0.99 ± 0.00 | 0.99 ± 0.01 | 0.99 ± 0.01 |
| | PULSE | 0.98 ± 0.01 | 0.98 ± 0.01 | 0.97 ± 0.02 | 0.98 ± 0.01 |
| | CoLOR | **1.00 ± 0.00** | **1.00 ± 0.00** | **1.00 ± 0.00** | **1.00 ± 0.00** |
| | DD | **1.00 ± 0.00** | **1.00 ± 0.00** | **1.00 ± 0.00** | **1.00 ± 0.00** |
| | uPU | **1.00 ± 0.00** | **1.00 ± 0.00** | **1.00 ± 0.00** | **1.00 ± 0.00** |
| AUPRC | nnPU | **1.00 ± 0.00** | **1.00 ± 0.00** | **1.00 ± 0.00** | **1.00 ± 0.00** |
| | BODA | 0.18 ± 0.04 | 0.20 ± 0.06 | 0.34 ± 0.22 | 0.24 ± 0.15 |
| | ARPL | 0.20 ± 0.04 | 0.16 ± 0.04 | 0.25 ± 0.09 | 0.20 ± 0.07 |
| | ZOC | 0.19 ± 0.02 | 0.21 ± 0.04 | 0.25 ± 0.11 | 0.22 ± 0.07 |
| | CAC | 0.87 ± 0.10 | 0.91 ± 0.03 | 0.86 ± 0.08 | 0.88 ± 0.08 |
| | PULSE | 0.76 ± 0.08 | 0.76 ± 0.08 | 0.74 ± 0.11 | 0.75 ± 0.09 |
| | CoLOR | 0.99 ± 0.01 | 0.99 ± 0.01 | 0.99 ± 0.01 | 0.99 ± 0.01 |
| | DD | **0.99 ± 0.00** | **0.99 ± 0.00** | **0.99 ± 0.00** | **0.99 ± 0.00** |
| | uPU | **0.99 ± 0.00** | **0.99 ± 0.00** | **0.99 ± 0.00** | **0.99 ± 0.00** |
| OSCR | nnPU | **0.99 ± 0.00** | **0.99 ± 0.00** | **0.99 ± 0.00** | **0.99 ± 0.00** |
| | BODA | 0.94 ± 0.01 | 0.94 ± 0.01 | 0.92 ± 0.02 | 0.93 ± 0.02 |
| | ARPL | 0.82 ± 0.02 | 0.78 ± 0.03 | 0.80 ± 0.05 | 0.80 ± 0.04 |
| | ZOC | 0.50 ± 0.01 | 0.51 ± 0.02 | 0.45 ± 0.07 | 0.49 ± 0.05 |
| | CAC | 0.98 ± 0.01 | 0.99 ± 0.00 | 0.98 ± 0.01 | 0.98 ± 0.01 |
| | PULSE | 0.98 ± 0.01 | 0.98 ± 0.01 | 0.97 ± 0.02 | 0.97 ± 0.02 |
| | CoLOR | 0.98 ± 0.00 | 0.98 ± 0.00 | 0.98 ± 0.00 | 0.98 ± 0.00 |

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

| 1.0 | 0.51 | 0.12 | 0.94 | 0.65 |
| 0.9 | 0.50 | 0.12 | 0.93 | 0.62 |
| 0.8 | 0.49 | 0.13 | 0.92 | 0.58 |
| 0.7 | 0.46 | 0.13 | 0.91 | 0.53 |
| 0.6 | 0.43 | 0.14 | 0.89 | 0.47 |
| 0.5 | 0.44 | 0.14 | 0.86 | 0.39 |
| 0.4 | 0.34 | 0.17 | 0.81 | 0.31 |
| 0.3 | 0.21 | 0.27 | 0.70 | 0.21 |
| 0.2 | 0.11 | 0.46 | 0.46 | 0.12 |
| 0.1 | 0.01 | 0.96 | 0.09 | 0.07 |