# OpenReview forum: "Open-Set Domain Adaptation Under Background Distribution Shift: Challenges and A Provably Efficient Solution"
_TMLR — Accepted by TMLR_

### Review · Reviewer_52XA · 2025-12-20

**Summary Of Contributions:**

1. Addresses the underexplored OSDA scenario with concurrent background distribution shift and novel classes, filling a research gap.

2. Proposes CoLOR, a scalable constrained learning method with formal theoretical guarantees for identifiability and sample efficiency in overparameterized regimes.

3. Conducts comprehensive empirical validations across image (SUN397, CIFAR100) and text (Amazon Reviews) datasets, showing CoLOR outperforms baselines under background shift.

4. Provides new insights into how novel class size influences OSDA performance, especially for small novel class proportions.

**Audience:**

Yes

**Audience Explanation:**

Yes. TMLR’s audience (ML researchers focusing on domain adaptation, open-set recognition, and distribution shift) will find the findings valuable—especially the provably efficient CoLOR method and insights on background shift/novel class size.

**Broader Impact Concerns:**

N/A.

**Claims And Evidence:**

Yes

**Claims Explanation:**

Yes, the submission’s claims are well-supported:

1. Theoretical claims (CoLOR’s guarantees under background shift) are validated via rigorous proofs (Lemma 1, Theorem 1) and linear-Gaussian analysis.

2. Empirical claims (superior performance) are backed by comprehensive experiments on 3 modalities (SUN397, CIFAR100, Amazon Reviews) with consistent gains over 10+ baselines.

3. Insights on novel class size (α) are confirmed by quantitative analysis (Figure 3b) showing CoLOR’s robustness for small α.

**Requested Changes:**

1. The rationale for selecting the grid range of candidate novel class ratio (α = [0.02, 0.45]) lacks explicit justification. The paper does not explain why this specific interval is chosen over other ranges (e.g., lower than 0.02 or higher than 0.45) or how the grid density impacts model performance, leaving uncertainty about whether the optimal α is adequately covered.

2. The theoretical analysis is restricted to the simplified k=1 (PU-learning) scenario, but the empirical evaluations primarily focus on multi-class (k>1) settings. The paper fails to provide a rigorous justification for extending the k=1 theoretical guarantees to k>1 cases, including how inter-class relationships in multi-class scenarios affect the method’s effectiveness.

3. No ablation study is conducted to isolate the contribution of the shared representation architecture in CoLOR. It remains unclear whether the performance gains stem from the constrained learning rule itself or the shared encoder between classification and novelty detection heads, making it difficult to validate the architectural design’s necessity.

4. The paper does not systematically evaluate CoLOR’s robustness when the separability assumption (Assumption 1) is severely violated. While it mentions Amazon Reviews has imperfect separability, there is no quantitative analysis of how performance degrades as separability decreases, nor a comparison with baselines under varying degrees of non-separability.

---

> ### Author Response · Authors · 2026-01-11
> **Addressing Reviewer 52XA's comments**
>
> We thank the reviewer for their insightful and constructive feedback, and are glad to see that the reviewer finds our results valuable to the community. Below we address the concerns/changes suggested by the reviewer:
> > 1. a) The rationale for selecting the grid range of candidate novel class ratio ($\alpha$ = [0.02, 0.45]) lacks explicit justification...
>
> You’re right that more rationale for why that range was selected will be beneficial and we have included it in the revised manuscript (section A.5.3).
>
> $\boldsymbol{\alpha} = [0.02, 0.45]$ is not claimed to be universal, but chosen to study the most informative regime. We observed that when the size of the novel class (true size, not the one estimated by the model) is below 1%, all baselines failed. Accordingly, we set the lower bound of our search grid on that scale, but we show in experiments that including lower values does not change performance by much.
>
> As for the upper bound, we observe that our method is relatively robust when the true novel class size is small (i.e. the model wouldn’t choose a larger $\hat{\alpha}$ if it had the possibility). We cap the upper bound of the gris at $0.45$ because when the novel class constitutes a large fraction of the target data, novelty detection becomes relatively easier. In such regimes, even simple inspection of random data points or sampling strategies can easily detect novel instances, enabling labeling and the use of alternative solutions. Our primary focus is therefore on the more challenging and practically relevant regime where novel classes are relatively rare (typically between 0.02 and 0.2).
>
> We further confirm this by adding Table 22 in the appendix of the revised draft. We note that this lower bound is not fundamental and may depend on factors such as dataset characteristics, feature representations, and model architecture. In practice, we recommend determining an appropriate $\boldsymbol{\alpha}$ range via simulation: known classes can be temporarily treated as unknown, and their detectability can be evaluated under varying assumed novel class ratios. This procedure provides empirical guidance for both the bottom value and density of the grid range.
>
> > 1. b) how the grid density impacts model performance
>
> We added experiments on the SUN397 dataset using a ResNet50 backbone pretrained on ImageNet-1K (Table 21 and Figure 5 in the revised draft). These results show that CoLOR is performant for a fixed $\boldsymbol{\alpha}$ range with only minor performance variations across varying grid densities. This indicates that the method does not require finely tuned grids to achieve strong performance, and that the chosen density is sufficient to cover the relevant operating regime.

---

> ### Author Response · Authors · 2026-01-11
> **Continuing response to Reviewer 52XA**
>
> > 2. The theoretical analysis is restricted to the simplified k=1 (PU-learning) scenario...
>
> The goal of theory in our work is to mainly understand constrained learning for novelty detection in the overparameterized regime. There are no prior theoretical results in this regime (including the PU-learning literature) even in the binary (k=1) case particularly under background shift. Hence, the k=1 setting already captures several novel and technically challenging aspects from choosing an adequate formal setting to the resulting proofs. Extending these guarantees to multi-class (k>1) case would involve additional challenges related to inter-class interactions and optimization dynamics which we believe is beyond the scope of the present work. Existing works in this area (e.g., [1,2,3]) largely focus on characterizing common properties of different learning rules rather than providing performance guarantees.
>
> We do not claim that our theoretical guarantees directly carry over to multi-class case in a rigorous sense. Rather, the theory is meant to provide insight into why constrained learning can outperform domain-discriminator-based approaches under background shift and with overparameterized models. These mechanisms, such as how constraints affect margin allocation and generalization to minority subgroups, are not inherently tied to the binary setting and help explain the empirical behavior observed in the multi-class experiments. Extending our analysis to the multi-class setting under background shift is an interesting and important direction for future work.
>
>  From a practical perspective, scalability and support for multiple classes are central to our algorithmic and empirical contributions, which is why our experiments focus on realistic k > 1 settings. Empirically, we do not observe qualitatively different behavior between binary and multi-class cases with respect to the main phenomena studied (e.g., robustness to background shift and sensitivity to novel class size), which motivated us to concentrate the theoretical analysis on the simplest setting that still exhibits these effects.
>
>
> [1] Daniely, A. and Shalev-Shwartz, S., 2014, May. Optimal learners for multiclass problems. In Conference on Learning Theory (pp. 287-316). PMLR.
>
> [2] N. Brukhim, D. Carmon, I. Dinur, S. Moran and A. Yehudayoff, "A Characterization of Multiclass Learnability," 2022 IEEE 63rd Annual Symposium on Foundations of Computer Science (FOCS), Denver, CO, USA, 2022, pp. 943-955, doi: 10.1109/FOCS54457.2022.00093.
>
> [3] K. Wang, V. Muthukumar and C. Thrampoulidis, "Benign Overfitting in Multiclass Classification: All Roads Lead to Interpolation," in IEEE Transactions on Information Theory, vol. 69, no. 12, pp. 7909-7952, Dec. 2023, doi: 10.1109/TIT.2023.3320098.

---

> ### Author Response · Authors · 2026-01-11
> **Continuing response to Reviewer 52XA**
>
> > 3. No ablation study is conducted to isolate the contribution of the shared representation architecture in CoLOR...
>
> It is helpful to distinguish between two orthogonal design choices in our method, the optimization strategy and the architectural choice. Using a shared encoder is an architectural decision, whereas constrained learning determines how the two objectives for novelty detection and closed-set classification are optimized. Importantly, even if one were to use a shared encoder without constrained learning, some optimization rule would still be required to jointly learn the two tasks. The most direct alternative in this case is to replace constrained learning with a domain-discrimination objective while retaining a shared representation. This corresponds exactly to the DD baseline, which is included throughout our experiments (Tables 1–4, 6–14). These comparisons also include other learning rules that use shared representations, such as uPU, nnPU, and PULSE, allowing us to evaluate the effect of constrained learning beyond the choice of encoder sharing.
>
> Conversely, isolating the constrained learning rule while removing shared representations is not straightforward, since it is unclear what a meaningful alternative to a shared encoder would be in this setting. To further clarify the contribution of joint learning with a shared encoder, we added additional ablation studies in Tables 23 and 24 of the revised draft. These experiments compare three variants, (1) Source-only, which uses only closed-set classification heads;
> (2) ConOp, which uses only novelty detection heads trained via constrained optimization; and (3) CoLOR, which jointly trains both heads with a shared encoder.
>
> Across both ResNet50 and CLIP ViT-L/14 backbones, these results show that CoLOR consistently improves both novelty detection performance and closed-set classification accuracy relative to training either component in isolation. As expected, only CoLOR supports evaluation on the full open-set task (e.g., OSCR), since it is the only model that jointly addresses both objectives. Taken together, these results suggest that the performance gains arise from the interaction between constrained optimization and joint representation learning, rather than from either component alone.
>
> > 4. The paper does not systematically evaluate CoLOR’s robustness when the separability assumption (Assumption 1) is severely violated...
>
> We ran a simulated study to address the reviewer’s comments (Figure 6 and Tables 25, 26). In this study, we consider a simple Gaussian mixture setup following the theoretical setup defined in definition 2 of the main draft and vary the separability between known and novel classes. Further details can be found in appendix section A.8 of the revised draft.
> Separability is an inherent property of a dataset and is not something we can easily control in real-world benchmarks. A simulated setup is appealing in this case because, unlike real-world data, it allows us to precisely control violations of separability without changing any other aspect of the data. This complements our experiments on real-world datasets, where natural variation across datasets already demonstrates performance under different separability conditions.

---

### Review · Reviewer_Tqac · 2026-01-12

**Summary Of Contributions:**

The authors present a new method for Open-Set Domain Adaptation (OSDA), dealing with domain shift and the emergence of a new, separable class. Their method, CoLOR (Constrained Learning for Open-set Recognition), uses a constrained loss which tries to minimize the loss of the original distribution and the false positive rate, while maintaining a bound on the novel class proportions.

The authors analyze a simplified model of Gaussian mixture data classified via linear regression, and show that in this setting their method gives provable learning guarantees. They then describe the implementation of their method, which uses a Lagrangian formulation combined with a grid search over the candidate values of $\alpha$. This allows for training over all these $\alpha$ values simultaneously for a more efficient algorithm. The authors present positive results on SUN397, CIFAR100, and an Amazon reviews dataset.

The main strengths of the paper are the simplicity of the ideas, and the extensive experiments. The weaknesses lie in the theoretical motivation, as well as the underexplored nature of the $\alpha$ grid search which seems to be the most novel contribution here.

**Audience:**

Yes

**Audience Explanation:**

My answer here is a conditional yes; I think with some improvements to the theorem and the experiments the paper should be of interest.

I am not an expert in this area, so I will defer to other reviewers on this point.

**Claims And Evidence:**

No

**Claims Explanation:**

The theorems seem correct overall, but it is not clear to me that they are in a regime that's relevant for the real world setting --- which is troubling given that the written problem is simpler than real-world problems. In particular, the novel class proportion $\alpha$ is constrained to be $\alpha<1/1024$ --- while in the experiments, $\alpha$ is as high as $1/20$. It is also troubling that $N_{\mathcal{S}} = N_{\mathcal{T}}$ for the theorem to hold --- while I can imagine some condition necessary, this feels too strong.

The overall experimental results seem good, though I am not an expert in these benchmarks. The main missing feature to me is analysis in the main text of how the choices of number of grid points affects the method, and how this varies as a function of $L$. The biggest concern with the method is that it becomes computationally infeasible for larger numbers of classes (both pre-existing and novel).

**Requested Changes:**

The biggest changes requested are:

1. Reformulation of theorem 1 for larger $\alpha$ values, and a softening of the constraint $N_{\mathcal{S}} = N_{\mathcal{T}}$.

2. Analysis of the effectiveness of the method as the number of $\hat{\alpha}$ and $L$ are varied.

With these changes the paper can secure my recommendation.

Small comments:

In section 4.2, it would be helpful if the authors bolded the direction vectors $\mu$ and $\eta$, to make it clear that they live in the same space as $w$ and $x$.

For the theorem, why does $\alpha$ have to be so small? This suggests that the theorem does not have much power.

In (3), $\hat{\alpha}$ is used twice in an inequality; this is confusing as one of the $\hat{\alpha}$ values is a fixed constant, while the other is a property of the model. Please change this notation.

When presenting (3), it would be good to reiterate the definitions of $\alpha$ and $\beta$ either in $(3)$ or just before it. Some readers may want to start with the main method, or refer back to the paper as a reference. It would be ideal for the main loss description to stand on its own.

Table 4: I'm not sure why color results are bolded, they are not beating all other methods here especially when taking error bars into account.

FYI the authors accidentally unblinded themselves with the acknowledgments.

---

> ### Author Response · Authors · 2026-01-23
> **Addressing the Reviewer Tqac's comments**
>
> We appreciate the constructive comments provided by the reviewer which we address below:
> > The biggest concern with the method is that it becomes computationally infeasible for larger numbers of classes (both pre-existing and novel).
>
> The Open-Set Domain Adaptation (OSDA) problem generally involves classification of known (pre-existing) classes while also identifying samples that belong to any of the unknown classes typically by grouping them into a single unknown/novel category. Hence, the number of novel class types does not impact the computational complexity of the method, since discrimination among them is out of scope for OSDA. With increasing pre-existing classes, the method is as computationally expensive as any other supervised classifier needed to solve the problem. Hence, the cost due to the number of existing classes is unavoidable.
>
> The main additional cost comes from the added prediction heads for the novel class with different values of $\hat{\alpha}$. However, there are several knobs we can use to control this cost. Density of the search grid, $\boldsymbol{\alpha}$, is one of them (e.g. starting from a sparse grid and densifying it), and the expressivity of each prediction head for an alpha value is another. Generally, having a few more linear prediction heads was not a bottleneck in our experiments with deep architectures, but could require some minor tweaks if users wish to extend the method to vastly different regimes.
>
> > Reformulation of theorem 1 for larger $\alpha$ values, and a softening of the constraint $N_{\mathcal{S}} = N_{\mathcal{T}}$.
>
> We thank the reviewer for the suggestion on enhancing the main theorem.
> * We have updated the theorem in the revised draft with relaxed constraints on $\alpha,N_\mathcal{S}$ and $N_\mathcal{T}$.
> * The range of alpha is an artifact of the proof technique, where certain ranges of parameters are traded off to express the regime where performance gaps occur. The updated theorem statement emphasizes this flexibility in setting $\alpha$ while changing other quantities.
> * The theorem is meant to show the **existence** of a range of parameters where an **extreme** gap in performance is observed. In practice, the gap may be smaller yet still significant. This is demonstrated by our experiments, while the theory aims to explain the underlying mechanism and how crucial it can be.
>
> The theorem depicts the existence of a very extreme situation, where DD completely fails (AU-ROC smaller than 0.5), while our method achieves a very high accuracy (AU-ROC > 0.9), defined by certain relations between feature norms, sizes of groups, dimensions and noise. To show that this could theoretically happen, we intentionally stretched the bounds on problem parameters accordingly. Whether these ranges describe real problems is a different question, and it is likely that in order to have this dramatic performance gap some conditions need to hold which do not apply to some common real-world settings. However, the empirical results show that there is still a significant performance gap (even though it is not as extreme as the example in the theorem). So the theorem is primarily a didactic tool rather than a mathematical characterization of a typical dataset.
>
> Accordingly, the rigid constraint on $\alpha, N_\mathcal{S}$ or $N_{\mathcal{T}}$ is just a starting point to fix the range of one parameter of the problem without dependence on all the others. If we want a different range, we can propagate it through the inequalities and get other forms of constraints.
>
> Presumably, an alternate statement with relaxed constraint $\alpha\leq \frac{N}{4N_\mathcal{T}}$ can be derived by demanding a stronger bound on $\frac{r_\eta}{r_\mu}\leq \frac{1}{4N}$ to operate in the same regime of large performance gap. Similarly, we can also relax the constraints on the AUROC of both the methods thereby reducing their performance gap, which will allow the problem constraints to be more relaxed and closer to what we observe in practice. Hence, it doesn’t mean that the existing theory is powerless for lesser extreme and realistic regimes.
>
> > Analysis of the effectiveness of the method as the number of \hat{\alpha} and L are varied.
>
> To address this and the suggestions by reviewer 52XA, we have added Figure 5 and Table 21 in the appendix to evaluate the impact of grid density i.e. $L$ (number of $\hat{\alpha}$) on the model performance for a certain range of $\boldsymbol{\alpha}$. We see that the model is performant for a varying number of candidates $\hat{\alpha}$ as long as the search grid includes the values close to the true novel class size. We have further measured the impact of search grid range in Table 22 on the model performance.

---

> ### Author Response · Authors · 2026-01-23
> **Continuing response to Reviewer Tqac**
>
> Small comments:
>
> > In section 4.2, it would be helpful if the authors bolded the direction vectors \mu and \eta
>
> We thank the reviewer for the suggestion. We have bolded the direction vectors throughout the paper in the revised draft.
>
> > For the theorem, why does $\alpha$ have to be so small? This suggests that the theorem does not have much power.
>
> We have updated the theorem in the revised draft to relax the upper bound on $\alpha$. As mentioned in our detailed response above, we can further relax the constraint on $\alpha$ by propagating through the inequalities and demanding a stronger bound on $\frac{r_\eta}{r_\mu}\leq \frac{1}{4N}$ to operate in the same regime of large performance gap. Similarly, we can also allow smaller gap between AUROC of both the methods to operate in a more realistic setting which would enable relaxed constraints on $\alpha$ and other problem parameters.
>
> > In (3) $\hat{\alpha}$, is used twice in an inequality; this is confusing as one of the values is a fixed constant, while the other is a property of the model. Please change this notation.
>
> Thank you for the suggestion. We have changed the notation in the revised draft to avoid confusion (replaced $\hat{\alpha}(h)$ with $\hat{\alpha}_{emp}(h)$).
>
> > When presenting (3), it would be good to reiterate the definitions of $\alpha$ and $\beta$...
>
> Thank you for the suggestion. We have reiterated the definitions just before eq. (3) so that the main loss description can stand on its own.
>
> > Table 4: I'm not sure why color results are bolded, ...
>
> Thanks for pointing this out. We have corrected the highlights in the table.
>
> > FYI the authors accidentally unblinded themselves with the acknowledgments.
>
> We have anonymized the acknowledgements.

---

> > ### Comment · Reviewer_Tqac · 2026-02-24
> > **Response to revisions**
> >
> > I thank the authors for their revisions.
> >
> > Most of my criticisms have been addressed, but there is still the overall question of the theorem. I have some followup questions.
> >
> > The overall issue with $\alpha$ is still troubling. Following the author's arguments in responses, I further examined the SNR bound represented by the ratios of the $r$ values. If I understand correctly, increasing this SNR will increase the allowable $\alpha$; I think this scaling should be part of the theorem as it can provide important intuition for the practical setting.
> >
> > However, now I am very concerned with the SNR bound. If I understand correctly, the theorem requires STRONGER SNR as the dataset size increases. This is in contrast to typical behavior in machine learning where more data allows one to resolve WEAKER SNR! My guess is that the culprit lies in the assumptions relating $d$ and $N$, but this needs to be expanded on. Also, the choice $d>N$ is justified by the idea that "the models are expressive and can interpolate the data". I think this is backwards; the scenario presented represents a case where there is insufficient data to solve the machine learning problem. Indeed here the model size is directly proportional to the data complexity, unlike the usual understanding of interpolation where the data complexity is fixed but the model capacity increases to surpass it.
> >
> > To me this points to some issue of the interpretation and applicability of the theorem to the methods presented in the paper. I'm happy to hear more from the authors on this point; I consider the rest of my concerns sufficiently addressed. In its current state I think the paper would be stronger without the theorem.

---

> ### Author Response · Authors · 2026-03-02
> **Response to reviewer Tqac's follow-up**
>
> We first summarize our main points, and then elaborate on each below:
>
> 1. We first **provide a heuristic overview of the theorem** below and clarify the terminologies to ensure we are on the same page as the reviewer.
> 2. **Then we address specific reviewer’s comments.** If we have correctly understood the reviewer’s concerns, **our theoretical claim appears to be consistent with their intuition**. We would appreciate clarification on any aspects where our understanding and the reviewer’s perspective may differ. We further propose revising the manuscript to make these clarifications more explicit in the main text, while moving some of the more technical details to the appendix.
> 3. Finally, **we elaborate on the consistency of our theoretical claims and assumptions with relevant prior works** that operate in a similar setting while studying benign overfitting. We believe this broader perspective on classification with minority subpopulations is useful, as our result shares structural similarities with these lines of research. The key differences lie in the type of algorithms that can solve the problem and in the technical steps required for the proof, although the overall behavior is closely related to previous works [1,2].
>
> ## **Comments on SNR and Relation to Dataset Size**
> To ensure alignment in terminology, we would like to clarify that the problem involves two distinct notions of signal-to-noise ratio (SNR).
> * **Core-to-spurious signal ratio**.
> The first quantity, referred to by the reviewer, is $r_{\eta} / r_{\mu}$. We will call this the _core-to-spurious ratio_, drawing on parallels to the literature on spurious correlations discussed later in our response. This ratio measures how much easier it is to classify using the feature $\mu$ versus using the feature $\eta$. In particular, a small value indicates that classification based on $\mu$ is easier, which can lead to the undesirable behavior of ranking non-novel examples as more likely to be novel than truly novel examples. In contrast, relying on $\eta$ corresponds to the desired solution.
> * **Signal-to-ambient-noise ratio.**
> The second notion of SNR is governed by the variance parameter $\sigma^2$ (and its relationship to $r_{\mu}$ and $r_{\eta}$). This quantity determines how easily the features $\mu$ and $\eta$ can be distinguished from ambient Gaussian noise with variance $\sigma^2$. This SNR increases with $r_{\eta}$ and $r_{\mu}$, and decreases as $\sigma$ increases (recall $\sigma = 1/\sqrt{d}$).
>
> ## **Interpretation of the theorem**
> **The theorem works in "benign overfitting" regimes** [3]. That is, where the _signal-to-ambient-noise ratio_ is sufficiently high so classifiers such as max-margin or CoLOR do not overfit to the ambient noise. They rely mostly on $\eta$ and $\mu$ for classification.
>
> However, **the central question in our problem setup is not whether the features can be detected, but rather which feature a given learning rule prefers**. As the problem parameters vary, different algorithms may prioritize either the desirable feature $\eta$ or the spurious feature $\mu$, leading to different generalization behavior, particularly in novelty detection.
>
> Theorem 1 characterizes difficult problem regimes by specifying conditions on key parameters, including the proportion of novel classes ($\alpha$), the magnitudes of the spurious and core signals ($r_{\mu}$ and $ r_{\eta}$), and the dimensional setting $d \ge N$, corresponding to an overparameterized regime. These conditions delineate when the learning problem becomes challenging in the sense that algorithmic preference between $\eta$ and $\mu$ determines success or failure.
>
> The theorem identifies a regime in which **CoLOR succeeds while a baseline fails by a large performance gap**. This regime is characterized by:
> * the novel class proportion ( $\alpha$ ) is small,
> * a small core-to-spurious ratio (i.e. a weak core signal)
>
> Moreover, this behavior is not restricted to the overparameterized regime; it can also arise under favorable conditions such as abundant data provided there is sufficient support overlap among known classes and adequate separability between known and novel samples. **Under such data-rich conditions, however, the performance gap between algorithms tends to diminish as the problem becomes inherently easier.**
>
>
> **With this clarification in mind, we would like to better understand the reviewer’s concern as we address specific comments in the continued response below:**

---

> ### Author Response · Authors · 2026-03-02
> **Continued response to reviewer Tqac**
>
> > … increasing this SNR will increase the allowable ( \alpha ) …
>
> If we have interpreted this correctly, the reviewer is referring to the inequality
> $
> \frac{r_{\eta}}{r_{\mu}} \le \frac{4}{N}
> $
> together with the condition
> $
> \alpha \in \left(0, \frac{N}{1024N_{\mathcal T}}\right)
> $
> in the theorem. The reviewer’s point appears to be that relaxing the bound on the _core-to-spurious ratio_ (equivalently, decreasing $N$) increases the upper bound on $\alpha$.
>
> **Whether this change actually affects the admissible range of $\alpha$ depends on how the source and target dataset sizes scale**. This dependence is already reflected in the theorem statement. For example, if the target sample size ( $N_{\mathcal T}$ ) decreases proportionally with $N$ , or equivalently if the source sample size ( $N_{\mathcal S}$ ) increases at the same rate, then the upper bound on $ \alpha$ remains unchanged.
>
> We also note that the phrase “allowable $\alpha$” may be interpreted as suggesting that the problem becomes unsolvable beyond this bound. This is not the case. In fact, larger values of $\alpha$ make the problem easier and tend to reduce the performance gap between the two algorithms, making the setting less informative for distinguishing their behavior.
>
> **Clarification request.**
>
> Could we confirm whether this interpretation matches the reviewer’s intended concern? We would be very happy to discuss this further and clarify any remaining points.
>
> > … the theorem requires a stronger SNR as the dataset size increases …
>
> Here “SNR” corresponds to  what we called above the _core-to-spurious ratio_, then we note that the theorem imposes an upper bound inversely proportional to dataset size ( $N$ ). Consequently, as $N$ increases, the range of parameters where the two algorithms exhibit sharply different performance becomes smaller. In other words, **with more data, the distinction between the algorithms is less pronounced**. To the best of our understanding, **this aligns with the reviewer’s intuition**. If this interpretation is correct, we would be happy to emphasize this point more clearly in the revision.
>
> > … this contrasts with typical ML behavior where more data resolves weaker SNR …
>
> In fact, **our result is consistent with this principle**. As dataset size increases, the upper bound on the _core-to-spurious ratio_ decreases, meaning the _hard regime_ shrinks. If we raise the _core-to-spurious ratio_, e.g. by raising $r_{\eta}$ and fixing all other parameters, then CoLOR as well as the baselines will see improved performance, which may reduce their performance gaps. The theorem does not state that directly as it does not fall under the hard problem regime we are interested in solving for, but we are happy to add this clarification.
>
> **Clarification request.**
>
> Could we confirm whether this interpretation matches the reviewer’s intended concern? We would be glad to refine the exposition further to make this relationship clearer.
>
> > … the choice $d>N$ is backwards... unlike the usual understanding of interpolation where the data complexity is fixed but the model capacity increases to surpass it.
>
> We focus on the $d \ge N$ regime to model high-capacity networks that interpolate the training data. The _signal-to-ambient-noise ratio_ is set high so classifiers rely on $\eta$ and $\mu$ rather than overfitting the noise. With appropriate choices of $\sigma$ and bounds on $ r_{\eta}, r_{\mu}$, this places us in the benign overfitting regime [1,2], allowing us to study how different learning rules preferentially rely on either the desirable feature $\eta$ or the spurious feature $ \mu $, and the resulting differences in generalization, particularly for novelty detection.

---

> > ### Author Response · Authors · 2026-03-02
> > **Continued response to reviewer Tqac**
> >
> > ## **Consistency with Prior Work**
> >
> > **We note that our setup closely parallels prior theoretical studies on spurious correlations and overparameterization**. For example, [1] derive analytic results that impose a strict constraint on minority group size. In particular, their Theorem 1 assumes
> > $
> > p_{\text{maj}} \ge 1 - \tfrac{1}{2001}
> > \quad\text{(equivalently, } p_{\text{min}} \le \tfrac{1}{2001}\text{).}
> > $
> >
> > In our framework, the minority group corresponds to the novel class, while the majority group corresponds to the non-novel target examples.
> >
> > Under this correspondence, their parameters ($ \sigma^2_{\text{core}}$ ) and ( $\sigma^2_{\text{spu}}$ ) roughly align with the inverses of our signal strengths ( $r_{\eta}$ ) and ( $r_{\mu}$ ), respectively. Consequently, their analysis also exhibits a regime in which an effective signal-to-noise quantity decreases as dataset size grows. The exact scaling rates are not directly comparable, for example, they fix ( d = N + 2 ), and the problem formulations differ, but the qualitative phenomenon is similar.
> >
> > Both works characterize families of problem instances that become progressively more difficult in the sense that the core signal becomes harder to detect. They then show that max-margin classifiers fail in these regimes. As is standard in this literature, these theoretical predictions are complemented by empirical experiments demonstrating the phenomenon on real data, where it remains unclear how faithfully the simplified mathematical model captures practice.
> >
> > ## **Distinguishing Contribution**
> > A second key aspect of our theorem is that it identifies a separation between algorithms: we show that one method (CoLOR) succeeds over a strictly broader class of problem regimes than max-margin or domain-discriminator approaches. Results of this type, demonstrating both failure modes and separations between algorithms under sample-size-dependent SNR conditions, also appear in related work; see, for example [2].
> >
> >
> > ## **References** (as cited)
> >
> > [1] Sagawa, Raghunathan, Koh, Liang. An investigation of why overparameterization exacerbates spurious correlations.ICML 2020.
> >
> > [2] Wald, Yona, Shalit, Carmon. Malign Overfitting: Interpolation and Invariance are Fundamentally at Odds. NeurIPS Workshop, 2022.
> >
> > [3] Bartlett, Long, Lugosi, Tsigler. Benign overfitting in linear regression. PNAS, 2020.

---

> ### Author Response · Authors · 2026-03-14
> **Following up with reviewer Tqac**
>
> We appreciate the reviewer's thoughtful follow-up and have tried to address the questions raised about $\alpha$, the ratio of the $r$ values, and the connection to prior work. In our response, we also included a few clarification questions to ensure that we interpreted the reviewer's concerns correctly.
>
> We wanted to check whether our response sufficiently addresses the concerns. We would be very happy to clarify any remaining questions regarding the theoretical insights.
>
> Thank you again for your time and feedback.

---

### Review · Reviewer_eCWt · 2026-02-12

**Summary Of Contributions:**

This paper introduce a new method call CoLOR for Open-Set Domain Adaptation (OSDA). OSDA is a part of DA field where there are novel labels in the target domain.

In this paper, the authors propose a new pipeline to train model to efficiently be robust to new labels in the target dataset. With proper assumptions, they show the efficiency of  systematic identification of the novel class. Extensive experiments are proposed to show the robustness of CoLOR over dataset and model.


Weaknesses:
- My main concern on this paper is the lack of clarity. It is really hard to follow the challenges, the contributions and the experiments.
- The paper is missing of notation section that could help the reader to follow.
- The definition of the shift between domain is not clear. The authors introduce Background shift but in my opinion different definition is given. First the Figure 1 is illustrating a label shift problem where the class proportions are different. In the paragraph "background shift" they define the shift as inclusion of the support of Target distribution to the support of Source distribution. It seems to be a covariate shift ?
And then they define latter in the paper, a conditional shift. See Skada-bench [1] where shifts are properly defined.
- The method part, which I suppose is the section 5, is introducing CoLOR, but some model/loss for color already been introduced before which is misleading.
- Introduction of each big part directly go to notation.
- COLOR methods has not been introduces and the paper is already introducing specific model or loss for it,. Part 4

Small weakness:
- Figure 1 lacks of clarity. Most of the image is the bar chart and it is hard to read text and image without zooming.
- Architecture $h$ is not defines properly: $h(x) = \phi \circ w(x)$ to $h(x) = w^s \circ \phi (x)$

[1] SKADA-Bench: Benchmarking Unsupervised Domain Adaptation Methods with Realistic Validation On Diverse Modalities, Lalou et. al.,  2025

**Audience:**

No

**Audience Explanation:**

If OSDA is an interesting problem that can be beneficial for TMLR audience, this paper lacks of positioning. It is then hard to know in which case the paper could be helpful or not. As mentioned above, having a better structure of the paper, better positioning and more clarity will definitely enhance the paper and interest TMLR audience.

**Broader Impact Concerns:**

No concern

**Claims And Evidence:**

No

**Claims Explanation:**

The big weakness of the paper is the lack of clarity as mentioned above. This leads to an imprecise and non convincing paper.

The paper could be way more convincing with a work on the writing:
- Add a notation section.
- Have a good separation between section. It is better to introduce the section and then introduce any notation. Section 4 and Section 5 start directly with definition of model without any contextualization.
- CoLOR will need a better section to introduce all this component.
- Maybe introducing CoLOR than propose the theoretical background will help the understanding. Right now the assumptions made at the beginning is misleading for the next.

**Requested Changes:**

See above section

---

> ### Author Response · Authors · 2026-02-20
> **Addressing Reviewer eCWt's comments**
>
> We thank the reviewer for their feedback and address all the concerns below:
>
> > My main concern on this paper is the lack of clarity...
>
> **We have introduced several changes to the paper based on the reviewer’s comments**, in order to improve clarity. Please see the list of changes in our response below.
>
> > The paper is missing of notation section...
>
> We appreciate the constructive feedback. **We added a notations section in Appendix A.1** (Table 5) and is referenced in the main paper.
>
> > Add a notation section
>
> As addressed in our previous response, **we have added it in appendix section A.1** (Table 5) and referenced it in the main paper.
>
> > Introduction of each big part directly go to notation.
>
> We hope that the **new organization in the revised draft addresses this concern**.
>
> > Figure 1 lacks of clarity...
>
> **We have updated the figure**. We appreciate the feedback.
>
> **Definition of Background shift**
>
> Next we address the reviewer’s comment about the definition of background shift. In the revision, we elucidated it and included it in definition 1 to highlight the formal definition. We introduced clarifications in key points of the paper, which prevent misunderstandings of the setting by future readers. Below we expand on each part of the reviewer’s comment.
> > The definition of the shift between domain is not clear…
>
> **A background shift is an umbrella term for any distribution shift among the known classes**, i.e. between $P_{\mathcal{S}}$ and $P_{\mathcal{T}, [k]}$. This includes shifts like covariate, label and conditional as special cases (consistent with the taxonomy in SKADA-Bench [18]). We call this “background shift” instead of generically referring to it as a “shift”, to emphasize that this shift occurs in addition to the emergence of novel class ($P_{\mathcal{T}, k+1}$). Both $P_{\mathcal{T}, [k]}$ and $P_{\mathcal{T}, k+1}$ constitute the entire shift between source $P_{\mathcal{S}}$ and target $P_{\mathcal{T}}$. Following the reviewer’s comments, **we’ve stated these naming considerations explicitly in the paper and related them to definitions in the Skada-bench reference proposed by the reviewer.**
>
> > they define the shift as inclusion of the support, so it seems to be covariate shift...
>
> **Inclusion of supports alone does not imply covariate shift**, since it still allows $P_{\mathcal{S}}(Y | X)$ to change, which is not the case under covariate shift. Support overlap is simply an identifiability assumption needed to make OSDA solvable.
> However we see why including this as part of the definition of background shift, instead of an added assumption, may be confusing and we have separated it from the definition in the revised draft. **We further added appropriate clarifications to the theory and introduction sections**.
>
> > First the Figure 1 is illustrating a label shift problem.
>
> TL;DR
>
> **Figure 1 is not illustrating a label shift problem**. While known class proportions indeed change, the shift is driven by changes in latent subtypes within each known class (unobserved by the learner). Therefore the example in this figure does not fall squarely into one of the buckets covariate/label/conditional shift. **The revised caption in the new draft explains the naming conventions mentioned above and describes Figure 1 more carefully**.
>
> Longer explanation:
>
> The example is meant to illustrate one plausible real-world scenario that motivates our work. Here, ‘shopping’ and ‘workplace’ are the known classes, and they have subtypes (e.g. ‘bakery shop’, ‘biology lab’ respectively). **The shift is in the proportion of subtypes, but the subtypes are unknown to the learner**. Therefore, in terms of the distribution over observed variables, $X,Y$, the distribution $P_{\mathcal{S}}(Y)$ changes, but also $P_{\mathcal{S}}(X | Y)$ does. To see this explicitly, we can denote subtypes by a latent variable $Z$ and write the joint on $X,Y$ as $P_{\mathcal{S}}(X=x,Y=y) = \sum_{z}P_{\mathcal{S}}(Z=z)P(Y=y \mid Z=z)P(X=x | Z=z, Y=y)$. If we let $P_{\mathcal{S}}(Z=z)$ shift, then this does not reduce to a single covariate/label/conditional shift on $P_{\mathcal{S}}(X, Y)$. To solve the most general problem we can, we do not reduce our formalism just to this type of shift, and state our main assumption as separability and overlap.
>
> If additional structure about the shift is known (e.g., that it is conditional shift), specialized UDA methods could be combined with CoLOR for further gains. Exploring such combinations is beyond the scope of this work, which focuses on the open-set aspect of OSDA, but it is a promising direction for future research.
>
> > Architecture $h$ is not defines properly: $h(x) = \phi \circ w(x)$ to $h(x) = w^s \circ \phi (x)$
>
> We have corrected the definition in the revised draft. Thank you for the feedback.

---

> ### Author Response · Authors · 2026-02-20
> **Continuing response to reviewer eCWt**
>
> >  section 5 is introducing CoLOR, but some model/loss for color already been introduced before...
>
> TL;DR
>
> **CoLOR is not completely formulated before section 5**.  We added a roadmap in the Introduction (Sec. 1) in order to clarify this point, and additional contextualization around Sections 4 and 5 to make the transition from statistical learning rule to fully developed practical method.
> We follow this structure of progressing from a **0–1 loss-based statistical learning rule** (sec. 4, eq. 2 in the revision) → **surrogate optimization objective** (sec. 5, eq. 3) → **scalable practical implementation** (sec. 5.1–5.2, eq. 4, fig. 2, algorithm 1) as it is standard from a statistical learning perspective. We think this addresses the reviewer’s concerns and we would be happy to clarify any follow up concrete questions.
>
> To elaborate: We chose the current structure intentionally for the audience interested in understanding the method in a principled manner.
>
> **Section 3 formally defines the problem** and lays down the necessary assumptions to solve OSDA.
>
> **Section 4 analyzes only the statistical rule** for constrained learning on open-set problems in a simpler setting of $k=1$. Definition 2 provides the theoretical setup using linear Gaussians and introduces synthetic background shifts in it following the assumptions laid out in section 3. In this section, the rules are expressed in terms of 0–1 losses and analyzed as such. Particularly, the constrained learning rule intends to minimize FPR subject to a recall constraint. We cannot directly optimize this in practice using common gradient-based algorithms.
>
> **Section 5 then introduces CoLOR** to specifically tackle OSDA (with $k \geq 1$) under background shift. Eq 3 first proposes a concrete training objective that is used to approximate the statistical learning rule previously introduced. This is similar to how a surrogate loss (like logistic or hinge) is introduced to approximate the 0-1 loss in common treatments of classification problems in learning theory [1,2]. Section 5.1 introduces a scalable architecture that allows us to achieve the training objective in practice (also see Fig. 2). The actual loss is derived after solving the Lagrangian optimization problem in eq 3 and taking its dual. We proposed this in eq 4 and used it consistently to train all the models. Algorithm 1 lists all the steps of CoLOR including the critical criterion of selecting the most precise novelty head.
>
> > COLOR methods has not been introduces and the paper is already introducing specific model or loss for it,. Part 4
>
> TL;DR
>
> **Section 4 does not introduce the loss or model for CoLOR that can be used in practice**. Instead, it analyzes a simplified constrained learning rule, a statistical foundation that CoLOR builds upon, in a stylized setting where $k=1$. Eq 2 introduces the statistical learning rule in terms of **0–1 losses** (minimizing FPR subject to a recall constraint). Section 4.2 studies a linear–Gaussian setup (see **Definition 2**). Eq 2. cannot be optimized directly in practice with common gradient-based algorithms. **We emphasized this around sections 4 and 5 and reorganized them to clarify it further**.
>
> Further elaboration:
>
> **CoLOR (Constrained Learning rule for Open-Set Recognition) itself is introduced fully in section 5 to specifically solve practical OSDA problems with $k \geq 1$ under background shift**. Since CoLOR builds upon this constrained learning rule, we refer to the rule in section 4 as simplified CoLOR. To this end, section 5 first proposes:
> * **a concrete training objective** (Eq 3), that we used to approximately optimize the previously introduced statistical learning rule. This is similar to how a surrogate loss (like logistic or hinge) is introduced to approximate the 0-1 loss in common treatments of classification problems in learning theory [1,2].
> * **a scalable architecture** in sections 5.1 (Fig 2), to practically achieve the training objective which is not possible otherwise.
> * **the actual loss function** in sec 5.2 (Eq 4) used to train the models after solving the Lagrangian optimization problem stated in eq 3 and taking its dual.
> * **Algorithm 1** to clearly lay down the end-to-end steps of CoLOR including the critical novelty head selection heuristic.
> We follow this structure as it is standard from a statistical learning point of view.
>
> > CoLOR will need a better section to introduce all this component.
>
> Please refer to our reponses to the previous 2 comments.
>
> > Have a good separation between section. It is better to introduce the section...
>
> We hope that **the added paper organization and the revised draft** contextualize the sections and **address the concerns** about their purpose. We would like to note again that **section 4 only introduces the underlying statistical learning rule** that CoLOR builds upon. **It does not fully formulate the model or loss functions that can be used in practice**.

---

> ### Author Response · Authors · 2026-02-20
> **Continuing response to reviewer eCWt**
>
> > Maybe introducing CoLOR than propose the theoretical background will help the understanding.
>
> We appreciate the suggestion. To address the reviewer’s concern, **we have added a clear paper roadmap in the introduction to make this progression explicit and around Sections 4–5 to provide more contextualization**. If the reviewer still thinks that an alternative structure is preferable and other reviewers recommend it as well, then we are happy to reconsider.
>
> We chose the current structure intentionally for the audience interested in understanding the method in a principled manner: **Section 4 first introduces and theoretically analyzes the underlying statistical constrained learning rule** (“simplified CoLOR”) in a stylized $k=1$ linear–Gaussian setting to clarify _when_ and _why_ constrained learning outperforms standard domain-discriminator baselines under background shift. These theoretical insights directly motivate the practical design choices and are empirically validated in Section 6.
>
> **CoLOR itself is introduced in Section 5**, where we translate the statistical rule into a scalable OSDA method for $k \geq 1$, including the surrogate training objective (Eq. 3), efficient and scalable architecture (Sec. 5.1, Fig. 2), final loss (Eq. 4), and full procedure (Algorithm 1).
>
> > If OSDA is an interesting problem that can be beneficial for TMLR audience…
>
> **If the reviewer is implying that OSDA may not be relevant to TMLR audience**, then we quickly highlight that **a quick skim of last year’s TMLR/NeurIPS/ICML/ICLR raises 13 papers on OSDA/Open-Set Recognition** [5-17]. This number rises far higher when expanding the search to related settings such as universal domain adaptation or recognizing novel categories in continual learning where our theory and method may be beneficial as well. Besides this, there are hundreds of papers on this problem in recent years.
>
> > … this paper lacks of positioning.
>
> We are happy to address any concrete questions reviewers may have regarding positioning. To clarify, the paper is positioned as follows:
> * **Underexplored aspects of OSDA**. In the Introduction (3rd paragraph), we explicitly identify gaps in prior work, including the lack of formal characterization of background shift and limited theoretical guarantees in OSDA.
> * **Clear statement of contributions**. The core technical and empirical contributions are summarized towards the end of the Introduction.
> * **Solution to solve OSDA under background shift**. We propose a method with formal guarantees for settings involving background shifts that have not previously been characterized theoretically.
> * **Extensive empirical validation**. In Section 6, we structure experiments around three explicit research questions that prior OSDA methods have not systematically investigated.
> * **Actionable insights**. Our theoretical and empirical results directly answer these questions and provide guidance on _when_ CoLOR is expected to outperform existing methods.
>
> >  It is then hard to know in which case the paper could be helpful or not…
>
> Please see the list of changes we introduced to address comments on clarity and structure of the paper. We hope that these clarify why we hold that **providing results and analysis on when our method is helpful is a crucial part of our contributions, and goes beyond existing work on OSDA**. Our theory and experiments give very strong intuition on the cases where the proposed method is beneficial. Please see them below where we expand on each one. We would be very grateful for any concrete suggestions from the reviewer about the type of characterization they would find helpful here.
>  * **OSDA under support overlap and separability (assumption 1).** This condition is not necessarily specific to our method, as we do not know of other methods that work under weaker guarantees and can treat the general set of shifts that we tackle. Lemma 1 further shows that weaker conditions previously proposed for the narrower label-shift setting do not suffice under background shift, highlighting the necessity of our assumptions.
> * **When the proportion of the novel class is small and the shift is large.**
> Our main theoretical result (**Theorem 1**) characterizes parameter regimes where constrained statistical rule (simplified CoLOR) can achieve **arbitrarily better AUROC** than the standard domain-discriminator baseline, particularly when the novel class is rare and the shift among known classes is substantial. These insights are validated empirically in **Figure 3(b)** (varying novel class size) and additional appendix results (Table 19) studying the effect of shift magnitude. In figure 6, we further study the impact of separability between novel classes and known ones in the synthetic setup.
>
> We hope these revisions make the scope and applicability of our method clearer, and we welcome any further concrete suggestions on what additional characterization would be most helpful.

---

> ### Author Response · Authors · 2026-02-20
> **Continuing response to reviewer eCWt**
>
> > Right now the assumptions made at the beginning is misleading for the next.
>
> We are not sure why the reviewer finds the assumptions introduced in section 3 misleading, **they are relevant to all sections**. They lay the foundation for designing our theoretical problem set up as referenced in section 4.2. It is a common sufficient assumption in related settings for learning with infinite samples [1,2,3,4]. We just show that weaker assumptions are not sufficient for our case (Lemma 1).
>
> If the reviewer can elaborate on what is misleading in this assumption, we would be happy to clarify.
>
> **References:**
>
> [1] S. Shalev-Shwartz and S. Ben-David, Understanding Machine Learning: From Theory to Algorithms. Cambridge University Press, 2014.
>
> [2] P. L. Bartlett, M. I. Jordan, and J. D. McAuliffe, “Convexity, Classification, and Risk Bounds,” Journal of the American Statistical Association, vol. 101, no. 473, pp. 138–156, 2006.
>
> [3] Saurabh Garg, Sivaraman Balakrishnan, and Zachary C. Lipton. Domain adaptation under open set label shift, 2022. URL https://arxiv.org/abs/2207.13048.
>
> [4] Shai Ben David, Tyler Lu, Teresa Luu, and Dávid Pál. Impossibility theorems for domain adaptation. In Proceedings of the Thirteenth International Conference on Artificial Intelligence and Statistics, pp. 129–136.JMLR Workshop and Conference Proceedings, 2010
>
> [5] W. Wang et al., ‘COSDA: Counterfactual-based Susceptibility Risk Framework for Open-Set Domain Adaptation’, in Forty-second International Conference on Machine Learning, 2025.
>
> [6] I. Nejjar, H. Dong, and O. Fink, ‘Recall and Refine: A Simple but Effective Source-free Open- set Domain Adaptation Framework’, Transactions on Machine Learning Research, 2025.
>
> [7] S. Pyakurel and Q. Yu, ‘Learning State-Based Node Representations from a Class Hierarchy for Fine-Grained Open-Set Detection’, in Forty-second International Conference on Machine Learning, 2025.
>
> [8] A. K. Kundu, V. S. Patil, and J. JaJa, ‘Boosting Open Set Recognition Performance through Modulated Representation Learning’, in The Fourteenth International Conference on Learning Representations, 2026.
>
> [9] X. Chen et al., ‘Beyond the Known: An Unknown-Aware Large Language Model for Open-Set Text Classification’, in The Fourteenth International Conference on Learning Representations, 2026.
>
> [10] M. Sreenivas and S. Biswas, ‘Efficient Open Set Single Image Test Time Adaptation of Vision Language Models’, Transactions on Machine Learning Research, 2025.
>
> [11] X. Wang et al., ‘Activate and Adapt: A Two-Stage Framework for Open-Set Model Adaptation’, Transactions on Machine Learning Research, 2025.
>
> [12] H. Liu, ChenyuGuo, Y. Ren, J. Guan, and S. Zhou, ‘FedOpenMatch: Towards Semi-Supervised Federated Learning in Open-Set Environments’, in The Fourteenth International Conference on Learning Representations, 2026.
>
> [13] C. Li, S. Wang, and H. Zhang, ‘Adaptive Gaussian Expansion for On-the-fly Category Discovery’, in The Fourteenth International Conference on Learning Representations, 2026.
>
> [14] W. Feng, S. Zhou, Y. Jiang, and Z. Ge, ‘PRISM: Progressive Robust Learning for Open-World Continual Category Discovery’, in The Fourteenth International Conference on Learning Representations, 2026.
>
> [15] B. Peng et al., ‘OOD-Barrier: Build a Middle-Barrier for Open-Set Single-Image Test Time Adaptation via Vision Language Models’, in The Thirty-ninth Annual Conference on Neural Information Processing Systems, 2025.
>
> [16] Z. Zhong et al., ‘Gains: Fine-grained Federated Domain Adaptation in Open Set’, in The Thirty-ninth Annual Conference on Neural Information Processing Systems, 2025.
>
> [17] H. Dong, E. Chatzi, and O. Fink, ‘Towards Robust Multimodal Open-set Test-time Adaptation via Adaptive Entropy-aware Optimization’, in The Thirteenth International Conference on Learning Representations, 2025.
>
> [18] SKADA-Bench: Benchmarking Unsupervised Domain Adaptation Methods with Realistic Validation On Diverse Modalities, Lalou et. al., 2025

---

> > ### Comment · Reviewer_eCWt · 2026-03-02
> >
> > I thank the authors for taking time to answer all my questions and the effort to re-organized the paper for better reading. I still have some concern about the paper.
> >
> > - For the figure 1, I disagree with you. I understand the caption but visually there is no shift in the images. The only differences between source and target domains are different class proportion (target shift) and new class for target (open-set problem). There is no variation in the distribution of the features between the two domains and except the new class of target, the existing one are exactly the same with exactly the same images as example.
> >
> > - For the definition of background shift, I thank the reviewer for the clarification. So background shift is englobing all the shifts ( covariate, target and conditional shift). The assumption is very strong and knowing exactly which shift we have helps to adapt better. But as you mentioned one can use with CoLOR any other method to tackle a specific shift. Maybe having a proper paragraph saying that will be helpful.
> >
> > - I agree that my point on positioning lacked of details. In the third paragraph you point limits of literature, but in my opinion there very large. As you said, a lot of paper on open-set DA have been published and inside them paper already characterised shift [1], give theoretical insight [2] or look at the effect of new class proportions [3].
> > So the novelty you are pointing is limited. Especially that the background shift is very large and does not really focus on a very specific type of shift, so the characterisation of the shift is not really a strength of the paper. But maybe talking about versatility of CoLORS to different type of shift can emphasise the power of the method. Since it is not design to one specific shift.
> >
> > [1] Shao et. al., Open-set learning under covariate shift, 2022
> > [2] Fang et. al., Open Set Domain Adaptation: Theoretical Bound and Algorithm, 2020, IEEE
> > [3] Busto et. al., Open Set Domain Adaptation, ICCV, 2017

---

> > > ### Author Response · Authors · 2026-03-06
> > > **Response to Follow-up by Reviewer eCWt**
> > >
> > > We thank the reviewer for their follow-up and address the questions below:
> > >
> > > > "For Figure 1, I disagree with you."
> > >
> > > We believe there is a discrepancy between the reviewer's interpretation and what we think the figure conveys. **We have updated the figure and caption to resolve this discrepancy**. Specifically, we highlight that the bar plots are probabilities of latent subtypes within each known class. While the class proportions of known classes (shopping, workplace) may change under background shift, here, it is driven by changes in the proportion of latent subtypes which **changes the input distribution** between source and target datasets.
> > >
> > > **Figure 1 does not convey a pure label shift**. We first refer to the standard mathematical definition of label shift [4,5,6,7], and then provide an intuitive example to clarify the distinction.
> > > Formally, for source distribution $P_{\mathcal S}(x,y)$ and target distribution $P_{\mathcal T}(x,y)$, label shift occurs when
> > > $
> > > P_{\mathcal S}(y) \neq P_{\mathcal T}(y)
> > > $
> > > while the class-conditional distributions remain unchanged:
> > > $
> > > P_{\mathcal S}(x|y) = P_{\mathcal T}(x|y)
> > > $
> > > as defined in [1–4] and related literature.
> > >
> > > **To illustrate this intuitively**, consider a binary image classification task with labels $y \in [\text{dog}, \text{cat}]$. Suppose the source dataset contains mostly husky images for the dog class, whereas the target dataset contains primarily golden retriever images. In this case, the input distribution for the dog class changes between domains, i.e., $P(x|y)$ changes. Therefore, this is not a label shift.
> > >
> > > Our background shift formulation captures such changes in $P(x|y)$ while also allowing the class proportions $P(y)$ to vary between domains. Consequently, background shift encompasses more general forms of distribution change than pure label shift, covariate shift, or conditional shift. The purpose of Figure 1 is to illustrate this broader class of shifts that our method addresses.
> > >
> > > > “Visually there is no shift in the images… There is no variation in the distribution of the features between the two domains.”
> > >
> > > As explained in the previous response, **there is indeed a visual shift between domains**, though it differs from the commonly studied shifts like image corruptions, real-to-synthetic, real-to-clipart. Such shifts have already been extensively studied in domain adaptation literature, with many specialized methods benchmarked on datasets such as Office-31, OfficeHome, and VisDA.
> > >
> > > Instead, our work focuses on more natural yet underexplored shifts, captured by the background shift framework. These shifts arise from changes in the underlying data composition (e.g., variations within individual classes i.e. background context) rather than explicit visual corruptions. Besides images, we further demonstrate this on a text dataset, Amazon Reviews. In this setting, the task is to classify product categories, and the background shift between the source and target domains is defined by semantically positive versus negative reviews.
> > >
> > > We further propose a method that is more robust under these shifts and empirically outperforms existing approaches, including methods designed specifically for feature-level transformations.
> > >
> > > > “The background shift assumption is very strong…”
> > >
> > > We assume that the reviewer meant that this is a very **weak** assumption. As we see it, this means the method is broadly useful, and it is one of its strengths.
> > >
> > > > “Maybe having a proper paragraph saying that will be helpful.”
> > >
> > > **We added section A.6 in appendix In the revised draft to address this**. We have emphasized the possibility of further improvements to better explain the intuition and scope of the background shift assumption, especially when one knows more about the shift. We would like to know if the reviewer has any further concerns here.

---

> ### Author Response · Authors · 2026-03-06
> **Continuing response to reviewer eCWt**
>
> > .. a lot of paper on open-set DA have been published and inside them paper already characterised shift [1] give theoretical insight [2], look at the effect of new class proportions [3]  …So the novelty you are pointing is limited
>
> **Novelty of the learnability analysis:**
>
> Our insights on learnability under background shift are **novel**. While the definition/characterization of this shift is intentionally general and was not itself presented as a primary contribution, **our first contribution is the impossibility result (Lemma 1)**. This is explicitly stated in the second point of our key contributions on page 2. Specifically, we show that learning is impossible under background shift even when assumptions that are sufficient for other shifts (such as label shift) hold. We believe this negative result provides an important and previously unreported insight into the limits of learning under this type of distribution shift.
>
> **Difference from prior theoretical analyses:**
>
> **Our theoretical analysis is fundamentally different from the works referenced by the reviewer.** The existing bounds we are aware of (e.g.,[2,8]) rely on classical generalization analyses that assume limited model capacity. In contrast, our theorem is derived in the interpolation regime, which is the standard framework for studying highly expressive neural networks. Furthermore, **the implications of our results are different**: our analysis compares two learning rules and characterizes when their performance diverges, showing that the gap arises in practically relevant regimes where the novel class is small and the ratio $r_{\eta}/r_{\mu}$ is low. This focus on learning rule comparison and regime characterization is qualitatively different from the generalization bounds presented in prior work.
>
> **Methodological and empirical contributions:**
>
> Beyond theory, we introduce new methodological components that **enable efficient implementation of our approach on flexible architectures, datasets and modalities**. Our empirical study also focuses on an underexplored but natural form of distribution shift, where the distribution of latent label subtypes changes. This induces simultaneous shifts in both $P(Y)$ and $P(X \mid Y)$, as illustrated in Figure 1. We believe that both the formulation of this setting and the empirical findings we report provide useful insights for studying distribution shifts in practice.
>
> **On novelty as an evaluation criterion:**
>
> As a sidenote, **TMLR explicitly discourages using novelty as a primary criterion** for evaluation and instead emphasizes interest and significance. We believe our work satisfies both criteria: it contains novel technical elements (as discussed above) and addresses an interesting and practically relevant setting. The references mentioned by the reviewer study related but distinct questions. Interpreting their existence as negating the novelty of our contributions sets an unusually strict standard, particularly for a venue that does not prioritize novelty as its main evaluation criterion.
>
>
> **References:**
>
> [1] Shao et. al., Open-set learning under covariate shift, 2022
>
> [2] Fang et. al., Open Set Domain Adaptation: Theoretical Bound and Algorithm, 2020, IEEE
>
> [3] Busto et. al., Open Set Domain Adaptation, ICCV, 2017
>
> [4] Moreno-Torres, J. G., Raeder, T., Alaiz-Rodríguez, R., Chawla, N. V., & Herrera, F. (2012). A unifying view on dataset shift in classification. Pattern Recognition, 45(1), 521–530. doi:10.1016/j.patcog.2011.06.019
>
> [5] Lipton, Z., Wang, Y.-X., & Smola, A. (10--15 Jul 2018). Detecting and Correcting for Label Shift with Black Box Predictors. In J. Dy & A. Krause (Eds), Proceedings of the 35th International Conference on Machine Learning (pp. 3122–3130). Retrieved from https://proceedings.mlr.press/v80/lipton18a.html
>
> [6] Azizzadenesheli, K., Liu, A., Yang, F., & Anandkumar, A. (2019). Regularized Learning for Domain Adaptation under Label Shifts. International Conference on Learning Representations. Retrieved from https://openreview.net/forum?id=rJl0r3R9KX
>
> [7] Alexandari, A. M., Kundaje, A., & Shrikumar, A. (2020). Maximum likelihood with bias-corrected calibration is hard-to-beat at label shift adaptation. Proceedings of the 37th International Conference on Machine Learning. JMLR.org.
>
> [8] Wald, Y. & Saria, S.. (2023). Birds of an odd feather: guaranteed out-of-distribution (OOD) novel category detection. Proceedings of the Thirty-Ninth Conference on Uncertainty in Artificial Intelligence, in Proceedings of Machine Learning Research 216:2179-2191 Available from https://proceedings.mlr.press/v216/wald23a.html.

---

> > ### Comment · Reviewer_eCWt · 2026-03-13
> >
> > Thank you for the further explanation, your figure is clearer now.

---

### Author Response · Authors · 2026-02-20
**General comment summarising the reviews**

We thank all reviewers for their careful reading and constructive feedback. We are especially grateful for the positive evaluation of our contributions.

Reviewer **52XA** and **Tqac** **found our insights on under explored aspects of OSDA** with concurrent background distribution shift and novel classes **meaningful for the TMLR audience**. Reviewer **Tqac** emphasized the “simplicity of the ideas” proposed in the paper. We are glad that all the reviewers appreciated our extensive experiments across datasets and modalities backing the claims of the paper. Reviewer **52XA** appreciated our rigorous theoretical analysis and Reviewer **Tqac** positively engaged with the theoretical findings to further strengthen our claims. We appreciate this recognition of both the theoretical and empirical contributions.

Across the reviews, the main concerns fall into three categories. **We have carefully revised the paper to address all of these points**. Below we summarize the key comments by reviewers and the main changes introduced in the manuscript to address them:
* (Reviewer **Tqac**) **Relaxing theoretical constraints** on $\boldsymbol{\alpha}$, $N_{\mathcal{S}}$, $N_{\mathcal{T}}$. We explicitly addressed this by relaxing these constraints in Thm 1 in the revised draft.
* (Reviewers **Tqac**,**52XA**) **Additional experiments** over range and density of search grid $\boldsymbol{\alpha}$. To address this, we have added new results in tables 21-24 and figure 5. We have also added results on simulated experiments in figure 6 and tables 25 and 26 in the revised draft to study the impact of separability of novel classes on the performance of detecting novelties as suggested by Reviewer **52XA**.
* (Reviewer **eCWt**) **Organization of the manuscript**. We made several organizational changes, including adding a paper roadmap, and a notation section. We marked these changes in blue in the revision and elaborate on them in the response to the reviewer.

---

### Decision · Action_Editor_QYHV · 2026-04-10

**Recommendation:** Accept as is

**Additional Comments:**

All three reviewers are positive overall on the paper and have indicated they are leaning accept on accepting the paper for TMLR.  They have noted that the discussion and edits to the paper have been helpful in making the paper ready to be published.  The main lingering concerns deal with the gap between theory and practice (as noted by multiple reviewers in their recommendation) as well as the presentation, which one reviewer notes could be clearer.  I would recommend clarifying these points further for the final version, though I don't think we need to go through a review on a revision of the paper.

**Audience:**

Yes

**Audience Explanation:**

The general area of domain adaptation is a core problem in ML, with several existing papers in TMLR.  The fact that this paper treats open-set domain adaptation in a theoretical manner would definitely be of interest to many in the community, as well as the empirical results shown by the authors.

**Claims And Evidence:**

Yes

**Claims Explanation:**

Yes, the claims are supported.  In their initial reviews, the reviewers asked for additional evidence in terms of results.  The authors provided new results on grid density, ablation studies, and separability analysis.  The reviewers have indicated that the results are stronger now and have all supported publishing the paper.  The theoretical analysis seems to be correct.  In terms of clarity of presentation, reviewers initially had concerns but the authors have addressed these issues sufficiently after some back-and-forth with the reviewers (though see the additional comments below).